



# The future of Upernavik Isstrøm through ISMIP6 framework: Sensitivity analysis and Bayesian calibration of ensemble prediction

Eliot Jager[1,2], Fabien Gillet-chaulet[1], Nicolas Champollion[1], Romain Millan[1], Heiko Goelzer[3], and Jérémie Mouginot[1,†]

[1]IGE, Univ. Grenoble Alpes, CNRS, IRD, 38000 Grenoble, France
[2]Institute for Atmospheric and Earth System Research / Physics, Faculty of Science, University of Helsinki, Finland
[3]NORCE Norwegian Research Centre, Bjerknes Centre for Climate Research, Bergen, Norway
[†]deceased, 28 September 2022

**Correspondence:** Eliot Jager (eliot.jager@univ-grenoble-alpes.fr)

**Abstract.**

This study investigates the uncertain future contributions to sea-level rise in response to global warming of Upernavik Isstrøm, a tidewater glacier in Greenland. We analyze multiple sources of uncertainty, including shared socio-economic pathways (SSPs), climate models (global and regional), ice-ocean interactions, and ice sheet model parameters (ISM). We use weighting methods based on spatio-temporal velocity and elevation data to reduce ice flow model uncertainty, and evaluate their ability to prevent overconfidence. Our developed initialization method demonstrates the capability of Elmer/Ice to accurately replicate the historical mass loss of Upernavik Isstrøm. This provides confidence in the model's ability to project the future evolution of this region. Future mass loss predictions range from a contribution to sea level rise from 1.5 to 7.2 mm, with an already committed sea-level contribution projection from 0.6 to 1.3 mm. While all sources of uncertainty contribute at least 15% to uncertainty until the end of the century, SSP-related uncertainty dominates at 40%. We find that calibration does not reduce uncertainty of the future mass loss between today and 2100 of Upernavik Isstrøm (+2%) but significantly reduces uncertainty in the historical mass loss of Upernavik Isstrøm between 1985 and 2015 (-32 to -61% depending on the weighting method). Combining calibration of the ice sheet model with SSP weighting yields uncertainty reductions of future mass loss in 2050 (-1.5 %) and in 2100 (-32 %).

## 1 Introduction

The primary cause of present-day sea-level change is human-induced climate change, which will have far-reaching effects on coastal communities worldwide. To make informed decisions on protective measures, it is crucial to understand the extent and timing of sea-level rise. Predicting future local sea-level rise is a challenging task as it depends on many factors, such as the mean sea-level rise (SLR), ocean dynamics, local context and, of course, future mitigation of greenhouse gas emissions (Durand et al., 2022). As an important component to the local solution, it is essential to predict future mean sea-level rise for the end of the 21st century. As recent assessments by the Intergovernmental Panel on Climate Change have highlighted (Masson-Delmotte et al., 2021), future sea-level change is highly uncertain, especially the high-end scenarios. The main source



of uncertainty in SLR is the limited ability to model the future mass loss of the Antarctic and Greenland Ice Sheets (GrIS) due to the limited understanding of ice dynamics and climate forcings, as well as uncertainties in Ice Sheet Models (ISM) (Goelzer
et al., 2018; Seroussi et al., 2019; Goelzer et al., 2020; Seroussi et al., 2020).

To better understand uncertainties and enhance projections of the two ice sheets, a collective initiative has emerged: the Ice Sheet Model Intercomparison Project for CMIP6 (ISMIP6) framework (Nowicki et al., 2020). The outcomes of this endeavor have provided valuable insights into the behaviour of ISMs and the range of their variability. However, to improve estimates for decision-makers, Aschwanden et al. (2021) suggests two key areas of improvement. First, intrinsic uncertainties associated
with model structure, parameters, initial and boundary conditions must be better accounted for. Second, simulations should accurately reflect current observations within the limits of their uncertainty.

In addition of providing a more comprehensive quantification of uncertainties, sensitivity analyses play a crucial role in classifying uncertainties and prioritising their reduction. This approach has gained popularity in glaciology, as evident from case studies conducted with a single ISM in Antarctica (Hill et al., 2021) and Greenland (Aschwanden et al., 2019), as well as
the ISMIP6 analyses (Goelzer et al., 2020; Seroussi et al., 2020, 2023), which also facilitate the examination of model structure uncertainty through multiple ISMs. The first two individual ISM analyses revealed that the dominant origins of uncertainty were atmospheric forcings for Greenland and oceanic forcings for Antarctica. The ISMIP6 outcomes, in contrast, emphasize that uncertainties linked to ISMs persist significantly, akin to uncertainties originating from forcings and their application. These findings underscore the potential for reducing uncertainty in model projection by reconciling the differences among ISMs. In
this regard, a better use of observational data to calibrate these models and ensure their skill in reproducing recent data holds promise (Aschwanden and Brinkerhoff, 2022; Nias et al., 2023).

Calibration using observations has become a common practice in glaciology, as evidenced by previous studies on the SLR contribution from the GrIS (Applegate et al., 2012; McNeall et al., 2013; Chang et al., 2014; Aschwanden and Brinkerhoff, 2022; Nias et al., 2023), the Antarctic ice sheet (Gladstone et al., 2012; Ritz et al., 2015; DeConto and Pollard, 2016; Nias
et al., 2019; Gilford et al., 2020; Wernecke et al., 2020) or likewise the mountain glaciers (Rounce et al., 2023) and a review of the previous studies is made in the supplementary material of Aschwanden and Brinkerhoff (2022). These studies typically involve two steps: (i) quantifying uncertainties to establish an ensemble and projecting it into the future to forecast the future SLR contribution, and (ii) adjusting this ensemble by giving weights to the members according to their ability to reproduce past observations. However, these studies often employ all available observational data for calibration without incorporating
any form of validation to assess the improved performance of the calibrated ensemble compared to the non-calibrated one. This gives rise to concerns regarding the potential for overfitting and excessive confidence in future predictions of sea-level rise, especially in the context of a dataset of considerable size like ours.

In a previous study (Jager et al., 2024), the focus was directed towards investigating the ability of the ISM Elmer/Ice to replicate past variations of Upernavik Isstrøm (UI) during the period from 1985 to 2019. UI is a tidewater glacier situated in
the North-West sector of Greenland and is characterized by five distinct catchments: UI-NN, UI-N, UI-C, UI-S, and UI-SS (Figure 1), as named in Mouginot et al. (2019). The diverse dynamics of their front enable multiple tidewater glacier studies to be conducted within this comprehensive catchment, providing more robust results. Moreover, UI has experienced substantial





mass loss since 1985, contributing to 0.47 mm of sea-level rise, or more than 3% of Greenland's total contribution during this period, indicating significant temporal changes (Mouginot et al., 2019). The extensive satellite observations spanning 1985 to 2019 make UI an ideal candidate for evaluating the ability of a large-scale ISM to reproduce available observations of a local glacier. Furthermore, the pronounced spatial and temporal heterogeneity of this case study helps prevent unwarranted overconfidence in the model's performance.

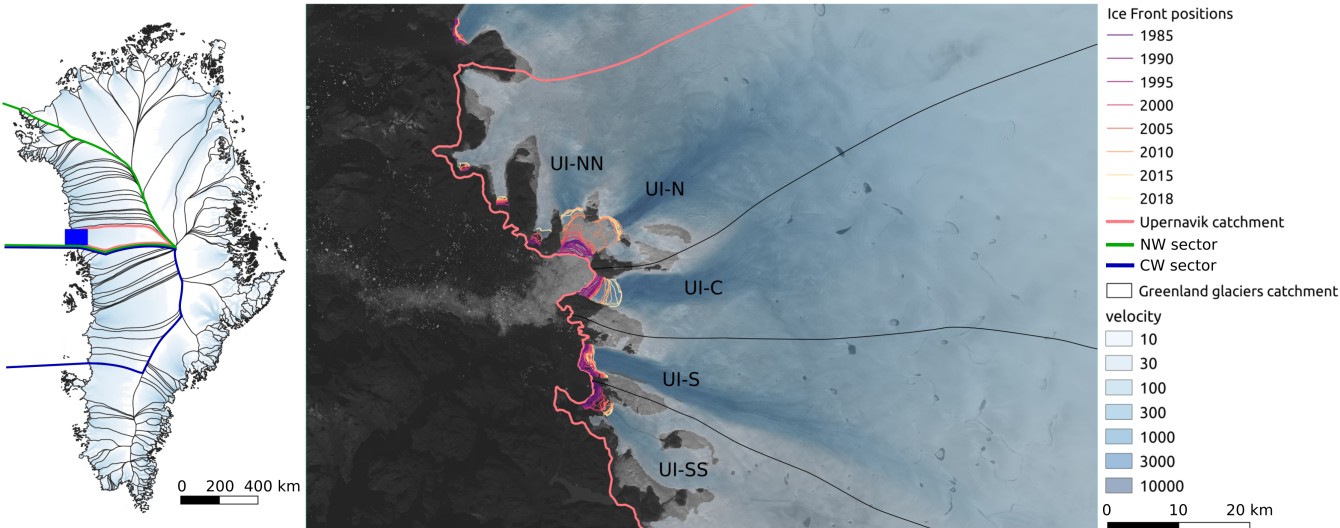

**Figure 1.** Left, GrIS drainage basins with the catchment of UI in pale red, NW in green and CW in blue sectors as defined in Slater et al. (2019). The blue box is the validation area shown in the right with the four different catchments (UI-N, UI-C, UI-S and UI-SS), the front positions between 1985 and 2018 (Wood et al., 2021) and the surface ice speed (Mouginot et al., 2019) overlaid on a Landsat image (2017-08-13). All the data collected are in this validation area.

To reproduce past changes of UI using Elmer/Ice, Jager et al. (2024) introduced a new initialisation method employing a model ensemble that incorporates various uncertainties within the ISM, including different basal friction field calibrations, initial surface elevation, and model parameters. Additionally, we prescribed the front positions and Surface Mass Balance (SMB) for each year. Subsequently, the performance of two ensembles, using two different basal friction relationships, was compared against a comprehensive dataset comprising spatio-temporal series of velocities and elevations, ice discharge, and mass loss. Jager et al. (2024) indicate the necessity of accounting for a reduction in friction near the glacier front to accurately reproduce these observational data. The sensitivity analysis, made possible by the ensemble approach, underscored the predominant role of the initial friction field compared to the initial surface or surface mass balance in shaping the historical variations of ice mass loss.

The objective of this study is to assess the UI's contribution to SLR throughout the 21st century and to enhance the quantification of associated uncertainties. The following aspects will be addressed: (i) projecting the future SLR contribution, (ii)



evaluating the sensitivity of SLR to the ISM compared to other sources of uncertainty, including forcings, and (iii) utilizing the extensive dataset for ISM calibration to ensure the reliability of the calibrated ensemble.

To address the first question, we adopt the ISMIP6 framework for the GrIS (Nowicki et al., 2020). This approach involves prescribing a SMB and the future positions of tidewater glacier fronts. The SMB is derived from a Regional Climate Model (RCM) that downscales outputs from an Atmosphere-Ocean Coupled Global Climate Model (AOGCM) associated with a specific Shared Socio-economic Pathway (SSP). Future front positions are estimated using a parameterisation that incorporates RCM runoff and AOGCM ocean temperatures as input variables, while allowing for consideration of different retreat sensitivities.

Regarding the second aspect, an exhaustive sensitivity analysis will be conducted by incorporating uncertainties associated with RCMs, AOGCMs, SSPs, front sensitivities, and the ice ISM itself. This analysis probes the impacts of these multiple sources of uncertainty on SLR and quantifies the potential reduction in uncertainty due to the ISM.

To address the last aspect, we propose several weighting methods and have designed a rigorous cross-validation approach to ensure robust calibration of the model ensemble. The validation process assesses the performance of the calibrated ensemble against independent data. Additionally, we investigate the sensitivity of the calibration to different assumptions, evaluating calibration performance through the validation procedure. Once the optimal calibration has been determined, we analyze the implications of this calibration on the selection of model parameters and its impact on SLR predictions. We also study the reduction potential when we change the weighting of the SSPs used.

## 2 Method

### 2.1 Model ensemble

In this section, we delineate the methodology employed for initialising and propagating the ensemble into the future, following a framework akin to the ISMIP6 framework for the GrIS using a single ISM (Nowicki et al., 2020). The chosen ISM for this investigation is Elmer/Ice, a parallel finite-element code (Gagliardini et al., 2013). Further details regarding the model characteristics can be found in Appendix A. This methodology entails conducting an ensemble of historical simulations, denoted as Historical prior ensemble (Hpr), covering the period from 1985 to 2019, using the initialisation methodology developed in Jager et al. (2024). Subsequently, we extend these ensemble members into the future from 2015 to 2100, employing two experiments based on the ISMIP6 framework: a control run with constant forcing, referred to as Control prior ensemble (Cpr), and a run with realistic forcing, termed Predicted prior ensemble (Ppr). We end by explaining how we propagate and characterize the uncertainty of the different sources identified in Hpr, Cpr and Ppr.

### 2.1.1 Historical prior Ensemble (1985-2019)

For the Historical prior Ensemble (Hpr), we use the same methodology than Jager et al. (2024) to create a model ensemble accounting for various uncertainties during the 1985-2019 period.





Regarding the SMB, we use directly the annual values provided by RACMO, forced by global reanalyses (Noël et al., 2018). Unlike our previous work, where MAR (Fettweis et al., 2017) was also employed, we focus solely on RACMO in this study. In our previous study, we demonstrated that ensemble members incorporating RACMO reproduced surface elevation observations better than those using MAR, while the choice of SMB model had minimal influence on the other model outputs analysed. Here, by using only RACMO, we improve the overall performance of the ensemble and restricts the parameter space to better
cover the other sources of uncertainty.

    The position of the UI calving fronts in the Hpr is prescribed at each time step by observations interpolated from Wood et al. (2021). However, as these observation uncertainties are small compared with the model mesh size (less than 60 m against more than 150 m), we are not taking this potential source of uncertainty into account in our analysis.

    For dynamics, we have identified several sources of uncertainty in our ISM and following the results of our previous study
(Jager et al., 2024), we have retained only those parameters with the most significant influence using factor fixing and removed parameter ranges that result in undesirable model behavior compared to observations using factor mapping. Factor fixing, or screening, is a technique to identify model components that have a negligible effect or make no significant contributions to the variability of the outputs or metrics of interest while factor mapping is a technique to identify which uncertain model factors lead to certain model behavior (see glossary of Reed et al. (2022)). More details on this parameter selection are given in the
appendix B1.

    Finally, the ISM uncertainty depends on 6 parameters, with two parameters influencing the calibration of the friction field ($\lambda_{reg}$ and $OBS_{inv}$), three parameters influencing the friction law ($f_{law}$, $f_{param}$ and $m$) and one parameter influencing the ice rheology ($E$).

### 2.1.2    Control prior Ensemble (2015-2100)

For the Control Prior Ensemble (Cpr), we extend the Hpr members into the future, keeping their parameter value and forcing (SMB and frontal position) constant. For the initial surface elevation, we use that of the Hpr member in 2015. For the SMB, we prescribe an average of RACMO between 1960 and 1990 to be consistent with the anomaly procedure (see below). For the front position, we use the observed front position of 2015 from Wood et al. (2021) and it remains constant. This ensemble starts from 2015 to be compared with the Predicted prior Ensemble.

### 2.1.3    Predicted prior Ensemble (2015-2100)

Regarding the Predicted Prior Ensemble (Ppr), we also extend the Hpr members into the future, maintaining their parameter values and starting from their 2015 state. However, several forcings are used for Ppr members compared to Cpr.

    For the SMB in the Ppr, we adopt the ISMIP6 framework for the GrIS (Nowicki et al., 2020; Goelzer et al., 2020). This approach employs an RCM to downscale an AOGCM associated with a specific SSP at the GrIS scale. These results are then
prescribed by anomalies which are added to the reference SMB used for the Cpr and with feedback on the elevation. The various combinations of SSP-AOGCM-RCM are presented in Table 1. To address the potential bias resulting from an over-representation of a particular SSP, we assign different weights to the various SSP-AOGCM-RCM combinations (see Table



**Table 1.** SSP-AOGCM-RCM combinations and their probabilities used in the Latin-Hypercube Sampling

| RCM | AOGCM | SSP | Probability |
|---|---|---|---|
| RACMO | CESM2 | SSP1-2.6 | 1/12 |
| RACMO | CESM2 | SSP2-4.5 | 1/6 |
| MAR39 | CESM2$^*$ | SSP5-8.5 | 1/12 |
| MAR312 | CESM2$^*$ | SSP5-8.5 | 1/12 |
| RACMO | CESM2$^*$ | SSP5-8.5 | 1/12 |
| MAR312 | MPI-ESM1-2-HR | SSP1-2.6 | 1/24 |
| MAR312 | MPI-ESM1-2-HR | SSP2-4.5 | 1/6 |
| MAR312 | MPI-ESM1-2-HR | SSP5-8.5 | 1/24 |
| MAR39 | CNRM-CM6-1 | SSP1-2.6 | 1/6 |
| MAR39 | CNRM-CM6-1 | SSP5-8.5 | 1/12 |

$^*$For practical purposes, this is the same physical model as CESM2 (CMIP6),
but a different ensemble member.

1), while trying to maintain balanced proportions between AOGCM and RCM. With these weights, we obtain the following proportions: 7/24 of SSP1-2.6, 8/24 of SSP2-4.5 and 9/24 of SSP5-8.5; 1/2 of CESM2, 1/4 of MPI-ESM1-2-HR and 1/4 of
CNRM-CM6-1; 1/3 of RACMO, 1/3 of MAR312 and 1/3 of MAR39.

In the Ppr, the front positions from 2015 to 2100 are initially computed on a Cartesian grid, based on the retreat information provided by the front parameterisation method described in Slater et al. (2019, 2020). Subsequently, these positions are interpolated onto the mesh. This parameterisation is contingent upon runoff data obtained from the RCM and ocean temperatures at the fjord outlet derived from the AOGCM. The distribution of a sensitivity parameter, designated $\kappa$, is calibrated for distinct
sectors over the period 1960-2018 as a function of these two variables. In our study, the UI is located in the north-western (NW) sector, situated just above the central-western (CW) sector (Figure 1). Therefore, given that the sensitivity of these two sectors is very different, we consider both sectors in our analysis. For the distribution of $\kappa$, we use three distinct levels: low, medium, and high. Specifically, the low sensitivity corresponds to 25% of the $\kappa$ values being smaller, the medium sensitivity represents 50% of the $\kappa$ values being smaller, and the high sensitivity includes 75% of the $\kappa$ values being smaller. To simplify,
the sensitivity of this front parameterisation is hereafter called front$_s$.

### 2.1.4  Propagation of uncertainty

Having identified the different sources of uncertainty, we proceed to propagate them through the model. It is important to note that, unlike other studies on ice sheets (e.g., Aschwanden et al., 2019), the SSP is not treated independently of the other sources of uncertainty. This makes it possible to quantify the contribution of this source to the total uncertainty, as done by Marzeion
et al. (2020) for glaciers.

For the Hpr and Cpr, only the uncertainties associated with the ISM are considered. To explore the various sources of uncertainty, we use a 200-member Latin hypercube sampling technique to cover the 10 different parameters, 6 ISM parameters



and 4 for the forcing (SSP, AOGCM, RCM and front$_s$). Subsequently, the initialisation method is applied to generate the 200 members of Hpr. For the 200 members of the Ppr and Cpr as already mentioned, we use output from the Hpr members, starting from the 2015 state.

We use the following indices to analyse the sensitivity of Ppr to the different parameters (Sobol, 2001):

$$S_i = \frac{\text{Var}\left(\text{E}\left[Y|X_i\right]\right)}{\text{Var}\,Y} \tag{1}$$

where $\text{Var}\,Y$ is the variance of an output $Y$ and $E[Y|X_i]$ is the expectation of having $Y$ given the parameter $X_i$. Here, $X_i$ is one of the 10 different parameters.

Accurately computing sensitivity indices usually requires a large number of simulations and methods have been developed to optimise the experimental design (Reed et al., 2022). However, due to the extensive computational demands of our model, conducting such a large number of simulations is impractical. Therefore, to simplify the approach, we opted to focus solely on first-order sensitivity indices (Eq. 4), which assess the individual impacts of the parameters without considering their interactions, which correspond to higher index orders. These sensitivity indices are computed using the ANOVA method (Brevault et al., 2013; Lamboni et al., 2011). When the probability distribution of $X_i$ is discrete, determining $\text{Var}\left(\text{E}\left[Y|X_i\right]\right)$ involves averaging the Y values for each value that $X_i$ assumes and then calculating the variance of the means of these distinct subgroups. For continuous probability distributions of $X_i$, we discretised the distribution, assuming only four distinct values for $X_i$. This approach simplifies the problem significantly but results in the loss of subtleties present in continuous distributions. Brevault et al. (2013) emphasized the importance of the number of levels chosen for discretisation. Different levels can lead to variations in the sensitivity indices, as small-scale variations in continuous parameters may be smoothed out during discretisation. So we tested the convergence of these indices as a function of the number of members considered, and found that they converged towards a value that changed by less than 3 percents from fifty members upwards. Given the primary objective of qualitative result discussion rather than precise estimation, the ANOVA method is well-suited for this purpose. Our focus remains on the identification of principal influences and overarching patterns, supported by the figures that offer approximate magnitudes of the observed effects.

## 2.2 Model ensemble evaluation

### 2.2.1 Observational data

To evaluate the performance of Hpr, we compiled an extensive dataset comprising observations of surface velocity, surface elevation, ice discharge, and ice mass loss.

For surface velocity and surface elevation, we used the same spatio-temporal data as presented in our previous work (Jager et al., 2024). These observational data have a grid resolution of 150 m and are annually averaged to improve spatial coverage. However, these data are somewhat unbalanced, exhibiting better coverage in both time and space from the 2010s onward compared to earlier years. To facilitate model-data comparison, the model fields are bilinearly interpolated onto the same regular grid as the observations.





The ice discharge data used in our analysis is a compilation of published data from Mankoff et al. (2019), King et al. (2018), and Mouginot et al. (2019). This data corresponds to the flow of ice through the gates, assuming that the average velocity over the thickness is equal to the observed surface velocity. As a result, the derived ice discharge data may exhibit variations depending on the positioning of the gates, the selection of ice heights, and the velocity measurements used. In addition, J. Mouginot has redone a set of discharges, this time using bedmachine rather than a flight line which we call MouginotV2, and

the data obtained are close to those of Mankoff et al. (2019). Our observation of Ice Discharge is then an average of these four dataset. For the model, we used the same methodology, by taking the gate defined in Mankoff et al. (2019).

The total ice mass loss at the catchment scale is assessed using the input-output method (Mouginot et al., 2019). This method entails subtracting the ice discharge from the RACMO surface mass balance. In the model, the volume is an output obtained by integrating the thickness over the entire active domain. Consequently, the change in volume encompasses the variations due

to front retreat, which are not considered in the input-output approach. Nevertheless, this change in volume was found to be negligible (less than 1%).

### 2.2.2   Metrics

To evaluate the performance of the Hpr, we use several ensemblist metrics. The Continuous Rank Probability Score ($CRPS$) measures the accuracy and sharpness (opposite of uncertainty/spread) of an ensemble, where lower values indicate improved

alignment between the ensemble mean and observations, as well as similarity between ensemble spread and observational uncertainty. To investigate whether changes in $CRPS$ result from a reduction in the difference between the ensemble mean and observations, we examine the Mean Absolute Error ($MAE$) of the ensemble mean. Similarly, to determine whether changes in $CRPS$ stem from alterations in the ensemble's sharpness, we analyze the spatio-temporal average of the standard deviation ($STD$) of the ensemble. Ultimately, the RMSE will serve as a metric for assessing the performance of individual ensemble

members, allowing us to calibrate the ensemble based on their respective performance.

$$CRPS = \frac{1}{n_{obs}} \sum_{j=1}^{n_{obs}} \int_{\mathbb{R}} \left( F_m^j(Q) - F_o^j(Q) \right)^2 dQ \qquad (2)$$

$$MAE = \frac{1}{n_{obs}} \sum_{j=1}^{n_{obs}} \left| \overline{Q}_m^j - Q_o^j \right| \qquad (3)$$

$$STD = \frac{1}{n_{obs}} \sum_{j=1}^{n_{obs}} \sqrt{\frac{1}{n_m} \sum_{i=1}^{n_m} \left( Q_{m,i}^j - \overline{Q}_m^j \right)^2} \qquad (4)$$

where $n_{obs}$ is the number of the different observations in the space and time, $n_m$ is the number of members, $Q$ is a phys-

ical quantity (velocity, elevation, ice discharge, change of volume), $\overline{Q}_m$ is the ensemble mean and $F_m(Q)$ is the cumulative distribution function of the ensemble. For the cumulative distribution function of the observation $F_o(Q)$, is common (Brown,



1974; Matheson and Winkler, 1976; Unger, 1985; Bouttier, 1994; Hersbach, 2000) to use the Heaviside function ($F_o(Q) = 0$ if $Q < Q_o$ and $F_o(Q) = 1$ if $Q > Q_o$, with $Q_o$ the observed quantity $Q$).

To evaluate the different members of the Hpr, we use the $RMSE$ :

$$RMSE_i = \sqrt{\frac{1}{n_{obs}} \sum_{j=1}^{n_{obs}} \left( Q_o^j - Q_{m,i}^j \right)^2} \tag{5}$$

### 2.3 Bayesian calibration

In the context of ice sheet forecasting, the focus lies on predicting the future contribution to global mean sea level rise ($SLR$) while leveraging a diverse array of information, including models, observations, and previous studies. In this study, we adopt the formalism introduced by Aschwanden and Brinkerhoff (2022), wherein they update a prior distribution of future $SLR$ by considering a model ensemble $\mathbf{M}$ consisting of $n_m$ members traversing the parameter space $\Sigma$, a collection of untraversed model assumptions $\mathcal{H}$, the evolution of external forcings $\mathcal{F}$, and a set of observations $\mathcal{B}$.

$$\underbrace{P(SLR|\mathcal{H},\mathcal{B},\mathcal{F})}_{\text{Posterior}} = \int_{\Sigma} \underbrace{P(SLR|\mathbf{M},\mathcal{H},\mathcal{F})}_{\text{Prior}} \cdot \underbrace{P(\mathbf{M}|\mathcal{B})}_{\text{Calibration}} \mathrm{d}\mathbf{M} \tag{6}$$

It is pertinent to acknowledge that in practical applications, SLR can be substituted with any other variable of interest, encompassing velocities, surface elevations, ice discharge, volume change, or even model parameter values. By not utilising all the observations for calibration, it is then possible to assess whether the calibration has improved the model ensemble utilising the CRPS (Eq. 2). Following the elucidation of our calibration methodology, we will delineate the process for conducting this validation.

#### 2.3.1 Weighted bootstrap

To compute the calibration term $P(\mathbf{M}|\mathcal{B})$, we employ an ensemble sampling method named weighted bootstrap (Smith and Gelfand, 1992), with particles $\mathbf{M}_i$ corresponding to different members, allowing us to approach the Bayesian problem:

$$P(\mathbf{M}|\mathcal{B}) = \sum_{i=1}^{n_m} w_i \cdot \delta(\mathbf{M} - \mathbf{M}_i) \tag{7}$$

with $\delta$ the Dirac function and $w_i = \frac{P(\mathcal{B}|\mathbf{M}_i)}{\sum\limits_{k=1}^{n_m} P(\mathcal{B}|\mathbf{M}_k)}$ the weight of the member $\mathbf{M}_i$ which represents the likelihood of having had the observation with member $\mathbf{M}_i$ and which is therefore higher for the members that are closest to the observations.

It is commonly assumed that the $n_{obs}$ observations $b_j$ of $\mathcal{B}$ are independent (Aschwanden and Brinkerhoff, 2022; Nias et al., 2023) and the distribution of $P(b_j|\mathbf{M}_i)$ is gaussian (Nias et al., 2023) leading to :



$$P(\mathcal{B}|\mathbf{M}_i) = P\left(\bigcap_{j=1}^{n_{obs}} b_j|\mathbf{M}_i\right) \tag{8}$$

$$= \prod_{j=1}^{n_{obs}} P(b_j|\mathbf{M}_i) \tag{9}$$

$$= \prod_{j=1}^{n_{obs}} A\exp\left[-\frac{(b_j - H_j(\mathbf{M}_i))^2}{2\sigma^2}\right] \tag{10}$$

$$= A\exp\left[-\sum_{j=1}^{n_{obs}} \frac{(b_j - H_j(\mathbf{M}_i))^2}{2\sigma^2}\right] \tag{11}$$

$$= A\exp\left[-\frac{n_{obs}}{2} \cdot \frac{RMSE_i^2}{\sigma^2}\right] \tag{12}$$

and

$$w_i = \frac{\exp\left[-\frac{n_{obs}}{2} \cdot \frac{RMSE_i^2}{\sigma^2}\right]}{\sum_{k=1}^{n_m} \exp\left[-\frac{n_{obs}}{2} \cdot \frac{RMSE_k^2}{\sigma^2}\right]} \tag{13}$$

where $H(\mathbf{M}_i)$ is the measurement operator corresponding to $Q_{m,i}$ in Eq. 5, which projects the state of the model onto observation $b_j$, and $\sigma$ is the standard deviation of the observation error, which is assumed to be constant.

In our study, Equation 12 presents two primary limitations. First, due to the substantial volume of data at hand, we encounter a challenge similar to particle filters framework, which tend to retain only one member, leading to overfitting (Leeuwen, 2010). To overcome this issue, a considerable number of ensemble members comparable to the number of observations is necessary. However, achieving such a large ensemble size proves impractical in this case, as the number of observations exceeds four million, even with a surrogate model as proposed in Aschwanden and Brinkerhoff (2022). Secondly, the assumption of independent observations is difficult to justify, given the strong temporal and spatial correlations of velocities and surface elevations. Higher values observed at one grid point or time step are likely to be similarly high at adjacent locations or subsequent time steps.

To address the challenge of spatial and temporal correlation, we use a performance metric approach, which uses a metric to evaluate the distance between the observed and modeled fields, effectively treating the multiple observations as a single observation (Pollard et al., 2016; Bondzio et al., 2018; Albrecht et al., 2020). This method substantially diminishes the influence of observations, thereby mitigating the risk of overfitting while potentially introducing underfitting, as previously identified by Wernecke et al. (2020). This performance metric is applicable across various model outputs, encompassing velocity, surface elevation, ice discharge, and cumulative ice discharge. Furthermore, it can be computed for each sub-basin (UI-N, UI-C, UI-S, and UI-SS, as illustrated in Figure 1), and potentially, for distinct sub-periods (as detailed below). Consequently, we establish the weights through the following procedure:





$$w_i = \frac{\prod\limits_{s=1}^{n_s} f(RMSE_{i,s}, \sigma)}{\sum\limits_{j=1}^{N} \prod\limits_{s=1}^{n_s} f(RMSE_{k,s}, \sigma)} \tag{14}$$

with $f(RMSE, \sigma)$ a probability density (e.g., gaussian gives $f(RMSE) = \exp\left[-\frac{1}{2}\frac{RMSE^2}{\sigma^2}\right]$) which depends on the parameter $\sigma$ and $n_s$ the number of different RMSEs used (i.e., the product of the number of outputs used, the number of sub-basins and the number of sub-periods), which must be independent (e.g., an RMSE on UI-N is quasi-independent of the RMSE on UI-C).

In this equation, $n_s$ denotes the count of distinct RMSE values utilised (i.e., the product of the number of output variables employed, the number of sub-basins, and the number of sub-periods) which have to be independent (e.g., the RMSE associated with UI-N should be largely independent of the RMSE associated with UI-C). $f(RMSE, \sigma)$ represents a probability density function (e.g., Gaussian, denoted as $f(RMSE, \sigma) = \exp\left[-\frac{1}{2}\frac{RMSE^2}{\sigma^2}\right]$), which depends on the parameter $\sigma$. This parameter $\sigma$ encapsulates the discrepancy variance, representing a composite of both observational and structural model errors. This variance characterizes the disparity between the model, optimized for the best parameter set, and real-world observations (Murphy et al., 2009; Nias et al., 2019; Edwards et al., 2019). However, accurately quantifying structural error remains a persistent challenge, often necessitating retrospective estimation. To mitigate this limitation, we adopt an assumption that leverages the distribution of RMSE values to estimate $\sigma$. Specifically, the minimum RMSE value serves as the lower bound for potential $\sigma$ values, encompassing both model and data errors. Remarkably, this minimum RMSE corresponds to the configuration with the least structural error concerning the observed data. This assumption aligns with the weighting methodology utilized in previous studies, such as Pollard et al. (2016) and Albrecht et al. (2020), where the median is employed as an estimate for $\sigma$.

### 2.3.2 Evaluation of weighting choices

The heart of this Bayesian calibration is the calculation of weights (Eq. 14). Several choices are possible for calculating them, and to assess the performance of these different choices, we have developed a cross-validation method. This process entails computing weights and calibrating the ensemble using data from three out of the four sub-basins, and subsequently employing the ensemble-based metrics previously defined (CRPS, MAE, and STD) to appraise the performance of the posterior ensemble with respect to the fourth sub-basin. These metrics are normalised with metrics obtained with the prior ensemble, wherein a value exceeding 1 signifies inferior performance of the posterior ensemble in contrast to the prior ensemble (Hpr), whereas a value less than 1 signifies enhanced performance.

The evaluation encompasses three distinct weighting approaches:

1. Full-period weighting

2. Sub-period weighting

3. $f_{param}$ weighting





In the context of Full-period weighting, which employs Eq. 14, several assumptions are examined:

1. The selection of probability density: Gaussian, following Nias et al. (2023), or Student's-t, as in Aschwanden and Brinkerhoff (2022).

2. The choice of $\sigma$ estimate: minimum, mean, or median of the RMSE distribution.

3. The choice of data source: surface elevations, surface velocities, ice discharge, or cumulative ice discharge.

*Full-period weighting*

In the case of full-period weighting, the weighting of ensemble members depends on their ability to, on average, replicate the temporal evolution of various sub-basins throughout the entire period from 1985 to 2019. To determine the final weight, we compute the RMSE for each sub-basin over the entire observation period and then apply Equation 14 to combine these RMSEs.

*Sub-period weighting*

In the case of sub-period weighting, the weighting of ensemble members depends on their ability to, on average, replicate the temporal evolution of different sub-basins across various sub-periods, such as the pre-retreat, retreat, and post-retreat periods. To accomplish this, distinct RMSE values are calculated for each combination of sub-basin and sub-period. For instance, for UI-N, RMSEs are computed for the periods 1985-2004, 2004-2010, and 2010-2019, while for UI-C, RMSEs are determined for the periods 1985-2009, 2009-2015, and 2015-2019 (see evolution of front on figure 1). Conversely, for sub-basins UI-S

and UI-SS, RMSEs are assessed over the entire period. To determine the final weight, we apply Equation 14 to combine all these RMSEs. Similar to the full-period weighting approach, the assessment of the posterior ensemble through cross-validation employs ensemble metrics spanning the entire period from 1985 to 2019.

*$f_{param}$ weighting*

In the case of $f_{param}$ weighting, the weighting of ensemble members depends on the presence or the absence of the parame-

terisation of the sub-hydrology effect on friction (Eq. A3). We then give a weight of $w_i = w$ for members with parameterisation ($f_{param}$ =True) and a weight $w_i = (1 - w)$ for members without parameterisation ($f_{param}$ =False) and test different values of $w$ (0.6, 0.7, 0.8, 0.9 and 1). Then we evaluate the performance of this ensemble with the CRPS, MAE, and STD.

This alternative weighting approach was investigated based on insights from a previous study, which demonstrated that the model's ability to reproduce observation data improved significantly when accounting for the reduction in friction near the

front. This also allows us to see the effect of our parameterisation in terms of the predicted future sea level rise contribution of Upernavik Isstrøm.

### 2.3.3  SSP weighting

Moreover, an exploration was conducted to investigate the potential of assigning weights to ensemble members based on an alternative hypothesis in our dataset, specifically the choice of SSPs. As mentioned previously, unlike other studies on the

polar ice caps, the SSP is not treated independently of the other sources of uncertainty in order to quantify the contribution of





this source to the total uncertainty. Without any preconceived ideas about their distribution, we have assigned an almost equal weighting to each SSP (see section 2.1.3). However, recent evidence suggests that each SSP is not equally likely to occur in the future, with higher probabilities associated with scenarios projected to reach 2 to 3.5°C of warming by 2100 (Raftery et al., 2017; Hausfather and Peters, 2020; Intergovernmental Panel on Climate Change (IPCC), 2022; Hausfather and Moore, 2022;

Jr et al., 2022). Drawing from the survey results presented in Tollefson (2021), we propose allocating probabilities of 1/10, 6/10, and 3/10 to SSP5-8.5 (representing more than or equal to 4°C of warming), SSP2-4.5 (indicative of warming between 2.5°C and 3.5°C), and SSP1-2.6 (corresponding to warming below or equal to 2°C), respectively. This weighting approach reflects the challenges associated with achieving SSP5-8.5 under current policies (Intergovernmental Panel on Climate Change (IPCC), 2022), which leads us to give more weight to SSP2-4.5. Similarly, SSP1-2.6 is deemed improbable due to the limited

extent of CO2 emission reductions to date (Raftery et al., 2017). However, it is important to acknowledge that these weights are approximate estimates and should not be taken at face value. By construction, this weighting has no effect on the Hpr, and its performance in terms of CRPS, MAE, and STD is not evaluated.

## 3   Results

To comprehensively assess the future sea-level rise contribution of Upernavik Isstrøm, we have organized the results section

into four key subsections. In the first subsection, we project the ensemble into the future after using the initialisation method established in Jager et al. (2024). This investigation sets the stage for determining a reference sea-level rise contribution and understanding the components of Upernavik Isstrøm's mass loss, with a particular focus on ice discharge and surface mass balance. It also highlights disparities between the Predicted prior Ensemble and the Control prior Ensemble. In the second subsection, we dissect uncertainty within the Predicted prior Ensemble, looking at the importance of different sources

of uncertainty such as Shared Socioeconomic Pathways (SSP), Atmosphere-Ocean General Circulation Models (AOGCM), Regional Climate Models (RCM), frontal sensitivity ($front_s$), and the Ice Sheet Model (ISM). This analysis underscores the potential of ISM calibration to reduce overall uncertainty effectively. In the third subsection, we explore the sensitivity of our Bayesian calibration methodology, assessing the conditions under which it enhances ensemble performance and its impact on the selection of six ISM parameters ($\lambda_{reg}$, $OBS_{inv}$, $f_{law}$, $f_{param}$, $m$, and $E$). In the final subsection, we look at the difference

between posterior and prior ensembles in term of sea level rise contribution. This will enable us to see how Bayesian calibration reduces the uncertainty when overlearning is avoided.

### 3.1   Prior ensemble

In figure 2, the ice mass change is depicted relative to 2015 for observations, for the historical prior ensemble (Hpr), for the predicted prior ensemble (Ppr) and for the control prior ensemble (Cpr).

During the historical period spanning from 1985 to 2019, the Historical prior Ensemble (Hpr) yields a median mass loss of 200 Gt, ranging from 100 to 250 Gt (95% confidence interval). The Hpr median reproduces very faithfully the observations. This result confirm the ability of the methodology established in Jager et al. (2024) to reproduce past observations.





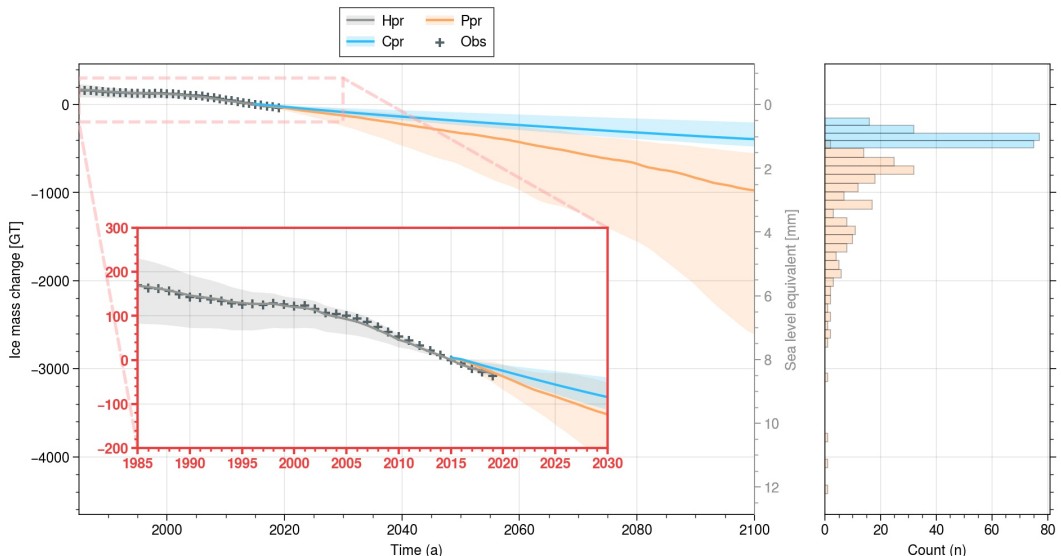

**Figure 2.** UI ice mass change relative to 2015 for Hpr (grey), Ppr (orange) and Cpr (blue). For each ensemble, the mean is represented in solid line and the shading include 95% of the ensemble members. Observations of the 1985-2019 period are represented by +. The red box shows a zoom to 1985-2030 period. The histogram on the right illustrates the distribution of Cpr and Ppr concerning the UI contribution to sea level rise spanning from 2015 to 2100.

By 2015, the UI had already contributed by 0.47 [0.23, 0.64] mm to sea level rise (SLR) since 1985, and the mass loss of Cpr and Ppr is projected to add an additional 1.1 [0.6, 1.3] mm and 2.7 [1.5, 7.2] mm, respectively, by 2100. Notably, the most
extreme values of the Ppr indicate a contribution to SLR exceeding 10 mm, while the majority of Ppr values range from 1 to 3.5 mm. It is worth noting that the distribution's tail for values above this interval is wider than for values below. Finally, the difference between Ppr and Cpr, i.e., the loss of mass due to future warming, gives us an additional contribution to SLR of 1.7 [0.7-6.3] mm.

The SMB and the ice discharge have two opposite trends at the end of the century (figure 3). Until the 2090s, some members
following the SSP5-8.5 see their discharge increasing sharply, reaching high values of 60 Gt/a, but with a sharp decrease between 2090 and 2100. We attribute this late period decrease to the fact that 2 of the 3 marine-terminated glaciers of the UI catchment become land-terminated from this point onwards for members with large retreat forcings. On the other hand, the median SMB remains close to current levels at around 6 Gt/a (mass gain) until the 2050s, before falling slowly to around 3 Gt/a. In 2050, members forced by SSP5-8.5 start to have a negative SMB, which becomes permanently negative from 2070
onwards. Looking at the discharge and SMB of Cpr, it can be seen that UI has still not reached an equilibrium in 2100, with a discharge of 13 [11.1,13.9] Gt/a, while the SMB is 9 Gt/a, resulting in a negative mass balance.



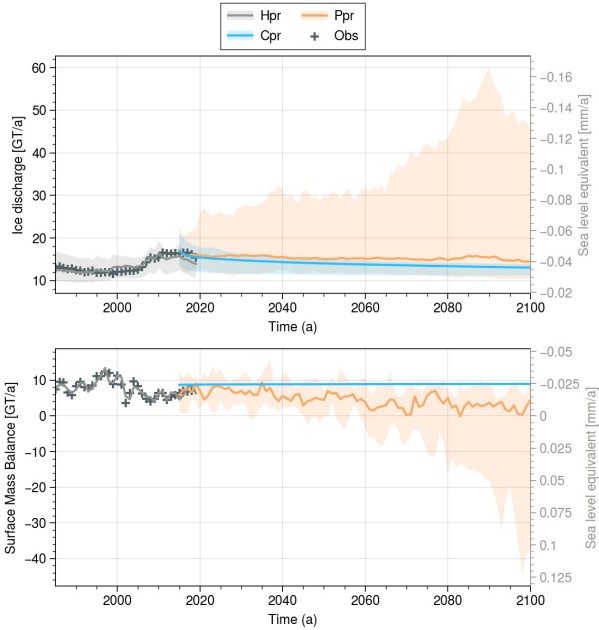

**Figure 3.** UI ice discharge and SMB over the period 1985-2100 for the historical (grey), the predicted (orange) and the committed mass loss (blue) ensemble simulations. For each ensemble, the mean is represented in solid line and the shading include 95% of the ensemble members. Observation from Mouginot et al. (2019) of the 1985-2019 period are represented by +.

## 3.2 Sensitivity analysis

Figure 4 depicts the evolution from 2015 to 2100 of the sensitivity indices computed with the Predicted prior Ensemble (Ppr), for the volume, the ice mass change, the cumulative SMB and the cumulative ice discharge. Sensitivity to ice mass change is equivalent to the sensitivity of UI's contribution to SLR. To make things simpler, we sum all the indices influencing the ISM ($f_{law}$, $f_{param}$, $m$, $E$, $\lambda_{reg}$, $OBS_{inv}$) and compare them with the indices associated with the SSP, AOGCM, RCM, and front parameterisation. Neglecting the sensitivity indices of the parameter combinations leads to a small underestimation of the impact of the dynamics, since part of its influence comes from the parameter combinations.

The sensitivity indices provided in the figure are presented in their non-normalized form. It should be emphasized that a sum of sensitivity indices less than 1 means a substantial impact of specific parameter combinations, e.g. the fact that the influence of the combination of emission scenario and front sensitivity is stronger than the sum of the influences of each, due to non-linearities. On the contrary, a sum greater than 1 implies interdependencies between input parameters, e.g. the fact that SSP, AOGCM and RCM are not independent in our case.

As expected, in 2015 the initial volume is independent of the choice of SSP, RCM, AOGCM and front$_s$ (Figure 4a) and the sum of the ISMs sensitivity indices is equal to 0.65. The value being smaller than 1 is attributed to the interactions between





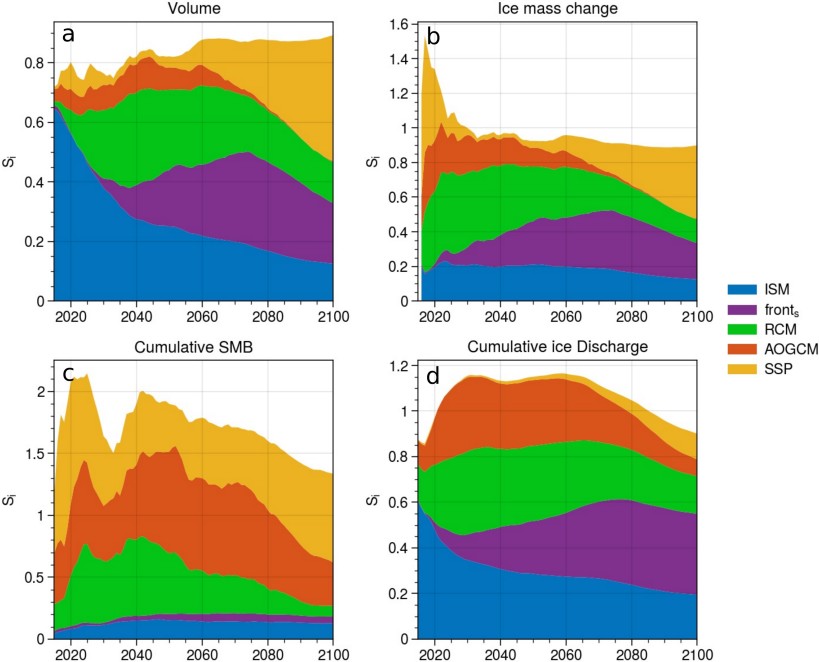

**Figure 4.** Sensitivity indices for the five sources of uncertainty of Ppr (the dynamics, the front parameterisation, the RCM, the AOGCM and the SSP) for the volume (a), the ice mass change relative to 2015 (b), the cumulative SMB since 2015 (c) and the cumulative ice discharge since 2015 (d).

various ISM parameters. The influence of the ISM only diminishes as we move away from this initial state, with the influence of the other sources increasing with very different characteristics.

By 2040, the RCM exhibits the most significant increase in influence on the volume, with a sensitivity index of 0.3, equal to that of the ISM (Figure 4a). Subsequently, from 2040 to 2075, the sensitivity indices associated to RCM and ISM gradually decreases to 0.2. During this period, the influence of the AOGCM diminishes from 0.1 to 0. Conversely, the sensitivity index associated with the front parameterisation experiences the most pronounced increase, rising from 0.1 in 2040 to 0.3 in 2075.

Beyond 2075, the sensitivity indices of the ISM on the volume, front parameterisation, and RCM gradually decline until they reach 0.1, 0.2, and 0.1, respectively (Figure 4a). Meanwhile, the impact of the SSP starts to emerge, becoming non-negligible in the 2050s and significantly accelerating from 2070 onwards. By 2100, the SSP becomes the most influential parameter, with a sensitivity index of 0.45. Throughout this period, the influence of the AOGCM remains at zero.

For the ice mass change in the year 2100, the impact of the parameters exhibits similarities to their influence on total volume, contrasting with cumulative SMB and cumulative ice discharge (refer to Figure Figure 4b,c,d). Specifically, for total mass loss, the influence of the SSP is substantial (0.4), while the front parameterisation (0.2), the RCM (0.1), and the ISM (0.15) also exhibit discernible but lesser effects. In contrast, the AOGCM demonstrates no discernible influence on total mass loss.





As anticipated, the cumulative ice discharge is primarily influenced by the ISM parameters and the front parameterisation front$_s$, which demonstrate the most pronounced impact. Additionally, the roles played by the SSP, AOGCM, and RCM are not negligible. The combined sensitivity indices of ISM and front$_s$ exhibit a peak value of 0.6 by 2075 and 0.55 by 2100. This heightened influence is also reflected in ice mass loss, with a peak sensitivity indices sum of 0.5 in 2075. In contrast, the sensitivity indices of SSP, AOGCM, and RCM peak at 0.65 in 2030 and gradually decrease to 0.35 towards the later stages.

Conversely, the cumulative SMB demonstrates strong sensitivity almost exclusively to the SSP, AOGCM, and RCM, with their sensitivity indices reaching approximately 2 at the maximum and 1.1 towards the end of the analysis period. For the ISM and front$_s$, their influence on cumulative SMB remains limited, with sensitivity indices not exceeding 0.2 (maximum 0.15 for dynamics and 0.05 for front parameterisation), owing to feedback interactions with elevation and ice-covered area.

Significant changes in the influence of the SSP emerge since 2050, notably impacting the cumulative ice discharge, the total
mass loss, the volume and the cumulative SMB. Except for the cumulative SMB, the SSP influence is almost zero before 2050, before becoming the most important parameter after 2090 for both the total mass loss, the volume, and the cumulative SMB.

The influence of the AOGCM demonstrates an intriguing trend. From 2050 to 2080, the AOGCM's impact gradually decreases until it reaches zero for ice volume and ice mass change. Concurrently, its effect on cumulative SMB and cumulative ice discharge also diminishes, though it never reaches zero. This intriguing behavior is a result of an equilibrium phenomenon,
where AOGCMs with the smallest surface mass balance gains correspond to those associated with the lowest ice discharge losses.

Concerning the sensitivity indices of the ISM parameters for volume, ice mass loss, cumulative SMB, and cumulative ice discharge, the friction parameterisation $f_{param}$ exhibits the highest significance at the end of the analysis period (Fig. B1). In 2100, its sensitivity index is 0.06 for volume, 0.05 for ice mass loss, and 0.1 for cumulative ice discharge, i.e., at least a third of
the ISM total. Additionally, the observation used for the friction calibration $OBS_{inv}$ has a substantial impact at the beginning, with a sensitivity index of 0.5 for volume. However, its influence gradually diminishes over time and becomes negligible by 2080 (less than 0.01). Lastly, the parameter $m$ emerges as another significant factor in 2100 for dynamics, with a sensitivity index of 0.04 for volume, ice mass loss, and cumulative ice discharge.

## 3.3 Calibration

### 3.3.1 Cross-validation

In Appendix C, we present the outcomes of our validation process for the weighting methodologies employed to prevent overfitting and excessive confidence in our new ensemble. We utilized the Continuous Ranked Probability Score (CRPS), a standard metric for ensemble assessment, on the validation set across various observed fields such as velocity, surface elevation, ice discharge, and cumulative ice discharge. The analysis of weighting sensitivity unveiled the following key findings:

1. Weighting using the Student distribution generally exhibited superior performance compared to the Gaussian distribution: by assigning less preference to the best members, the Student distribution effectively reduced total variance and mitigated overfitting. Similarly, increasing $\sigma$ led to reduced emphasis on the best members and aided in avoiding overfit-





ting. However, excessively high values of $\sigma$ resulted in decreased CRPS performance due to underfitting. We determined that an optimal compromise is achieved by utilizing the median or mean of the RMSE distribution for determining $\sigma$.

2. Utilizing surface elevations and velocities in the weighting process yielded the most robust outcomes, reducing CRPS across all variables. Weighting solely based on ice discharge or volume change improved CRPS for these specific quantities but not for surface velocities and elevations.

3. Introduction of multiple periods into the weighting process enhanced CRPS for volume changes but not for surface elevation and velocity, as it excessively reduced overall variance. To mitigate this effect, it is advisable to increase
$\sigma$ by selecting the 3rd quartile of the RMSE distribution, thereby balancing undesirable reductions in variance while preserving desirable outcomes.

4. The $f_{param}$ weighting scheme generally yielded superior CRPS for volume change and ice discharge but exhibited poorer performance for surface elevation and velocity compared to alternative weighting approaches.

### 3.3.2   Factor mapping

Following the evaluation of calibration's influence on ensemble performance, we explore its repercussions on the selection of Ice Sheet Model (ISM) parameters. This process aligns with the previously established factor mapping, facilitating the identification of parameter ranges yielding model outputs closely aligned with observations.

Presented in Figure 5, the distribution of the six ISM parameters, $\lambda_{reg}$, $OBS_{inv}$, $f_{param}$, $f_{law}$, $m$, and $E$, is depicted both for prior and after calibration applying Full-period weighting and Sub-period weighting. For Full-period weighting, we adopt a
Student's distribution with the median as the estimate of $\sigma$, along with the integration of he combination of velocity and surface elevation data (ZSxV). In the case of Sub-period weighting, we maintain these characteristics, except for the $\sigma$ estimate, which is determined by the 75th percentile (SP_Q75) of the RMSE distribution.

While the primary findings are presented herein, additional details can be accessed in appendix B3:

1. Full-period weighting favors members initialised with friction data from the 1990s and 2000s due to lower RMSE values,
while members with inversions conducted in 2010 or 2017 exhibit poorer performance, particularly in ice-free areas pre-retreat, due to extrapolation needs. Confidence in predictions increases as data is faithfully reproduced after the ice front retreats.

2. The presence or absence of $f_{param}$ is a significant factor influencing the distribution shift between prior and posterior in Full-period weighting, highlighting its crucial role in accurately reproducing data.

3. Excessively high regularization weight ($\lambda_{reg}$) values result in elevated RMSEs due to overly smooth friction fields, emphasizing the importance of balancing regularization strength and model fidelity.

4. Parameters $m$, $law$, and $E$ show no substantial trends in the difference between prior and posterior distributions. However, higher weights are observed for certain values of $m$, $E$, and $law = W$ due to the influence of $\lambda_{reg}$, $OBS_{inv}$, and $f_{param}$.





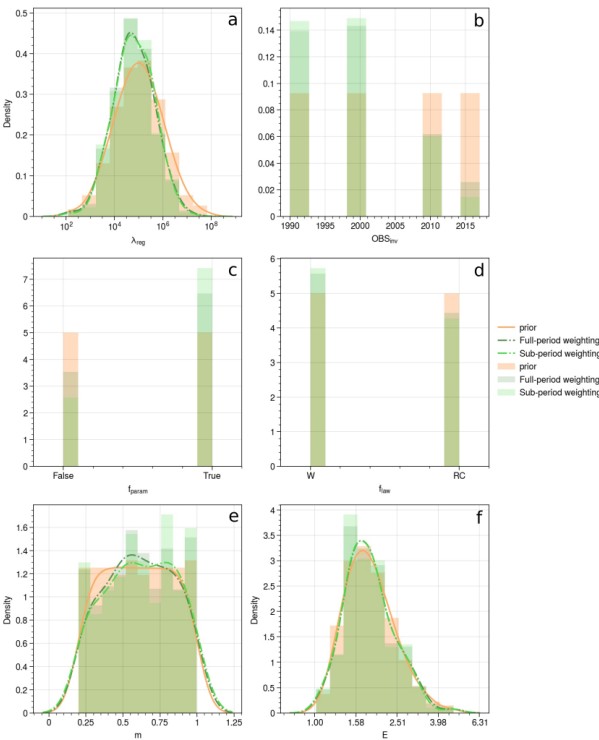

**Figure 5.** Distribution of $\lambda_{reg}$ (a), $OBS_{inv}$ (b), $f_{param}$ (c), $f_{law}$ (d), $m$ (e), and $E$ (f) for the prior ensembles (Hpr, Ppr and Cpr) and for the calibrated ensembles (Hpo, Ppo and Cpo) with Full-period and Sub-period weightings.

5. Sub-period weighting amplifies discrepancies in $f_{param}$ selection, indicating a greater likelihood for accurately replicating distinct periods. This indicates that the members that best reproduce changes in dynamics are those that use parametrisation proposed in Jager et al. (2024).

## 3.4 Posterior ensemble

In this section, we delineate the sea level rise (SLR) outcomes resulting from the previously discussed Bayesian calibration.
Initially, we present the findings pertaining to ISM bayesian calibration approaches: Full-period weighting, Sub-period weighting, and $f_{param}$ weighting. Subsequently, we examine the outcomes when the SSPs are weighted differently than our a priori assumptions, employing our SSP weighting presented in section 2.3.3. Given that the uncertainty associated with SSPs is most pronounced in 2100 (Fig. 4), this analysis enables us to assess the extent to which ISM bayesian calibration can subsequently mitigate uncertainty regarding SLR.

Figure 6 illustrates the ice mass change relative to 2015 for various ensemble configurations and weightings. The depicted ensembles encompass observations (Obs), the historical prior Ensemble (Hpr), the Predicted prior Ensemble (Ppr), the Historical posterior Ensemble (Hpo), and the Predicted posterior Ensemble (Ppo). These posterior ensembles (Hpo and Ppo)





are calibrated using the bayesian calibration (see section 2.3) under different weightings, including the Full-period weighting, $f_{param}$ weighting, Sub-period weighting, and SSP weighting. For Full-period weighting, we adopt a Student's distribution with the median as the estimate of $\sigma$, along with the integration of a combination of velocity and surface elevation data (ZSxV). In the case of Sub-period weighting, we maintain these characteristics, except for the $\sigma$ estimate, which is determined by the 75th percentile (SP_Q75) of the RMSE distribution. Additionally, Figure 7 presents a similar representation for the combined Sub-period and SSP weightings, achieved by multiplying the weights of these two weightings.

### 3.4.1 ISM weighting

Throughout the historical period, weightings based on ISM performance over the period 1985-2019, as the Full-period, $f_{param}$, and sub-period weightings, has considerably narrowed the mass loss distribution around the observations (Figure 6). This narrowing of the distribution is particularly pronounced for $f_{param}$ weighting (-51% of the 95% confidence interval in 1985) and sub-period weighting (-61%), surpassing that achieved by Full-period weighting (-32%). For $f_{param}$ weighting, the notable reduction in uncertainty mainly arises from adjusting the weights assigned to members with the greatest mass loss, rather than to members with the lowest mass loss. Conversely, the opposite trend is observed for the other two weighting methods. This second pattern is attributed to the selection of members based on the year of inversion (see Figure 5), rather than the presence or absence of $f_{param}$ parameterisation. Specifically, members initialised before the retreat and not employing $f_{param}$ show lower mass losses, which are less consistent with the observed data. Sub-period weighting emerges as a compromise between the other two weighting approaches. It incorporates the more precise selection criterion of $f_{param}$ weighting while retaining the inclusion of lower members, similar to the members using inversion data before the retreat.

The results of the mass loss analysis of the posterior control ensemble (Cpo), are not shown in this section, but these results present similar trends to those observed over the historical period. In particular, the reduction in uncertainty is greater for the $f_{param}$ weighting and sub-period weighting approaches. In these cases, the main impact is the exclusion of ensemble members characterized by the lowest mass loss, leading to projected contributions in 2100 of 0.83 to 1.31 and 0.83 to 1.25 mmSLE, respectively. This contrasts with the prior ensemble (Cpr), which ranges from 0.56 to 1.31 mmSLE. For the full-period weighting approach, the uncertainty reduction is more symmetrical, affecting the ensemble members with the highest and lowest mass loss, resulting in a range of 0.64 to 1.26 mmSLE.

Regarding the prediction for the year 2050 and 2100, both the Full-period weighting and sub-period weighting methods exhibit minimal changes in the posterior ensemble, as depicted in Figure 6. The median contribution of Upernavik Isstrøm to sea level rise by the end of the century remains unchanged at 2.7 mm, consistent with the earlier ensemble. However, notable revisions are observed in the 95% confidence interval, which has been adjusted upwards from [1.5-7.2] to [1.6-7.4] for both weighting. Similar revisions can be observed for the 25th and 75th percentiles. For the Full-period weighting, it ranges from 2 to 4.3 mm, while for the sub-period weighting it ranges from 2.1 to 4.4 mm. In contrast, the prior ensemble had a [25th,75th] percentile range of 2.0 to 4.1. These distribution adjustments indicate a slight expansion of the confidence intervals, with increased consideration given to higher values. In the year 2050, both the Full-period weighting and Sub-period weighting methods result in an increase in the median estimate from 0.90 mm to 0.94 and 0.95 mm. The 95% confidence interval shows





**Figure 6.** Evolution of Upernavik Isstrøm ice mass loss over the period 1985-2100 for the Hpr (grey), the Hpo (dark green), the Ppr (orange), and the Ppo (light green) prior for different weightings (Full-period weighting on top-left, $f_{param}$ weighting on top-right, Sub-period weighting on bottom-left, and SSP weighting on bottom-right). Each ensemble's mean is represented by a solid line, and the shaded area encompasses 95% of ensemble members. Observations from the 1985-2019 period are indicated by the symbol +. The red box highlights a zoomed view of the period 1985-2050.





a slight reduction, shifting from [0.52, 1.90] to [0.53, 1.86] and [0.57, 1.87]. Furthermore, the 25th and 75th percentiles shift from [0.70, 1.1] to [0.68, 1.13] and [0.70, 1.16].

In contrast, when using $f_{param}$ weighting for weighting, significant changes are observed in the prediction of the posterior
ensemble, leading to a larger projected loss of mass. The median SLR contribution in 2100 increases to 3.0 mm compared to 2.7 mm in the prior ensemble. Moreover, the 95% confidence interval expands significantly from [1.5-7.2] to [1.7-9.1], and the 50% interval widens from [2.0, 4.1] to [2.2, 4.4]. In the year 2050, the median contribution is also revised upwards from 0.90 mm to 1.02 mm. Similar patterns are observed for the 95% confidence interval, which shifts from [0.52, 1.90] to [0.60, 2.05], and the 50% one, which expands from [0.70, 1.1] to [0.77, 1.19].

### 3.4.2   SSP and combination weightings

The SSP weighting on the future prediction has a very significant effect, reducing the median contribution of UI in 2100 from 2.7 mm to 2.2 mm. The 95% confidence interval is also revised downwards, from [1.5-7.2] to [1.5-5.7], as the 50% confidence interval, from [2.0,4.1] to [1.9,3.0]. In 2050, on the other hand, the range of the 95 % confidence interval becomes wider, going from [0.52-1.90] to [0.46-2.05], while the median is revised slightly downwards, from 0.90 to 0.79 mm.

Through the combination of ISM weighting, specifically Sub-period weighting, with the existing SSP weighting (Figure 7), we are able to constrain the wider 95% interval in short-term predictions (2050). This combined approach results in a reduced interval of [0.53, 1.89], compared to the [0.46, 2.05] interval achieved by SSP weighting alone, and is slightly smaller than the prior interval of [0.52, 1.90]. Moreover, the combination leads to a slight upward shift in the median from 0.79 mm to 0.80 mm.

In the context of long-term predictions (2100), the combination with Sub-period weighting also results in an upward shift compared to SSP weighting alone. The 95% confidence interval shifts from [1.5, 5.7] to [1.6, 5.5], while the median experiences a slight increase from 2.18 mm to 2.25 mm.

By employing this weighting combination, we are able to capitalize on the long-term reduction achieved by SSP weighting, while simultaneously leveraging the uncertainty reduction facilitated by dynamic performance-based weighting in the short
and medium term. Notably, dynamic performance-based weighting also contributes to the reduction of long-term uncertainty by excluding members that underestimate past mass loss and provide the lowest SLR contributions.

## 4   Discussion

### 4.1   Prior ensemble

The control run in the ISMIP context aims to address models' limitations in accurately replicating recent observed ice sheet
changes due to artificial model drift, facilitating the evaluation of each projection's deviation from this drift (Goelzer et al., 2020; Fettweis et al., 2020; Nowicki et al., 2020). However, the control run represents the average state of the recent period, accounting for both model drift and climate change already experienced, such as a 0.5°C warming in 1990 compared to pre-

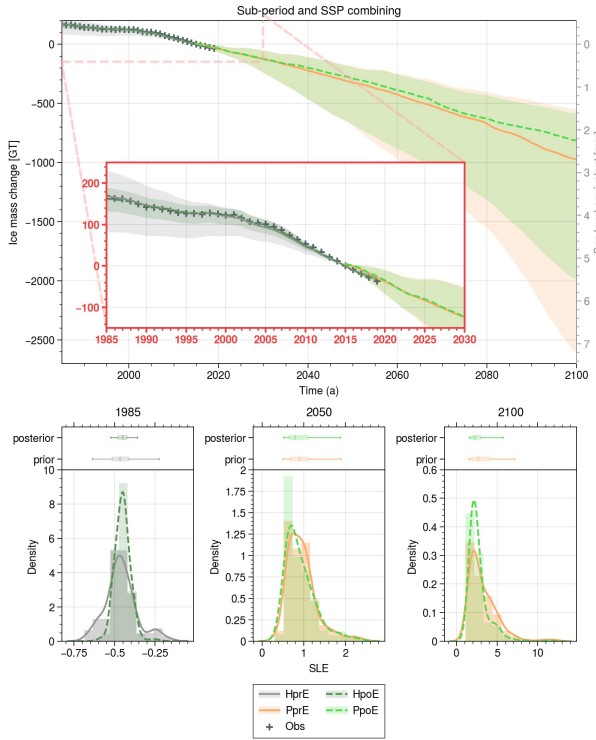

**Figure 7.** Evolution of UI ice mass loss over the period 1985-2100 for the historical (grey), the predicted (orange) and the committed mass loss (blue) prior and posterior ensemble simulations (combined Sub-period and SSP weightings, achieved by multiplying the weights of these two weightings). For each ensemble, the mean is represented in solid line and the shading include 95% of the ensemble members. Observation from Mouginot and others (2019) of the 1985-2019 period are represented by +. The red box shows a zoom to 1985-2050 period.

industrial conditions (Masson-Delmotte et al., 2021). Consequently, the results obtained by differentiating between a simulation with realistic forcing and a control simulation with constant forcing do not allow us to predict the future evolution of Sea Level

Rise (SLR), as they do not take into account the mass loss already underway as a result of past global warming.

The initialisation method developed in Jager et al. (2024) effectively reproduces the past UI trend, negating the need for control run differentiation as practiced in the ISMIP6 framework. In our case, since we can successfully reproduce recent observations, the prediction (Ppr) offers a comprehensive SLR prediction encompassing this committed mass loss. Subtracting the control (Cpr) from the prediction from the prior ensemble (Ppr) would have underestimated UI's contribution to SLR by

approximately 1 mm, almost 35% of the median value of Ppr. This implies also that stabilizing the forcing at present levels does not stabilize the ice sheet, which would continue to melt. However, it's important to note that in this study, we do not employ a constant forcing from the present day in the control experiment or "Cpr". Instead, we utilize a prescribed Surface Mass Balance (SMB) representative of the period between 1960 and 1990, along with a prescribed front characteristic of the year 2015. It's worth mentioning that using an SMB averaged over more recent years, such as those from the 2010 decade, would yield a





higher estimate of melt for Cpr and consequently result in a even greater committed mass loss. Additionally, considering the current climate conditions, it is more likely for the front to retreat than to advance, leading to increased discharge and more mass loss.

## 4.2 Uncertainty of future prediction

Our sensitivity study on the contribution of sea-level rise differs from previous studies (Aschwanden et al., 2019; Goelzer et al., 560 2020; Hill et al., 2021) by incorporating the SSP into the parameters, rather than conducting separate analyses for each SSP. However, this approach is not unique and is similar to studies carried out for glaciers outside the Greenland and Antarctic ice sheets (Marzeion et al., 2020), as well as for global temperature and precipitation (Hawkins and Sutton, 2009). As the SSP is included in our sensitivity analysis, in contrast to previous studies on the Greenland ice sheet up to at least 2100, such as Goelzer et al. (2020) and Aschwanden et al. (2019), we have had to reassess the estimates of uncertainty associated with the 565 SSP, the ISM, the RCM, and the AOGCM used in those studies. Herein, we delineate our approach to this reassessment process and demonstrate that these revised estimates yield results consistent with those obtained in our study.

Specifically, the ISMIP6 study highlights a substantial difference of approximately 58 mm of sea-level equivalent between the means of the RCP8.5 and RCP2.6 scenarios, with the combined uncertainty spanning 125 mm of sea-level equivalent, representing a variability of over 45%. Similarly, the variance between RCPs in Aschwanden et al. (2019) accounts for nearly 570 35% of the overall uncertainty. In our study, which focuses on a single tidewater glacier in Greenland, we observe a comparable magnitude of uncertainty (40%). Notably, when examining the ISM, the uncertainty attributed to this factor is considerably lower in our study (15%) compared to ISMIP6 (35%). Conversely, the uncertainty associated with front parameterisation is higher in our study (20%) than in ISMIP6 (15%), because we are looking at uncertainty due to parametric differences in one model compared to different models in ISMIP6. The uncertainty of the RCM is not investigated in ISMIP6, preventing direct 575 comparisons. Furthermore, our study reveals zero sensitivity of the AOGCM, in contrast to ISMIP6 where it accounts for almost 30% of the overall variability (36 mm of the 125 mm total uncertainty). This lack of influence can be attributed to a compensatory effect between ice discharge and surface mass balance, as explain with the figure 4. Finally, Rohmer et al. (2022) demonstrated that spatial resolution and front parameterisation were the two most influential parameters in the ISMIP6 framework, which notably does not account for RCM and SSP uncertainties. In our analysis, we have not factored in the 580 uncertainty associated with spatial resolution within the global sensitivity assessment. However, we conducted mesh sensitivity tests during the historical period, increasing the resolution by factors of 2 and 4. These tests revealed that such modifications resulted in approximately a 30% alteration in local velocities at a given time, primarily driven by the high sensitivity to the front's resolution. Nevertheless, the overall mass loss across these varying meshes exhibited minimal variation, amounting to less than 5%. Indeed, at the front, one element more or less strongly changes the velocity field, but this is much less noticeable 585 when we look at the total mass loss, which changes by 5 % between the simulations for the historical period. Regarding the significance of front retreat parameterisation, as highlighted by Rohmer et al. (2022), our study aligns with this observation, demonstrating its substantial influence. In fact, it emerges as the second most influential factor after SSP, which was not explored in the mentioned study.



Considering the constrained sensitivity of our study to the ISM, our calibration analysis reaffirms the limitation of reducing uncertainty solely through ISM calibration. Notably, our prior results indicate a considerable reduction in uncertainty compared to broader intercomparison studies like ISMIP6 when employing a single model. This discrepancy can be partly attributed to ISMIP6's more comprehensive consideration of structural uncertainties within the models. Additionally, the prior ensemble already demonstrates a high level of skill in reproducing past observations, as shown in Figure 2. However, it is important to acknowledge that all models must overcome this challenge before a future intercomparison study like ISMIP7, as suggested in Aschwanden et al. (2021). Once this hurdle has been successfully addressed by the dynamics modeling community, it will be essential to focus on reducing other sources of uncertainty. This task can be pursued concurrently with the initial task of improving dynamics modeling.

Our findings regarding the weighting of SSPs underscore the significant reduction in uncertainty, particularly for long-term predictions, that can be achieved through this approach. This opens up possibilities for similar studies aimed at assigning weights to other model assumptions, such as front parameterisation, the selection of RCMs, or AOGCMs. For short-term predictions, which are of great interest to some practitioners, it appears that the primary sources of uncertainty are associated with RCMs, AOGCMs, and front parameterisation. While there remains a substantial level of uncertainty associated with ice sheet dynamics, it appears that we are nearing the practical threshold for achievable reductions while maintaining the robustness of our results concerning these dynamic processes.

In the realm of RCMs, multiple studies have been conducted to compare these models with data obtained from the Greenland ice sheet (Fettweis et al., 2020; Vernon et al., 2013). These comparative analyses serve to identify the biases inherent in different RCMs and guide efforts towards their correction in subsequent iterations. However, despite these endeavors, the various models continue to yield significantly divergent results, attributable in part to disparities in the underlying physics employed and the down-scaling techniques used. For example, in our previous study (Jager et al., 2024), we demonstrated that members using RACMO, which employs a 1 km statistical down-scaling approach to the 5.5 km grid (Noël et al., 2016), better reproduce past trends in surface elevation than the MAR model without statistical down-scaling (Fettweis et al., 2017). One potential solution to address the disparities among RCMs is to incorporate multiple ensemble members from these models, accounting for their associated uncertainties. This presupposes that the RCMs themselves undertake uncertainty quantification to follow the bayesian approach proposed in Aschwanden et al. (2021). By doing so, it becomes possible to generate a range of forcing scenarios for both historical and future periods and evaluate their performance against past surface elevation observations. Efforts can also be directed towards reducing uncertainty by promoting convergence among different RCMs, contingent upon them duly accounting for their intrinsic uncertainties.

In the context of front retreat parameterisation, despite its foundation in observational data, there remains significant room for reducing associated uncertainties. In our study, we have considered two sectors, the CW and NW, at three distinct sensitivity levels (low, medium, and high) to prevent unwarranted confidence in our findings. We compared the observed and parameterised retreat in these sectors over the historical period, with the CW sector demonstrating higher sensitivity, as indicated by its greater $\kappa$ value for a given sensitivity level. Consequently, the high-level of sensitivity retreat in the CW sector resulted in a 6 km recession between 1985 and 2015, exceeding that observed in the UI-N by 0.3 km (Figure 1). In contrast, the UI-S



branch exhibited the smallest retreat at 1.1 km, while the low-level of sensitivity from NW sector exhibited a retreat of 0.9 km. To improve the parameterisation, it will be necessary to take into account additional factors beyond currently considered runoff and far-field ocean temperature changes, which do not allow for the difference in ice dynamics between the different ice streams as shown here for Upernavik Isstrøm. Furthermore, given the significant influence of this front parameterisation, as revealed by the sensitivity analysis, it seems important for the scientific community to engage in further research aimed at improving this characterization of front retreat and introducing a more physics-based formulation of this parameterisation. Such efforts would require a comprehensive analysis of historical behavior, along the lines of previous studies (Wood et al., 2021), followed by calibration efforts for an appropriate calving law (Bondzio et al., 2018), and investigations into the complex interactions between ocean, atmosphere and outlet glaciers (Slater et al., 2019).

Considering the substantial impact of the SSP on uncertainty, it would be valuable to conduct a more thorough examination of this uncertainty, particularly given that SSP2-4.5 and SSP1-2.6 exhibit similar outcomes in this study. Currently, many studies primarily focus on the SSP5-8.5 scenario, which yields striking results due to its high level of warming (Hausfather and Peters, 2020), and only a few include SSP1-2.6 or SSP2-4.5, as used in this study. However, considering the notable differences in results between SSP2-4.5 and SSP5-8.5, a more refined discretisation of future scenarios would provide a more comprehensive understanding of uncertainty in future sea-level rise projections. This will also help macro-studies such as McKay et al. (2022) to better identify at what level of warming the GrIS and AIS tipping points may be exceeded.

## 4.3 Cross-validation method for bayesian calibration

To address the challenge of spatial and temporal correlation and its impact on model weighting, various approaches have been previously explored. We discuss three methods here. The first approach involves the utilisation of aggregated data, such as volume and discharge changes, as it was used with a single global value in Ritz et al. (2015) for calibrating the future of Antarctic Ice sheet with the mean rate of change for each sub-basin. However, by using time-series, it does not effectively resolve the issue of temporal correlation as used in Aschwanden and Brinkerhoff (2022) with the mass calibration. The second approach employs a performance metric, which can be interpreted as the distance between the observed and modeled fields, effectively treating multiple observations as a single observation (i.e., $n_{obs}$ is then equal to one in Eq. 13). This is the method used in our study. For example, in a different context focused on constraining a calving law, Bondzio et al. (2018) proposed an approach to weight ensemble members using a metric that measures the distance between each member's front and the observed one. In other contexts of Antarctic Ice Sheet modeling, Pollard et al. (2016) and Albrecht et al. (2020) also proposed weighting methods based on a metric that measures the performance of each member. A third option is to use one observation for each mode (distinct group of ensemble members with similar characteristics) of the ensemble using principal component decomposition. In the domain of glaciology, Wernecke et al. (2020) implemented the third approach by employing it in the context of the Amundsen Sea bay. The calibration process utilised two-dimensional satellite data reflecting surface elevation change. Notably, this investigation conducted a comparative analysis, by comparing this mode-based approach with both the first method (i.e., aggregated data approach) and an approach that kept all the information encapsulated in the field of view (Eq. 13). The study's results indicated that this mode-based approach succeeded in reducing uncertainty to a greater extent compared



to the approach utilising aggregated data but not as effectively as an approach that harnessed the complete observation field. The study postulated that this second aspect could signify potential overconfidence in the retrieved parameter values or, conversely, a

more efficient exploitation of the available information. However, it's worth noting that the study did not perform an evaluation of the calibrated ensemble's performance, leaving a distinction between these two possibilities uncharted.

Our validation method responds to the limitations raised above and represents a significant advance in Bayesian calibration within model ensembles of ice sheet modelling. We used a cross-validation approach that allows us to examine the diverse impacts of weighting choices and mitigate the risk of overfitting. However, it is important to acknowledge that the selection

of hyperparameters (e.g., number of parameters taking into account for the ISM sensitivity analysis) itself may contribute to overfitting, and we have yet to identify an effective strategy to address this challenge.

While there is room for further improvement in our method, the unique characteristics of glaciology pose challenges in drawing inspiration from other scientific disciplines. In contrast to hydrology, meteorology, or oceanography, where a wealth of events can be used for weighting or calibration, glaciology often deals with a limited number of observed events. For in-

stance, in hydrology, multiple flood events can be employed for weighting and calibration, with additional events available for validation (Hallouin et al., 2020). In contrast, glaciology typically involves only a single observed retreat event per catchment, as demonstrated in our study of UI. Consequently, the application of such techniques becomes unfeasible in glaciology. Nonetheless, the notion of calibrating and validating parameters on a basin-specific basis holds great promise, as it would enable a more targeted parameter selection within individual basins rather than considering the entire ice sheet as a whole.

To effectively validate the calibration of parameters on a per-basin basis, it is imperative to identify a glacier exhibiting dual events (e.g., two major retreats of the same front since the 1980s). Subsequently, the model ensemble can be calibrated using the initial retreat data, followed by a comparison of the calibrated model's CRPS performance against that of the non-calibrated model. Such a case study could also serve as a basis for comparing the calibration with weights as developed here, against other transient data assimilation methods as developed in Goldberg et al. (2015).

Furthermore, the validation approach employed in this study has demonstrated the additional benefits of transient calibration compared to snapshot inversion, i.e. the traditional inverse method of friction calibration. It is particularly in scenarios where a front retreat occurs, as evident in the case of the substantial retreat of the UI-N and UI-C fronts. In contrast, when there are no significant front retreats (UI-S) or velocities are low, implying a limited role for dynamics (UI-SS), calibration does not seem to offer any discernible improvements. This observation is likely attributable to the fact that, in the absence of substantial

changes in dynamics, all ensemble members can effectively reproduce these dynamics through inversion alone.

Our study underscores the substantial impact of calibrating with velocity and elevation data (Full-period weighting) in diminishing the uncertainty linked to historical ice mass loss in the UI region, especially when considering their temporal aspects (sub-period weighting). This reduction in uncertainty opens up possibilities for data assimilation of past velocity and elevation data inspired of Goldberg et al. (2015) or Gillet-Chaulet (2020), offering a way to reconstruct discharge with better-

characterised uncertainties compared to the conventional input-output method. Using advanced transient data assimilation techniques can lead to enhanced performance in terms of cumulative ice discharge, moving beyond the limitations of the simplistic gate-based approach. By incorporating velocity and elevation data through data assimilation, uncertainties related



to velocities, surface elevation, and bed elevation can be effectively addressed, making the use of gates unnecessary. This approach represents a promising advancement in improving the accuracy and reliability of ice discharge reconstructions.

## 4.4 Insights for future studies

Thanks to our validation methodology, designed to mitigate the risk of overfitting, we can assert the reliability of our findings concerning future sea-level rise and the interpretation of related outcomes. In this context, we offer valuable insights that hold significance for ice sheet modelers concerned with bayesian calibration through the weighting choices and retrospective modeling.

Our investigation reveals that the selection of weighting strategies for bayesian calibration, encompassing the probability distribution shape, the determination of an appropriate standard deviation ($\sigma$), and the incorporation of multiple periods, can precipitate over-adjustment during the calibration process. Opting for an overly narrow distribution or favoring a Gaussian distribution over a Student distribution can result in overfitting, wherein only a few high-performing members are emphasized, thereby disregarding crucial information. This observation resonates with findings presented by Jiang and Forssén (2022), wherein they highlight significant challenges arising from the utilization of complex likelihood functions (as encountered in our study with multiple periods) or extremely small data errors (represented by excessively small $\sigma$ values). Such circumstances may lead to the selection of a limited number of members, potentially overlooking vital aspects of the posterior distribution. Consequently, cautious consideration is warranted when employing these techniques, and our validation approach serves as a safeguard against such pitfalls.

Regarding the selection of data, our analysis revealed a notable asymmetry between spatialized data, such as speeds and elevations, and global data, such as ice discharge and total mass loss. Specifically, when members were chosen based on their ability to accurately reproduce velocities and elevations, it resulted in an overall enhancement in the ensemble's performance for both spatialized data and global indicators like ice discharge and mass loss. However, the converse was not observed to hold true. This discrepancy can be attributed to potential compensatory effects, wherein a model that closely matches the observed discharge may exhibit excessively high velocities and disproportionately low elevations. Conversely, a model that accurately represents velocities and elevations will inherently yield a satisfactory discharge estimation. Additionally, selecting members based on spatialized data also facilitates improved reproduction of other phenomena that indirectly influence discharge, such as shear margins.

We conducted an analysis of the influence of sub-periods that characterize different phases, namely before, during, and after glacier retreat. Our findings indicate that the use of sub-periods results in a slightly improved selection of members using parameterisation compared to Full-period weighting, as the former exhibits better representation of glacier acceleration. This approach can be considered as an intermediate method between $f_{param}$ weighting and Full-period weighting, particularly during the historical period. However, it should be noted that the selection process is still strongly influenced by the choice of inversion data, which becomes less influential in future predictions. A potential future approach could involve using only pre-retreat data to create a new ensemble, enabling a slightly more refined selection based on other model parameters. This calibration method also provides more relevant information, as it demonstrates the model's ability to accurately reproduce





past total mass change data, which aligns with our ultimate objective. Consequently, we place slightly more confidence in Sub-period weighting than in Full-period weighting.

The most influential parameter within the ISM, regarding the reproduction of past Upernavik Isstrøm behavior, is the choice

of input data for the inverse method of the friction field. Weights are notably larger when using inversion data from the pre-retreat period (1990s and 2000s). This is because no extrapolation is needed in the ice-free areas during this period, in contrast to post-retreat inversions where extrapolation is necessary due to ice front retreat, introducing additional uncertainties in their performance assessment. Consequently, post-retreat inversions exhibit lower performance in ice-free areas before the retreat. The accurate reproduction of data when the front retreats instills greater confidence in our predictions, aligning with

the observed trend of front retreat. However, if one wishes to delve further back in time or undertake a paleo-climatic study, extrapolation becomes necessary (e.g., Haubner et al., 2018). Thus, the choice of friction field extrapolation will become a crucial issue as it significantly influences the result, as previously shown in Jager et al. (2024).

Regarding the weighting approach that assigns higher weights to ensemble members utilizing a friction law that accounts for the sub-hydrology effect in a parameterised manner, our findings indicated a somewhat reduced overall performance compared

to the Full-period weighting approach. However, it is noteworthy that the outcomes obtained through this approach align with the expectations outlined in Jager et al. (2024). This prior study suggested that our parameterisation would likely lead to increased mass loss. Moreover, the findings related to parameter selection indicate that the role of parameterisation is more crucial than the specific choice of friction law formulation when it comes to reproducing the observed data. In Joughin et al. (2019), reducing friction near the front probably contributed as much, if not more, to obtaining better agreement with observed

data than using a regularized Coulomb law. This suggests that a Budd law, whose formulation is close to that used here for members using Weertman's law and which takes subglacial hydrology into account in a parameterised way, would perform just as well as the regularized Coulomb law. For Greenland tidewater glaciers, Choi et al. (2022) also showed that friction laws that include a parameterised dependence on the effective pressure better reproduce the observed acceleration and mass loss of the past decade in Northwest Greenland. However, despite the promising results from our previous paper, the predominance of

inversion data had a moderating effect on the extent of the observed improvements. In the previous study, only front-end post-processing data were employed for the inversion process. In contrast, the current study incorporates data from both the pre- and post-retreat periods, which noticeably influenced the calibration due to the necessity for extrapolation. In the future, as the front continues to retreat, it is anticipated that this influence will diminish, rendering extrapolation unnecessary in ice-covered regions.

**5 Conclusions**

In conclusion, we have shown than our initialisation method for Elmer/Ice effectively captures trends of ice mass change and enhances the credibility of future tidewater glacier contributions to sea-level rise, aligning with recommendations from Aschwanden et al. (2021). This approach not only characterizes model uncertainties but also reproduces past observations, akin to successful efforts with ISMs such as ISSM and PISM for Greenland (Aschwanden and Brinkerhoff, 2022; Nias et al., 2023).



By addressing model drift, our study moves beyond conventional projections and sensitivity analyses (Goelzer et al., 2020; Seroussi et al., 2020), signaling a paradigm shift towards more localised and precise sea-level rise predictions, particularly for polar ice sheets.

Our sensitivity analysis emphasizes that, in 2100, the most significant factors affecting the future contribution of Upernavik Isstrøm to sea-level rise are the shared socio-economic pathways (SSP), followed by the front retreat parameterisation. Regional

climate models (RCM) and ISM have a slightly lesser impact on sea level rise contribution at long term, while atmospheric ocean general circulation models (AOGCM) play a minor role. However, in the short and medium term, for results that may be of interest to public policy, the influence of the SSP is much less, with uncertainties coming mainly from the other 4 sources.

Furthermore, our ISM calibration with different weightings brings about marginal improvements in 2100 due to its relative low impact on the ice mass loss sensitivity. However, the combination of multiple weightings shows promise, suggesting

that a more holistic approach may yield greater benefits. In addition, our methodology combining bayesian calibration and cross-validation has generated noteworthy findings of relevance to the scientific community: (i) Spatially-based weighting demonstrates enhanced robustness compared to globally-based weighting strategies; (ii) Temporal partitioning of the calibration period, particularly considering calving events (prior, during, and post-calving), significantly reduces overall uncertainty while preserving comparable model performance; (iii) The model initialisation using inverse methods exhibits robustness,

particularly in scenarios involving glacier front retreat, with friction initialisations derived from pre-retreat data yielding superior performance. These insights contribute to advancing our understanding of ice sheet modeling and calibration techniques, offering avenues for further research and improvement in future studies.

Looking ahead, it would be interesting to extend our methodology to the scale of the Greenland ice sheet. This would involve creating frontal masks dating back to the 1980s, collecting velocity and elevation data over this historical period for the

peripheral regions of the ice sheet, and running ensemble simulations for comprehensive comparisons. Such an undertaking could lead to a better understanding of ice sheet dynamics and improved forecasting capabilities.

*Code and data availability.*  jupyter-notebook, conda environnement and data to plot figures of this article are available here: https://doi.org/10.5281/zenodo.10794469

## Appendix A: Model description

The ISM employed in this study is the parallel finite-element code Elmer/Ice (Gagliardini et al., 2013). The model domain corresponds to the UI catchment, as depicted in Fig. 1. The model used here follows the methodology presented in Jager et al. (2024), and we provide a concise overview of its main aspects in this section. For a more comprehensive understanding, we refer readers to the original paper.

The Shelfy-Stream Approximation (SSA, MacAyeal (1989)) is used for the force balance equations together with the non-

linear Glen's flow law (Glen and Perutz, 1955) for the constitutive relation. It relies on three parameters: the Glen exponent



$n$, the rate factor $A$, and the enhancement factor $E$. Thermo-mechanical coupling is disregarded due to the short time period considered (Seroussi et al., 2013), and for simplicity, the rate factor $A$ is assumed to be constant over time. The initialisation of $A$ involves using a present-day 3D ice temperature field computed with SICOPOLIS (Greve, 1997), which is preceded by a paleo-climatic spin-up and incorporates the prefactors and activation energies provided by Cuffey and Paterson (2010).

Uncertainties related to this flow law are commonly accounted for through the enhancement factor $E$, which serves as a scaling factor to $A$.

In this study, two distinct friction laws governing the relationship between basal velocity $\boldsymbol{u_b}$ and basal shear stress $\boldsymbol{\tau_b}$ are employed for grounded areas:

– A Weertman friction law (Weertman, 1957):

$$\boldsymbol{\tau_b} = -\beta_W ||\boldsymbol{u_b}||^{\frac{1}{m}} \frac{\boldsymbol{u_b}}{||\boldsymbol{u_b}||} \tag{A1}$$

– A regularized-Coulomb friction law (Joughin et al., 2019):

$$\boldsymbol{\tau_b} = -\beta_{RC} \left( \frac{||\boldsymbol{u_b}||}{||\boldsymbol{u_b}|| + u_0} \right)^{\frac{1}{m}} \frac{\boldsymbol{u_b}}{||\boldsymbol{u}||} \tag{A2}$$

Both equations (A1) and (A2) involve a friction parameter ($\beta_W$ or $\beta_{RC}$ respectively), a positive exponent $m$, and a threshold velocity $u_0$ in the case of the regularized-Coulomb friction law (Joughin et al., 2019). The friction parameter $\beta$ can either

remain constant over time or take into account the effective pressure in a parameterised way (Jager et al., 2024):

$$\beta = \beta_{ref} + \beta_{lim} \frac{d}{d + d_{lim}} \tag{A3}$$

where $\beta_{ref}$ represents a time-independent reference field, $d$ denotes the distance to the front, and $\beta_{lim}$ and $d_{lim}$ are two parameters accounting for the dependence of $\beta$ on this distance.

Significant uncertainties surround the parameter $\beta$, often initialised based on current topography and surface velocity obser-

vations using an inverse approach that minimizes a composite cost function. This cost function comprises terms assessing the discrepancy between observed and modeled velocities, a regularisation term promoting a smooth friction field solution, and a third term that penalises flux divergence anomalies (Gillet-Chaulet et al., 2012). This two last terms are weighted with the parameters $\lambda_{reg}$ and $\lambda_{div}$ that are adjusted using a L-curve approach (Gillet-Chaulet et al., 2012). The inversions in this study are conducted at the UI scale, distinguishing it from our previous study that used inversions previously made at the GrIS scale

(Gillet-Chaulet et al., 2012).

For the evolution of the bottom and top free surfaces, we solve the continuity equation for the ice thickness using the flotation condition. As we do not resolve the thermo-mechanical coupling, we neglect the basal melt rate in grounded areas. We also set it to 0 in floating areas, as they remain small during our simulations. For the surface mass balance $\dot{a}_s$, we use outputs from a RCM.





The unstructured mesh is refined near the ice front and in areas where high velocity or thickness curvatures are observed, featuring element sizes ranging from 150 m to 600 m within the initial 50 km and increasing to around 5 km further upstream. A time step of 5 days is used.

The temporal variation of the glacier fronts is treated as an external forcing, with their positions considered fixed within each time step. The mesh remains unchanged, and the effective ice-ocean boundary is defined by the edges connecting glaciated and
deglaciated elements, resulting in discrete changes over time. Deglaciated elements are subsequently deactivated and excluded from the numerical solution.

**Appendix B:  ISM parameters**

**B1    Factor fixing and factor mapping**

As detailed in the primary manuscript, modifications have been made to the dynamic parameters in comparison to Jager et al.
830   (2024):

- the ice rheology : Considering the enhancement factor ($E$), we ascertained that members with $E$ between 1 and 3.5 yield better results. Consequently, we transformed the distribution from continuous between 0.5 and 5 to a log-normal distribution with parameters $\mu = 0.8$ and $\sigma = 0.5$ to enhance the values within the range [1, 3.5].

- the friction law : To distinguish between the influence of the choice of friction law and the presence of the parame-
terisation described in Jager et al. (2024), we introduce two new parameters. The first parameter, denoted as $f_{law}$, is characterized by two states: "RC" when the member employs the regularized Coulomb friction law and "W" when utilizing the Weertman friction law. The second parameter, termed as $f_{param}$, possesses binary values: "true" to signify the friction parameter $\beta_{RC}$ or $\beta_W$ evolving according to Eq. A3, and "false" when the friction parameter remains constant. We also keep in our parameter space the choice of exponent of the friction law $m$. Finally, the impact of $u_0$ on model
outputs was found to be less significant than that of the other ISM parameters. We therefore use a single $u_0$ of 300 m.a$^{-1}$.

- the calibration of the friction field : For the friction field, we do not take into account the uncertainty of the whole field (i.e., one parameter per mesh node) but we consider only the uncertainty of the hyper-parameters of the inverse method, which considerably reduce the parameter space. Our two hyper-parameters are the regularisation weight $\lambda_{reg}$, and the observed data used for the inversion $OBS_{inv}$. In the previous study, the impact of $\lambda_{div}$ on friction was found to be
less significant than that of the other ISM parameters and we therefore use a single $\lambda_{div}$. In terms of the overarching framework, we have implemented modifications to our inversion procedure. In our previous study, a 40-member inversion had previously been carried out at the scale of Greenland. In contrast, the present study involves individual inversions for each member at Upernavik Isstrøm scale, affording enhanced continuity within the parameter space of inversion. A L-curve analysis was conducted to determine the revised distribution profile of $\lambda_{reg}$, alongside the optimal value for $\lambda_{div}$.
While our previous study used five observational velocity datasets from the 2010s, aligned with BedMachine surface



elevations, the current approach employs average velocities and altitudes representative of the entire temporal span from 1985 to 2020. These averages were computed using our own dataset, spanning distinct periods for $OBS_{inv}$: 1985-1995, 1995-2005, 2005-2015, and 2015-2020.

- the initial geometry : With regard to the initial surface elevation, we have established that it exerts a minimal influence
on both ice mass loss and ultimate volume. Compared with this previous study, we have therefore fixed the parameters influencing only this initial surface, namely the period for relaxation, which we set at 5 years, and the period over which the SMB was averaged, which is now the average over the period 1960-1990. By doing so, the initial surface depends only on the ISM parameters (material property, friction law and friction field calibration).

## B2 Sensitivity analysis

Figure B1 illustrates the evolution of sensitivity indices for ISM parameters from 2015 to 2100, computed using the Predicted prior Ensemble (Ppr), concerning ice mass, ice mass change, cumulative SMB, and cumulative ice discharge. The sensitivity indices provided in the figure are presented in their non-normalized form.

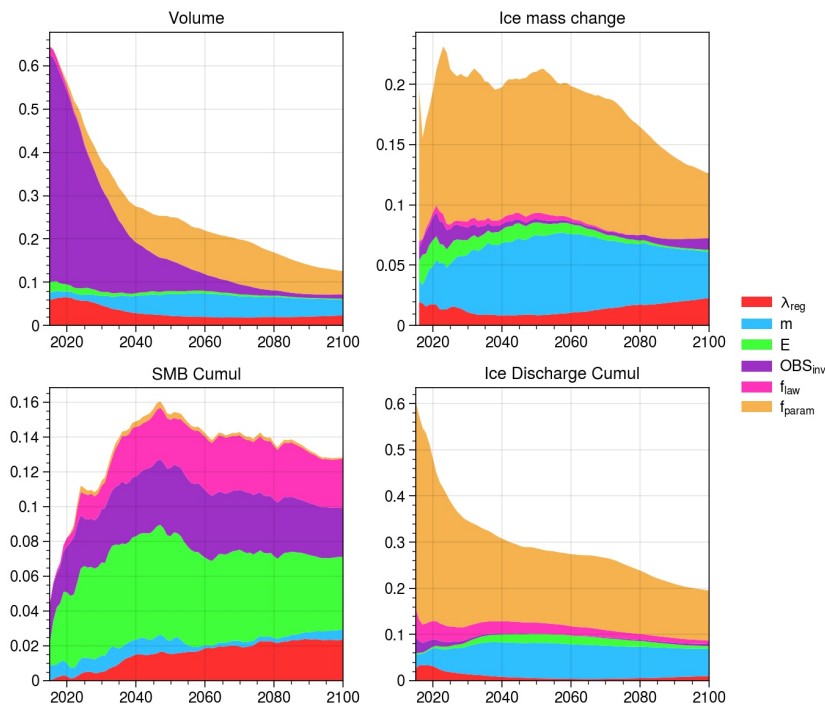

**Figure B1.** Sensitivity indices for the six ISM parameters of Ppr ($\lambda_{reg}$, m, E, $OBS_{inv}$, $f_{law}$, $f_{param}$) for the volume (a), the ice mass change relative to 2015 (b), the cumulative SMB since 2015 (c) and the cumulative ice discharge since 2015 (d).





## B3 Weighted parameters

The Full-period weighting method used shows a clear preference for members for whom friction initialisation was done with
data from the 1990s and 2000s, as shown by the weights assigned (Fig. 5.b). This preference is attributed to the comparatively
lower RMSE values observed for these members. Conversely, members with inversions conducted in 2010 or 2017 necessitate
extrapolation in regions where the ice front has retreated, thereby introducing additional uncertainties in their performance
assessment. Consequently, these later inversions exhibit poorer performance in ice-free areas before the retreat due to the need
for extrapolation. Notably, the influence of the inversion year diminishes considerably after the retreat of the ice front. The fact
that we faithfully reproduce the data when the front retreats gives greater confidence in the results of our predictions, as the
front tends to retreat.

The presence or absence of the $f_{param}$ emerges as the second prominent factor influencing the distribution shift between the
prior and posterior in the Full-period weighting process, subsequent to $OBS_{inv}$ (Fig. 5.c). This finding aligns with the findings
of Jager et al. (2024), where we demonstrated the crucial role of this parameterisation in enabling the model to accurately
reproduce data spanning the period from 1985 to 2020.

Regarding the regularisation weight of the cost function, $\lambda_{reg}$, the distribution tends to shift towards lower values (Fig.
5.a). This implies that excessively high $\lambda_{reg}$ values result in elevated RMSEs due to an overly smooth friction field. This
result is noteworthy, as it suggests that solutions with less smoothness, potentially influenced by data noise, are not necessarily
of inferior quality. This scenario is preferred over excessive regularization, highlighting the importance of striking a balance
between regularization strength and model fidelity.

Regarding the parameters $m$, $law$, and $E$, there are no substantial trends in the difference between the prior and posterior
distributions (see Figure 5 d, e and f). However, for the parameter $m$, members with values exceeding 0.4 or approximately
0.25 exhibit higher weights, although this outcome could be attributed largely to the influence of $\lambda_{reg}$, $OBS_{inv}$, and $f_{param}$.
Likewise, members characterized by an $E$ value near 1.9 and $law = W$ demonstrate increased weights. In hindsight, our initial
choice of distribution for these three parameters proves to be suitable due to the absence of significant changes observed in
their posterior distributions.

In terms of Sub-period weighting (see Figure 5), the notable alteration is the amplification of the discrepancy in $f_{param}$
selection. Specifically, the posterior probability rises from 6.4 to 7.4, indicating a greater likelihood for parameterised members
to accurately replicate distinct periods. These findings provide further validation and support for the outcomes reported in Jager
et al. (2024).

## Appendix C: Cross-validation of bayesian calibration choices

Figure C1 visually presents the normalized Continuous Rank Probability Score (CRPS) as a performance metric for distinct
calibrated ensemble configurations. These configurations encompass Full-period weighting, Sub-period weighting, and $f_{param}$
weighting approaches. The CRPS are computed for ice discharge, cumulative ice discharge, surface elevation, and surface
velocity, providing a comprehensive view of the calibrated ensembles' performance across various datasets. In addition, it is




important to note that similar figures illustrating the Mean Absolute Error of the ensemble mean (MAE, Eq. 3), the standard deviation of the ensemble (STD, Eq. 4) and the non-normalized values are not included to avoid duplication (see supplementary material, figures S1 to S5). Moving beyond the graphical presentation, this section undertakes a thorough analysis of the calibrated ensemble's performance in relation to Hpr under different weightings. We begin by scrutinizing the full-period weighting approach, delving into how variations in probability density form, estimate used for $\sigma$, and data choice for the calibration. Subsequently, we evaluate the outcomes of sub-period weighting and $f_{param}$ weighting methodologies.

In general, all the various calibrations result in notable improvements in the CRPS of the ensemble. This improvement is evident by the prevalence of blue shades (67%), indicating lower CRPS values, compared to red shades (33%), indicating higher CRPS values, and the presence of more dark blue shades than dark red shades. For all combinations, the STD is reduced by 20 [4,56]% for cumulative ice discharge, 18 [5,47]% for ice discharge, 10 [2,23]% for surface elevation, and 17 [2,34]% for velocity. Furthermore, the MAE is also reduced overall, with a decrease observed in 70% of the cases.

Notably, significant differences are observed between the CRPS of the different sub-basins: for the cumulative ice discharge and ice discharge, the CRPS rarely increases for UI-S and UI-SS, while the opposite is true for UI-N and UI-C. This disparity corresponds to lower CRPS values on cumulative ice discharge and ice discharge before calibration. For UI-S and UI-SS, the CRPS before calibration on cumulative ice discharge is 0.07 Gt and 0.06 Gt, respectively, compared to 0.22 Gt and 0.43 Gt for UI-N and UI-C. Similarly, their CRPS on ice discharge is also lower, at 1.1 Gt/a and 0.8 Gt/a, compared to 1.7 Gt/a and 4.1 Gt/a for UI-N and UI-C, respectively. The regions where CRPS is lower before calibration correspond to areas where the front retreat was not brief, as seen in the case of UI-N and UI-C. This can be attributed to the fact that, due to the inversion process, and with no major change in dynamics as observed for UI-N and UI-C, all the members are already capable of reproducing the observed data, rendering the calibration process less impactful on the inversion ensemble in these cases.

Additionally, a disparity is observed in the response of different data types, with notably greater reductions or increases observed for global data, such as cumulative ice discharge and ice discharge, compared to spatio-temporal data, such as surface elevation and velocity. Furthermore, no significant patterns are discernible between sub-basins concerning these spatio-temporal data.

## C1 Influence of the data used

Among the Full-period weightings using different data, the calibration using RMSE of both surface velocity and surface elevation (ZSxV) demonstrates the highest robustness in reducing the CRPS for various observations. For the UI-N sub-basin, ZSxV significantly reduces CRPS for cumulative ice discharge, ice discharge, surface elevation, and velocity by -8.4%, -0.3%, -5.4%, and -0.7%, respectively. Similarly, for the UI-C sub-basin, it leads to substantial reduction in CRPS for cumulative ice discharge, ice discharge, surface elevation, and velocity by -39%, -11.8%, -0.8%, and -5.7%, respectively. Additionally, ZSxV also contributes to reducing almost all the CRPS for surface elevation and velocity in the UI-S sub-basin by +1.3% and -2.3%, respectively, and in the UI-SS sub-basin by -1.5% and -0.8%. This is due to a reduction in MAE in the majority of cases (75%), as well by a reduction of -17 [-11, -25] % of the STD.





On the other hand, the cumulative ice discharge (CID) calibration enhances CRPS for the UI-N and UI-C sub-basins con-

cerning cumulative ice discharge (-18% and -28%), ice discharge (-1% and -14%), and velocity (-2% and -2%), but does not yield significant improvements for surface elevation (+0% and +1%). However, for the UI-S and UI-SS sub-basins, cumulative ice discharge calibration results in increased CRPS values (88 % of the cases). These observed increases are likely indicative of overfitting, as evidenced by a reduction in the MAE in most cases and the STD in all cases.

In the case of the ice discharge (ID) calibration, it primarily improves CRPS for ice discharge itself but does not have a

considerable impact on other observations, such as cumulative ice discharge, surface elevation, and velocity. For most instances, this improvement is accompanied by an increase in MAE, suggesting that the ice discharge calibration may not effectively identify the "best" members due to the presence of noisy or imprecise data.

Lastly, applying a weighting system based on surface elevation (ZS) or velocity (V) leads to improved CRPS for each type of observation on the UI-N and UI-C sub-basins, including cumulative ice discharge, ice discharge, surface elevation, and

velocity. However, the degree of improvement is less pronounced compared to the weighting with the combined use of both variables (ZSxV), e.g., -29% and -20% for ZS and V calibrations for cumulative ice discharge CRPS on UI-C, as opposed to -39% for ZSxV. Notably, there is a reduction in MAE, primarily for surface elevation and velocity observations, and STD, though not as significant as when employing the combined data. Consequently, relying solely on velocity or surface elevation (ZS) for calibration assignment appears to result in an under-utilisation of data.

**C2   Influence of the form of the probability density on ZSxV**

To assess the sensitivity of the calibration to the choice of probability density functions, we explore two distinct distributions: Gaussian (G) and Student's (S). These distributions are combined with three different estimates for the parameter $\sigma$, which are based on the minimum (min), median (med), and maximum (max) values within the ensemble of RMSEs. We do not show the results with the mean, which are almost identical to those with the median. In Figure C1, the first letter (G or S) denotes the

distribution type, while the second part signifies the specific $\sigma$ estimate employed.

Among the six different distribution shapes (G_min, G_max, G_med, S_min, S_max, S_med) used for calibration, the utilisation of the Student distribution leads to marginally lower CRPS compared to its Gaussian counterpart across all variables and basins, with an average reduction of -1.6% compared to -1.3%. In most cases, employing the Student's distribution for weighting exhibits lower reduction for STD (-14% against -15% in average) but lower increase of MAE (+0.18% against

+0.42% in average). However, based on these results, it remains inconclusive whether weighting using a Gaussian distribution yields a poorer CRPS because it is less close to observations with the MAE, or whether it is due to being too confident with the STD, i.e., too low a sharpness.

Similarly, calibration using the median or the mean of RMSE as an estimate of $\sigma$ demonstrates better CRPS in most cases compared to those using the minimum or maximum of RMSE as an estimate, which assigns respectively greater and lower

weights to members that fit the data best. Thus, it leads to an average CRPS reduction of -1.8% for the median, versus -1.0% for the minimum and -1.2% for the maximum. Weighting with the maximum leads to a reduction in the Mean Absolute Error (MAE) by an average of -0.69%, whereas using the minimum and median weights results in an increase of +1.2% and +0.43%,





respectively. However, when considering the STD, the values of the min-estimate are consistently lower, with reductions of -23% on average for the minimum compared to -15% and -6% for the median and maximum. Therefore, the relatively modest
reduction in CRPS with the minimum estimate can be attributed to its overconfident nature, whereas the limited reduction in CRPS with the maximum estimate is indicative of its underconfident nature. The median RMSE estimate for $\sigma$ appears to strike the best balance, with additional tests indicating that using the mean as an estimate of $\sigma$ yields results similar to those obtained with the median.

## C3    Use of sub-periods

The Sub-period weighting SP_mean, using the following characteristics : Student's distribution, mean of RMSEs for the estimate of $\sigma$, and utilisation of surface elevation and velocity data (ZSxV); does not yield improvements in the CRPS for ice discharge, surface elevation, and velocity (+8%, -0.9%, and -2% respectively, against an average of +4.6%, -1.6%, and -2.4%). However, a reduction in CRPS is observed for cumulative ice discharge (-10.6% against an average of -7.7%). Despite this, SP_mean leads to a decrease in the MAE of the mean in 69% of the cases and a significant decrease in STD for these variables
(-26% against an average of -14%). These findings indicate that the SP_mean weighting may be overconfident, as it excessively reduces the model's uncertainty.

To address the issue of overconfidence, alternative proxies for $\sigma$ were tested. It was found that using the 75th percentile (SP_Q75) of the RMSE distribution as a estimate for $\sigma$ resulted in better CRPS values for ice discharge, surface elevation, and velocity (+6.5%, -1.4%, -2.3% respectively, compared to +8%, -0.9%, and -2% for SP_mean previously). However, the
CRPS for cumulative ice discharge becomes worse and closer to S_med (-9.3% versus -10.6% for SP_mean previously). Consequently, we decided to analyze the results of this calibration choice as the Sub-period weighting.

## C4    The $f_{param}$ weighting

To evaluate the effectiveness of an ensemble that assigns greater significance to the parameterisation developed in Jager et al. (2024), we conducted an analysis of the $f_{param}$ weighting's performance in terms of CRPS, MAE, and STD. This also enables
us to assess whether the intricacies of the validation analysis conducted in the initial study, which encompassed elements such as initial state maps and the temporal evolution of velocity and surface elevation, can be effectively captured within the broader scope of this global analysis.

The CRPS analysis reveals that the $f_{param}$ weighting outperforms both the Full-period and Sub-period weightings for cumulative ice discharge and ice discharge, achieving reductions of -1.6% and -13% for P90, respectively. In contrast, the
SP_Q75 weighting shows CRPS changes of +5.5% and -7.7% for cumulative ice discharge and ice discharge, while the S_med weighting yields CRPS changes of +4.6% and -7.7% for the same variables, on average. However, for surface elevation and velocity, the $f_{param}$ weighting results in slightly lower CRPS reduction (-0.6% and -0.8% for P90) compared to the CRPS reduction of -1.6% and -2.2% for SP_Q75, and -1.6% and -2.4% for S_med on average.

Consistent patterns are observed for the MAE of the mean and the STD. For cumulative ice discharge and ice discharge, the
P90 weighting yields lower MAE values (-2% and +1.6% for P90, respectively) compared to the SP_Q75 weighting (+4.8%



and +7%). Similarly, the P90 weighting leads to lower STD values for cumulative ice discharge and ice discharge (-19.6% and -23.4%, respectively) in contrast to the SP_Q75 weighting (-19.1% and -21.4%). However, for surface elevation and velocity, the P90 weighting results in higher MAE values (-1.7% and -2.1%, respectively) compared to the SP_Q75 weighting (-3% and -5.3%). Similarly, the P90 weighting leads to higher STD values for surface elevation and velocity (-8.4% and -11.2%, respectively) in contrast to the SP_Q75 weighting (-10.6% and -19.6%).


*Author contributions.* EJ conducted the study through the supervision of FGC, NC and JM. RM and JM prepared and processed the satellite data of surface elevation, surface flow velocities and ice discharges. HG provided data and assistance for the retreat forcing approach. EJ wrote the paper with comments from all co-authors.

*Competing interests.* The contact author has declared that none of the authors has any competing interests.

*Acknowledgements.* This work was funded through the project SOSIce from the french Agence National de la Recherche (grant no. ANR-19-CE01-0011-01). EJ also benefits of the ICEMAP funding (Research Concil of Finland 355572). Most of the computations presented in this paper were performed using the GRICAD infrastructure (https://gricad.univ-grenoble-alpes.fr), which is supported by Grenoble research communities. HG has received funding from the European Union's Horizon 2020 Research and Innovation Programme under grant agreement no. 869304, PROTECT and from the Research Council of Norway under project 324639. High-performance computing and storage resources

were provided by Sigma2 - the National Infrastructure for High Performance Computing and Data Storage in Norway through projects NN8006K, NN8085K, NS8006K, NS8085K and NS5011K. SMB forcing data was provided in the context of PROTECT by Xavier Fettweis for MAR and Brice Noël for RACMO. This is PROTECT publication number #. The authors would also like to thank Hélène Seroussi and étienne berthier for their comments on an earlier version. To help with the writing of this manuscript, we used DeepL and chatGPT, for which we would like to acknowledge their help. The first was used to translate some sentences from French into English. The second was used to

reformulate certain paragraphs by asking it to rewrite the paragraph in a scientific style.



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



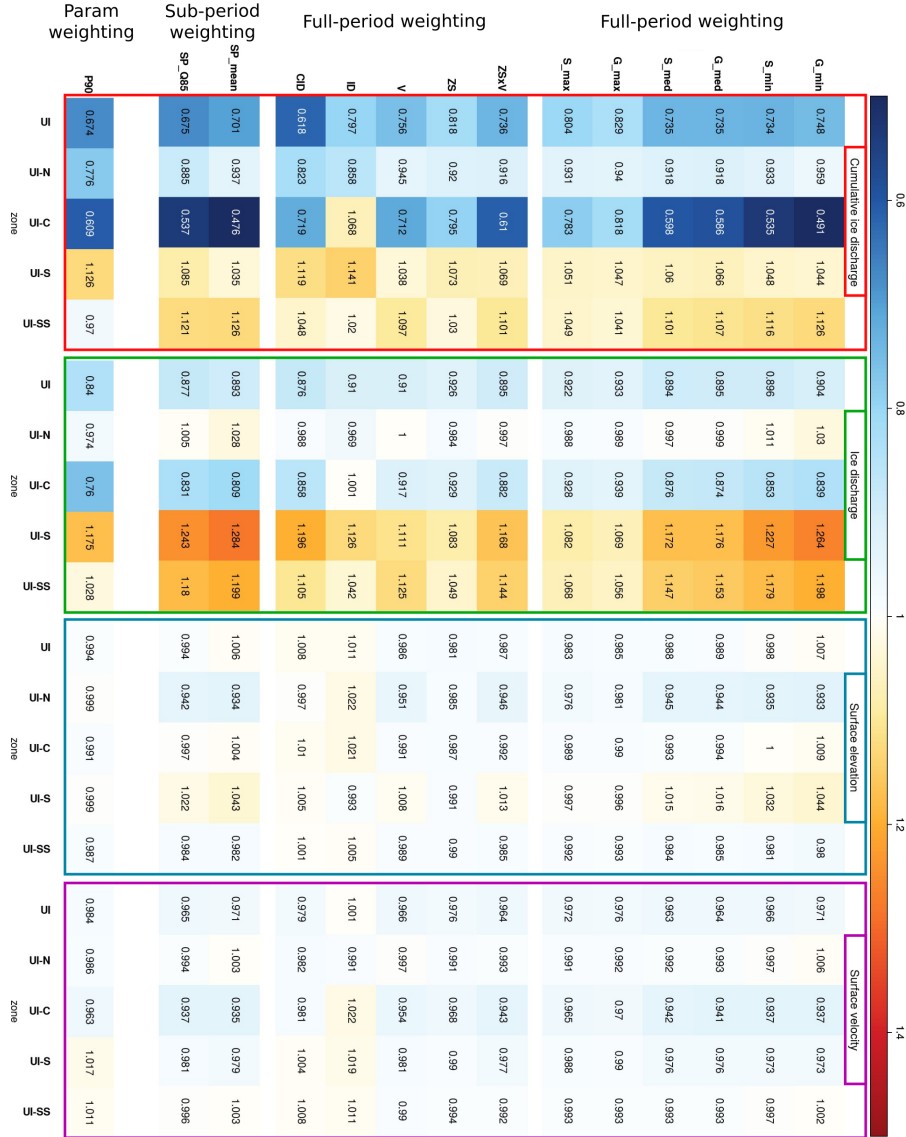

**Figure C1.** Normalized CRPS on cumulative ice discharge (red), ice discharge (green), surface elevation (blue) and surface velocity (purple) for different calibrated ensemble : Full-period weighting (G_min : Gaussian distribution with $\sigma = min(RMSEs)$; G_max : Gaussian distribution with $\sigma = max(RMSEs)$; G_med : Gaussian distribution with $\sigma = median(RMSEs)$; S_min : Student's distribution with $\sigma = min(RMSEs)$; S_max : Student's distribution with $\sigma = max(RMSEs)$; S_med : Student's distribution with $\sigma = median(RMSEs)$; ZSxV : calibration with RMSE on surface elevation and velocity; ZS : calibration with RMSE on surface elevation; V : calibration with RMSE on surface velocity), Sub-period weighting (SP_mean : $\sigma = mean(RMSEs)$; SP_Q75: $\sigma = quantile_{0.75}(RMSEs)$), $f_{param}$ weighting (P100 : $w_i = 1$ if $f_{param}^i = True$, else $w_i = 0$; P90 : $w_i = 0.9$ if $f_{param}^i = True$, else $w_i = 0.1$; P80 : $w_i = 0.8$ if $f_{param}^i = True$, else $w_i = 0.2$). The "UI" line represents the calibration using data from all four sub-basins and is evaluated over the entire validation area. On the other hand, the "UI-N," "UI-C," "UI-S," and "UI-SS" lines correspond to calibrations using data from the three other sub-basins, and they are evaluated over their respective sub-basins (Fig. 1).