# Peer review of "The future of Upernavik Isstrøm through ISMIP6 framework: Sensitivity analysis and Bayesian calibration of ensemble prediction"

_EGUsphere, 2024_

## Referee Comment (RC1)

**Review of 'The future of Upernavik Isstrøm through ISMIP6 framework: Sensitivity analysis and Bayesian calibration of ensemble prediction' by Jager et al.**

Doug Brinkerhoff

May 22, 2024

**1 Summary and Main Points**

This paper presents a probabilistic forecast for mass change at Upernavik Ice Stream in West Greenland. In very broad terms the forecast is produced by running an ensemble of predictions over different model parameters using the ice sheet model Elmer/Ice, and then weighting those ensemble members based on their agreement with a variety of observations. The paper is remarkable in that performs a very detailed methodological exploration of different schemes for weighting ensemble members. The paper also presents an index based sensitivity analysis, allowing for an interesting temporal discussion of the influence of variance in model parameters on the variance in predictions.

Overall, I think that this paper is exceptional and represents an important advance in data-constrained forecasting for Greenland. In particular, the paper makes (and justifies) a rather significant claim, which is that at centennial scales, the dominant source of uncertainty for projection remains elements of climate forcing and that contemporary observational constraint on ice sheet models exhibits diminishing returns. I have no major methodological issues with this work and I find the science to be sound. I do think that the paper could be made more accessible to a broad audience – in particular one that isn't complete versed in the language of probabilistic forecasting, and for whom this paper would still be a very useful read – by some clarification in exposition. In particular, I think that the use of acronyms could be reduced, some sections could be shortened, and the lanauge of Bayesian inference modified to be in closer accordance with standard usage.

**2 Line-by-line comments**

**L23** 'limited' used twice.

**L24** What is the difference between 'limited understanding of ice dynamics' and 'uncertainties in ISMs'?

**L30** It's worth noting that in Aschwanden (2021) the authors state that ISMIP6 actually already does a really good job with quantifying uncertainty in model structure! For the other considerations, yes, those must still be better accounted for.

**L34** This would be an appropriate reference for sensitivity studies in Greenland as well: https://doi.org/10.5194/tc-13-1349-2019

**L47** For item (i), I think it makes sense to characterize this as 'establishing prior distributions' over uncertaint model parameters.

**L50** It might be worth noting that this lack of cross-validation is often done because there's just not that much data to work with usually, but the authors' point generally a very fair one. I hope that future studies incorporate the authors' suggested cross-validation framework.

**L111** The specification of the calving front position rather than it being a prediction from the model was and remains one of the most contentious aspects of the ISMIP6 experiments. While the position of the calving front may be specified with precision by the parameterization, that doesn't necessarily mean that the position will be accurate in the future, and successfully simulating calving rates and front positions remains one of the largest open challenges in glaciology. It is worth noting here that by ignoring this source of uncertainty in model projections, the authors' are making a large and potentially critical assumption. I don't think it's a problem, but it really does need to be discussed.

**2.1.1–2.1.3** I struggle a bit with the semantic separation of the historic ensemble (hpr) from the two future ensembles (cpr and ppr), given that it is the initial period for both. I don't think it's too important but the authors for clarity may wish to refer to the historic ensemble as the 'shared hindcast period' or similar for the two prognostic ensembles.

**L141–150** I find this section to be a little bit confusing. It might help to have a more complete discription of the parameterization and specifically the role of $\kappa$.

**L195** 'Ice Discharge' should be lower-cased.

**L197** Given that this producet is based on a model result (RACMO), is there the potential for this product to contain systematic bias?

**L203** 'ensemblist' $\rightarrow$ ensemble

**L225** I think that $\mathbf{M}$ needs to be understood as a random vector of model parameters, with $P(\mathbf{M})$ its prior distribution. It will be the case that a sample will be drawn from this distribution, which will be used to create the ensemble, but $\mathbf{M}$ is not the ensemble itself.

**Eq. 6** I think that some of these components are mislabelled. In particular, Eq. 6 is not Bayes' theorem, so I'm not sure it makes sense to refer to a posterior and prior as such. Rather, Eq. 6 is the definition of the 'posterior predictive distribution' (which is what the left-hand side should be labelled), which is the distribution of future sea level outcomes conditioned on data. This is decomposed into the two terms on the right side,

$$P(\mathbf{M}|\mathcal{B}),$$

which is what would usually be called the 'posterior distribution', 'parametric posterior distribution', or 'calibration' (as it already is in the paper) to disambiguate, and

$$P(SLR|\mathbf{M}, \mathcal{B}),$$

which is the distribution of model predictions given a particular parameter value (perhaps called 'model prediction' or 'projection').

**Eq. 7** An important condition here is that

$$\mathbf{M}_i \sim P(\mathbf{M}),$$

which is to say that the realizations of the particles need to be drawn from the prior distribution. If that's not the case, then the prior needs to appear in the numerator and denominator of the term in L. 237.

**L240** 'gaussian' should be capitalized.

**Eq. 8** This intersection notation is weird – I think it would be better to just start with Eq. 9.

**Eq.11** $A$ should be $A^{n_{obs}}$ or a new constant should be used.

**L248** Be explicit as to what this measurement operator is, e.g. the evaluation of the Elmer/Ice FEM basis representation of the velocity field at the desired locations in space and/or time.

**L255** It is also worth noting that even if observational uncertainty were IID, model error *definitely* is not, which is what ultimately leads to the egregiously peaked distributions over ensemble members and heavily weighting only a single one. Ultimately the problem is that – priors aside – we don't know an appropriate likelihood function for comparing models to data! As such, the more *ad hoc* methodology described in this work is justified.

**L265–282** This section is really great. It has significant similarities to lots of previous work on Bayesian inference in the presence of model misspecification, and it might be worthwhile to frame the discussion in terms of that. This is a good reference: https://doi.org/10.1146/annurev-statistics-040522-015915

**L280** –282] I'm not sure I understand this statement.

**L304** It would be worth describing whether this weighting scheme is more or less permissive than full-period weighting - I don't have a good intuition. It might also be helpful to introduce an equation for each of the weighting schemes to be concrete.

**L308** I don't really understand the introduction of $f_{param}$ weighting. This is very much tied to the particular parameterization and behavior of the authors' existing model setup (thoroughly described in a separate paper) and it's challenging for someone not so involved with that work to understand what this specific experiment is trying to capture. Can this be expanded to provide more substantial justification?

**L232** This sentence changes from passive to active voice in the middle. Probably best to stick with active voice.

**Sec.2.3.3** While I appreciate the desire to include SSP as a random variable, I also think that doing so sort of obscures the influence of all the other aspects of the model since this is expected (and turns out to be) a dominant factor in determining predictive variance. Is it possible in later plots to also present ensemble ranges conditioned on SSP (i.e. the sub-ensembles of particles using just SSP2.6, SSP4.5, SSP8.5)? That would be helpful for comparison against existing similar work and would also facilitate a 'best-case versus worst-case' analysis for climate change impacts.

**L343** It would be super helpful to reiterate here what the difference is between the Ppr and the Cpr. I had initially thought to suggest more instructive names, but I can't think of any, so at the very least a brief reminder of the assumptions of each would be great.

**L356** Is the agreement between the Hpr median and mass loss observations by design or a happy accident?

**Fig. 2 and 3** Perhaps consider changing the symbology scheme to something friendly for greyscale/colorblindness, e.g. cross-hatching one of the two shaded regions.

**L361** It might be worth contextualizing this with respect to Robel, 2019: https://doi.org/10.1073/pnas.190482211. The skew in the distribution is perhaps not surprising.

**Sec. 3.3.1** It's a little cumbersome to start a section with a reference to another section. I understand shunting the methodological details to the appendix, but a recapitulation of the methodological approach would be helpful here.

**Sec. 3.3.1** More generally, all four points introduced here seem a bit *ad hoc*. Do there exist references that could help place the current procedure on more sound probabilistic footings? Seems like this problem has to have been studied before.

**L446** Where is factor mapping previously established?

**L448** The parameter $f_{law}$ sometimes appears throughout the manuscript as just *law*. Please revise for consistency.

**Fig. 5** The overlying transparent bars aren't really readable.

**Sec 3.4.1** I think that this section would benefit from a bit of extra subdivision. I think it would be helpful to break this into individual subheadings describing the historical period and the forecasts. I think it would also be helpful to separate the principal conclusions about the relative insensitivity of long term forecasts to ISM parameters from the details of weighting. I also don't think that it makes sense to refer to the changes in ranges described around L506 as 'notable' – the more notable thing is that they're almost exactly the same!

**Fig. 6** The font in this figure is too small.

**Sec 3.4.2** Again, I would like to reiterate that presenting ensemble results which each of the SSPs held fixed would be useful here, and would help to ameliorate some of the challenges associated with trying to guess the probabilities of future human carbon emissions (which is why previous works have treated these as fixed hypotheses rather than as random variables).

**Sec. 4.1** I am not quite sure I understand the reference to ISMIP6 here. How is that relevant to the present model being able to reproduce observations?

**Sec. 4.2** Again, I think that this section would be clarified by adding some more sub-headings. e.g. at L589, this paragraph could be called 'reducing uncertainty through ISM calibration), whereas at L598, this could be called 'reducing uncertainty through climate forcing calibration', or something like that.

**L576** The referenced compensatory effect is not clear to me from Fig. 4 or elsewhere. Could this please be clarified?

**L585** If the front parameterization has such as significant influence, then perhaps this calls into question the validity of imposing the front at all. Would it be possible to make a statement about how the predictions might be influenced if the front were allowed to evolve freely or based on a different parameterization?

**L602–604** This is a very significant assertion that would have major implications for how ice sheet modeling proceeds in the future! What should the community be doing if improving ice dynamics isn't likely to help? (note that I don't disagree with the assertion – I am genuinely curious where effort should be allocated instead).

**L618** Its foundation in observational data is sort of the problem – no data available in the future.

**L703** There are other possibilities than the Gaussian or T.

**Sec. 4.4** There is a lot of good in this section, but there is also a lot of material that is only applicable to the authors' own modeling setup (which has already been covered) it might be worthwhile to take a critical read to assess which of these insights are going to be generally applicable, and which are more like notes to guide the authors' own continuing work.

**Sec. C2** The student-t distribution has an additional degree of freedom, namely the number of degrees of freedom. What was used for this, or how was it estimated?

---

## Referee Comment (RC2)

**Review of "The future of Upernavik Isstrøm through ISMIP6 framework: Sensitivity analysis and Bayesian calibration of ensemble prediction" by Jager et al.**

In the paper 'The future of Upernavik Isstrøm through ISMIP6 framework: Sensitivity analysis and Bayesian calibration of ensemble prediction', Jager and co-authors study several aspects associated with the evolution of the Upernavik Isstrøm Glacier, Greenland. Based on a statistical framework, numerical results obtained with the Elmer/Ice finite-element code, and observational data, they quantify the uncertainties associated with predictions of the future sea-level contribution. They improve the robustness of their analysis by considering a cross-validation step and by studying several weighting methods for assigning a likelihood score to the uncertain parameters.

I think that the paper will make a great addition to the scientific literature as it deals with an important topic, namely the quantification of uncertainties, and, more generally, the study of the methods that are used to produce such analyses. Nonetheless, I have a series of comments that I would like the authors to address prior to the publication of the manuscript. As described hereafter, those are mainly related to the form of the paper, rather than its scientific content.

**General comments**

My main comment is related to the way the paper is written. I have found the methods and results to be particularly interesting, but the style of the paper makes it quite difficult to grasp them efficiently. My main complaints are that the whole paper is very long (45 pages), that some parts are difficult to follow because of the lack of visual data, and that some subsections are particularly long. I would suggest the following changes:

- Streamlining the manuscript, in particular by focusing on the key points in each paragraph, and removing unnecessary repetitions.

- Focusing on the novel aspects of your study. To my understanding, these are the cross-validation (which I believe has not really been done previously in a glaciological context), and the use of different weighing methods.

- Adding figures/tables/schematics that allow to understand the content of the text in a visual and summarized way. For example, in Section 2, you could create a table with the different scenarios (Hpr, Cpr, Ppr), and for each of these scenarios specify the SMB that is used, as well as the front position and the uncertain parameters. For the observational data, you could create a table with the different types of observations that you have, their type, and where they come from.

- I wonder if the $f_{\text{param}}$ weighing makes much sense, giving that this parameter is one of the uncertain parameters that are calibrated in the Bayesian process. Note that a classical way to favor specific values of $f_{\text{param}}$, given your knowledge of its importance, would be to change its prior distribution.

To reiterate, I find the paper to be both useful and significant. But I still think that it is important to improve its style, as it will help the audience to better understand the key points presented in the manuscript.

**Specific comments**

(1) [Line 24] It is a bit unclear at this stage what distinguishes the 'limited understanding of ice dynamics ' and 'uncertainties in Ice Sheet Models'. Maybe specify what you mean for the latter, e.g., 'uncertainties in the parameters of Ice Sheet Models'.

(2) [Line 34] A paper that is missing for Antarctica is Bulthuis et al. (2019).

(3) [Line 63] 'initialisation' → 'initialization' as you use American English in your manuscript. Also check Lines 97, 158, 340, 546, 756, 774, 775, 792, and 864.

(4) [Line 65] The use of the active voice in this sentence is a bit weird here, given that the rest of the paragraph is written with the passive voice.

(5) [Line 168] I am guessing that the equation mentioned here should be Eq. 1, not Eq. 4.

(6) [Line 168] It is a bit confusing that the sensitivity indices $S_i$ are called 'first-order sensitivity indices' here, and not before. I would suggest discussing why the $S_i$ are called 'first-order indices', or directly mentioning Line 161 that the indices that you introduce are of first order. Otherwise, the reader might wonder which indices you are talking about in this paragraph, as it is not clear that you are talking about the $S_i$ indices.

(7) [Line 172] 'Y' needs to be written in italics ($Y$) here.

(8) [Equation 6] The first factor is not a prior distribution for the problem considered in the paper. Going back to Aschwanden and Brinkerhoff (2022), a possible name for this factor would be 'projection'.

The distinction between prior and posterior distributions (i.e., Bayes' theorem) appears later, implicitly, through the computation of the term $P(\mathbf{M}|\mathcal{B})$ in equation (6). Specifically, Bayes' theorem writes

$$P(\mathbf{M}|\mathcal{B}) = \frac{P(\mathcal{B}|\mathbf{M})P(\mathbf{M})}{P(\mathcal{B})} = \frac{P(\mathcal{B}|\mathbf{M})P(\mathbf{M})}{\int P(\mathcal{B}|\mathbf{M})P(\mathbf{M})\,\mathrm{d}\mathbf{M}}, \tag{R1}$$

where:

- $P(\mathbf{M}|\mathcal{B})$ is the posterior distribution;
- $P(\mathcal{B}|\mathbf{M})$ is the likelihood distribution;
- $P(\mathbf{M})$ is the prior distribution.

For all intents and purposes, you will find at the end of this review a few equations that show how, starting from (R1), I arrive at your equation (7).

(9) [Line 214] Ideally, you should define every variable that appear in the equations, so $F_m^j, F_o^j, Q_{m,i}^j, \ldots$ should be defined. To save space, it makes sense no to do so, but please at least mention in this paragraph that the subscript $i$ is associated with the $i$-th member of the ensemble and that the superscript $j$ is associated with the $j$-th observation.

(10) [Line 216] I am guessing there is an 'it' missing before 'is common' here.

(11) [Line 237] It really is a detail, but please avoid using fractions within the text. Instead, write the definition of $w_i$ as a full new equation, or write it as $w_i = P(\mathcal{B}|\mathbf{M}_i)/\sum_{k=1}^{n_m} P(\mathcal{B}|\mathbf{M}_k)$. Same comment for the factions that appear later on in the text.

(12) [Equations (8)–(12)] I suggest removing equations (8)–(10), as these equations do not add much to the discussion, and might even appear unnecessarily technical. It seems to me that the reader should be able to deduce from the Gaussian and independence assumptions that $P(\mathcal{B}|\mathbf{M}_i)$ has the form shown in (11), which is quite standard.

(13) [Line 248] Technically, $H$ is the measurement operator, not $H(\mathbf{M}_i)$ (which is the value taken by this operator when $\mathbf{M} = \mathbf{M}_i$).

(14) [Line 266] To be consistent, write $f(RMSE, \sigma)$, not just $f(RMSE)$.

(15) [Line 271] Please read again this paragraph, it seems that you repeat yourself.

(16) [Subsection 2.3.1] Overall, I think that this subsection is not structured in an efficient way: you first present the equations (12) and (13), corresponding to the 'classical' approach. Reading the beginning of this section, it seemed to me that you are going to use those expressions. But then you discuss limitations (which always is a real plus), and consequently modify you formulas. It might make more sense to directly state that while expressions (12) and (13) are the usual approach, you are not going to use them, and instead will use the formula (14) instead. On a related note, it is a bit surprising that you mention Line 249 that $\sigma$ is the standard deviation of the observation error (while it is common, as you mention later, to include the model error in it). So maybe directly state the difficulty associated with $\sigma$, and that your equation (14) is a possible solution for it.

(17) [Line 294] I wonder if the discussion of the assumptions that must be examined should not be included in the 'full-period weighting' item, Line 300.

(18) [Line 382] I do not agree with the contradiction indicated by the 'On the contrary' here: the fact that the sum of the first-order Sobol indices is greater than one does not prevent a substantial impact of specific parameter combinations. Furthermore, the fact that the sum of the first-order Sobol indices is smaller or equal than one does guarantee that the inputs are independent.

(19) [Line 462] $law \rightarrow f_{\text{law}}$.

(20) [Line 462] The fact that the priors and posteriors distributions are similar for several parameters is an important result. Maybe you could elaborate on that, both in terms of the interpretation that you give to this observation, and the conclusions that can be drawn for it.

(21) [Line 468] As before, this 'posterior ensemble' is a bit confusing as you are looking at the distribution of ice mass discharge, rather than the distributions of the inferred parameters (which have been analyzed in the previous subsection). Maybe use another name for this subsection, or precise in that name that you are going to talk about SLR predictions.

(22) [Figure 5] This figure is difficult to read. Consider using brighter colors and larger labels.

(23) [Line 553] It's $\rightarrow$ It is.

(24) [Line 595] dynamics modeling community $\rightarrow$ ice-sheet dynamics modeling community?

(25) [Line 660] it's $\rightarrow$ it is.

(26) [Line 746] Would that still be true if you looked beyond 2100? I have in mind the study of Brondex et al. (2017, 2019) which showed that the form of friction laws does have a strong impact (in particular, there is a significant difference between the regularized Coulomb and Budd laws).

(27) [Line 789] Maybe add that 'SSA' also stands for Shallow-Shelf Approximation.

(28) [Lines 811, 843, 876] regularisation $\rightarrow$ regularization.

(29) [Line 840] Maybe add that this value of $u_0$ is similar to the one chosen in Joughin et al. (2019).

(30) [Line 884] $law \rightarrow f_{\text{law}}$.

**Additional equations – comment (8)**

If $P(\mathbf{M})$ is a discrete distribution, i.e., if $\mathbf{M}$ can only take a finite number of values in the set $\{\mathbf{M}_1, ..., \mathbf{M}_{n_m}\}$, and if these values are equally probable, then

$$P(\mathbf{M}) = \frac{1}{n_m} \sum_{i=1}^{n_m} \delta(\mathbf{M} - \mathbf{M}_i), \tag{R2}$$

in which the factor $1/n_m$ is the normalization constant, necessary to obtain a distribution. Introducing this expression in (R1) yields

$$P(\mathbf{M}|\mathcal{B}) = \frac{P(\mathcal{B}|\mathbf{M}) \sum_{i=1}^{n_m} \delta(\mathbf{M} - \mathbf{M}_i)}{\int P(\mathcal{B}|\mathbf{M}) \sum_{i=1}^{n_m} \delta(\mathbf{M} - \mathbf{M}_i) \, d\mathbf{M}}. \tag{R3}$$

This can be simplified: on the one hand, we have

$$P(\mathcal{B}|\mathbf{M}) \sum_{i=1}^{n_m} \delta(\mathbf{M} - \mathbf{M}_i) = \sum_{i=1}^{n_m} P(\mathcal{B}|\mathbf{M}) \delta(\mathbf{M} - \mathbf{M}_i) \tag{R4a}$$

$$= \sum_{i=1}^{n_m} P(\mathcal{B}|\mathbf{M}_i) \delta(\mathbf{M} - \mathbf{M}_i). \tag{R4b}$$

On the other hand,

$$\int P(\mathcal{B}|\mathbf{M}) \sum_{i=1}^{n_m} \delta(\mathbf{M} - \mathbf{M}_i) \, d\mathbf{M} = \sum_{i=1}^{n_m} P(\mathcal{B}|\mathbf{M}_i). \tag{R5}$$

Combining these results together yields

$$P(\mathbf{M}|\mathcal{B}) = \sum_{i=1}^{n_m} w_i \, \delta(\mathbf{M} - \mathbf{M}_i), \quad w_i = \frac{P(\mathcal{B}|\mathbf{M}_i)}{\sum_{k=1}^{n_m} P(\mathcal{B}|\mathbf{M}_k)}, \tag{R6}$$

which is precisely equation (7).

**References**

Aschwanden, A. and Brinkerhoff, D. J. (2022). Calibrated mass loss predictions for the greenland ice sheet. *Geophysical Research Letters*, 49(19).

Brondex, J., Gagliardini, O., Gillet-Chaulet, F., and Durand, G. (2017). Sensitivity of grounding line dynamics to the choice of the friction law. *Journal of Glaciology*, 63(241):854–866.

Brondex, J., Gillet-Chaulet, F., and Gagliardini, O. (2019). Sensitivity of centennial mass loss projections of the Amundsen basin to the friction law. *The Cryosphere*, 13(1):177–195.

Bulthuis, K., Arnst, M., Sun, S., and Pattyn, F. (2019). Uncertainty quantification of the multi-centennial response of the antarctic ice sheet to climate change. *The Cryosphere*, 13(4):1349–1380.

Joughin, I., Smith, B. E., and Schoof, C. G. (2019). Regularized Coulomb Friction Laws for Ice Sheet Sliding: Application to Pine Island Glacier, Antarctica. *Geophysical Research Letters*, 46(9):4764–4771.

---

## Author Comment (AC1)

Original reviewers comments are in *italic and black*, our answers are in blue.

**1 Summary and Main Points**

*In the paper 'The future of Upernavik Isstrøm through ISMIP6 framework: Sensitivity analysis and Bayesian calibration of ensemble prediction', Jager and co-authors study several aspects associated with the evolution of the Upernavik Isstrøm Glacier, Greenland. Based on a statistical framework, numerical results obtained with the Elmer/Ice finite-element code, and observational data, they quantify the uncertainties associated with predictions of the future sea-level contribution. They improve the robustness of their analysis by considering a cross-validation step and by studying several weighting methods for assigning a likelihood score to the uncertain parameters.*

*I think that the paper will make a great addition to the scientific literature as it deals with an important topic, namely the quantification of uncertainties, and, more generally, the study of the methods that are used to produce such analyses. Nonetheless, I have a series of comments that I would like the authors to address prior to the publication of the manuscript. As described hereafter, those are mainly related to the form of the paper, rather than its scientific content.*

**2 General**

*My main comment is related to the way the paper is written. I have found the methods and results to be particularly interesting, but the style of the paper makes it quite difficult to grasp them efficiently. My main complaints are that the whole paper is very long (45 pages), that some parts are difficult to follow because of the lack of visual data, and that some subsections are particularly long. I would suggest the following changes:*

*• Streamlining the manuscript, in particular by focusing on the key points in each paragraph, and removing unnecessary repetitions.*

*• Focusing on the novel aspects of your study. To my understanding, these are the cross-validation (which I believe has not really been done previously in a glaciological context), and the use of different weighing methods.*

*• Adding figures/tables/schematics that allow to understand the content of the text in a visual and summarized way. For example, in Section 2, you could create a table with the different scenarios (Hpr, Cpr, Ppr), and for each of these scenarios specify the SMB that is used, as well as the front position and the uncertain parameters. For the observational data, you could create a table with the different types of observations that you have, their type, and where they come from.*

*• I wonder if the fparam weighing makes much sense, giving that this parameter is one of the uncertain parameters that are calibrated in the Bayesian process. Note that a classical way to favor specific values of fparam , given your knowledge of its importance, would be to change its prior distribution.*

*To reiterate, I find the paper to be both useful and significant. But I still think that it is important to improve its style, as it will help the audience to better understand the key points presented in the manuscript.*

We thank the anonymous reviewer for their constructive review and positive comments. To improve the readability of the overall paper, we will make the following changes:

• We will try to clean up text that is redundant or heavy to read.

• We will change the structure of the headings in the results section to better highlight the two main parts of our study: the sensitivity analysis and the Bayesian calibration. We still think that these two parts are novel in our study, not only the novelties of the Bayesian calibration. No other study has explored the uncertainties of the ISMIP6 framework as exhaustively as we do here, with 3 SSPs and various RCMs.

• We will add two summarizing figures of the method. The first one will describe the different ensembles of the study to show their differences in forcing and summarize the different elements taken into account for the sensitivity analysis. The second one will summarize our methodology to produce robust Bayesian calibration. We hope this will help the readers and will also reduce the size of the text.

• We will add some justification for the use of the fparam weighting. It still makes sense for us to use it because it allows us to see the effect of the parameterisation developed in Jager et al. (2024) on the projections and shows that taking into account the effect of subglacial hydrology, at least in a parameterized way, significantly increases the projected mass loss of Upernavik Isstrøm. This also lets us evaluate this weighting against others to check if they can underscore the use of the parameterisation without the extensive detail used in the earlier study.

**3 Specific comments**

*(1) [Line 24] It is a bit unclear at this stage what distinguishes the 'limited understanding of ice dynamics ' and 'uncertainties in Ice Sheet Models'. Maybe specify what you mean for the latter, e.g., 'uncertainties in the parameters of Ice Sheet Models'.*

Yes it's quite similar, so we will change "ice dynamics" by "initial state".

*(2) [Line 34] A paper that is missing for Antarctica is Bulthuis et al. (2019).*

Agree, we will add it.

*(3) [Line 63] 'initialisation' → 'initialization' as you use American English in your manuscript. Also check Lines 97, 158, 340, 546, 756, 774, 775, 792, and 864.*

Thanks, we will correct it.

*(4) [Line 65] The use of the active voice in this sentence is a bit weird here, given that the rest of the paragraph is written with the passive voice.*

Agreed, we will change the sentence.

*(5) [Line 168] I am guessing that the equation mentioned here should be Eq. 1, not Eq. 4.*

Thanks, we will correct it.

*(6) [Line 168] It is a bit confusing that the sensitivity indices Si are called 'first-order sensitivity indices' here, and not before. I would suggest discussing why the Si are called 'first-order indices', or directly mentioning Line 161 that the indices that you introduce are of first order. Otherwise, the*

*reader might wonder which indices you are talking about in this paragraph, as it is not clear that you are talking about the Si indices.*

Agreed, we will change «the following indices» to «the first-order sensitivity indices» in the line 161 and change «Accurately computing sensitivity indices usually requires [...]» to «Accurately computing sensitivity indices of an order greater than one usually requires [...]» in the line 165.

*(7) [Line 172] 'Y' needs to be written in italics (Y ) here.*

Thanks, we will correct it.

*(8) [Equation 6] The first factor is not a prior distribution for the problem considered in the paper. Going back to Aschwanden and Brinkerhoff (2022), a possible name for this factor would be 'projection'.*

*The distinction between prior and posterior distributions (i.e., Bayes' theorem) appears later, implicitly, through the computation of the term P(M|B ) in equation (6). Specifically, Bayes' theorem writes*

$$P(M|B) = \frac{P(B|M)P(M)}{P(B)} = \frac{P(B|M)P(M)}{\int P(B|M)P(M)\,dM}$$

*where:*

*• P(M|B ) is the posterior distribution;*

*• P(B |M) is the likelihood distribution;*

*• P(M) is the prior distribution.*

*For all intents and purposes, you will find at the end of this review a few equations that show how, starting from (R1), I arrive at your equation (7).*

We had a similar comment from the other reviewer and will relabel the two terms from "Prior" and "Posterior" to respectively "Model Prediction" and "Posterior Prediction."

*(9) [Line 214] Ideally, you should define every variable that appear in the equations, so $F_m^j$ , $F_o^j$ , $Q^j_{m,i}$ , ... should be defined. To save space, it makes sense no to do so, but please at least mention in this paragraph that the subscript i is associated with the i-th member of the ensemble and that the superscript j is associated with the j-th observation.*

Yes agreed, we will add the following sentence: «The subscript i is associated with the i-th member of the ensemble and the superscript j is associated with the j-th observation.»

*(10) [Line 216] I am guessing there is an 'it' missing before 'is common' here.*

Thanks, we will correct it.

*(11) [Line 237] It really is a detail, but please avoid using fractions within the text. Instead, write the definition of wi as a full new equation, or write it as $w_i = P(B |M_i )/\sum_{k=1}^{n} P(B |M_k )$. Same comment for the factions that appear later on in the text.*

Agreed, we will change it to a new full equation while we will change the writing of the other ones.

*(12) [Equations (8)–(12)] I suggest removing equations (8)–(10), as these equations do not add much to the discussion, and might even appear unnecessarily technical. It seems to me that the*

*reader should be able to deduce from the Gaussian and independence assumptions that P(B |Mi ) has the form shown in (11), which is quite standard.*

Agreed, we will remove these equations.

*(13) [Line 248] Technically, H is the measurement operator, not H(Mi ) (which is the value taken by this operator when M = Mi ).*

Agreed, we will change this part of the sentence from «H(Mi) is the measurement operator corresponding to Qm,i» to «H is the measurement operator with H(Mi) corresponding to Qm,i»

*(14) [Line 266] To be consistent, write f (RMSE, σ), not just f (RMSE).*

We will delete this paragraph as it duplicate the one below.

*(15) [Line 271] Please read again this paragraph, it seems that you repeat yourself.*

Agreed. As said in the previous comment, we will delete the duplication.

*(16) [Subsection 2.3.1] Overall, I think that this subsection is not structured in an efficient way: you first present the equations (12) and (13), corresponding to the 'classical' approach. Reading the beginning of this section, it seemed to me that you are going to use those expressions. But then you discuss limitations (which always is a real plus), and consequently modify you formulas. It might make more sense to directly state that while expressions (12) and (13) are the usual approach, you are not going to use them, and instead will use the formula (14) instead. On a related note, it is a bit surprising that you mention Line 249 that σ is the standard deviation of the observation error (while it is common, as you mention later, to include the model error in it). So maybe directly state the difficulty associated with σ, and that your equation (14) is a possible solution for it.*

Thanks for the comment, we will restructure this section and rewrite partially some paragraphs for a smoother reading. We will start with the presentation of the different equations and mentioning than we will use equation 14. We will then explain the limitation of equation 13 and how the equation 14 allows us to bypass these limits.

About σ, we will add some elements in the paragraph dedicated to it. We will precise that it corresponds to the standard deviation of the observation error in equation 13, but in equation 14, it takes into account the structural error of the model as done in previous works (Murphy et al., 2009; Nias et al., 2019; Edwards et al., 2019).

*(17) [Line 294] I wonder if the discussion of the assumptions that must be examined should not be included in the 'full-period weighting' item, Line 300.*

Agreed, we will add it after the description of the full-period weighting.

*(18) [Line 382] I do not agree with the contradiction indicated by the 'On the contrary' here: the fact that the sum of the first-order Sobol indices is greater than one does not prevent a substantial impact of specific parameter combinations. Furthermore, the fact that the sum of the first-order Sobol indices is smaller or equal than one does guarantee that the inputs are independent.*

Agreed, it wasn't very clear that "on contrary" was there only to position the "smaller than 1". So, we will change the «On the contrary» to «otherwise».

*(19) [Line 462] law → flaw .*

Thanks, we will correct it.

*(20) [Line 462] The fact that the priors and posteriors distributions are similar for several parameters is an important result. Maybe you could elaborate on that, both in terms of the interpretation that you give to this observation, and the conclusions that can be drawn for it.*

We mention it briefly in the appendix B3: «In hindsight, our initial choice of distribution for these three parameters proves to be suitable due to the absence of significant changes observed in their posterior distributions.». We agree that it's an important result for our future perspectives but remains restricted to our study. Not everyone will use the same range of parameters because it can be specific to our catchment area. Moreover, you may need different parameters if you are not studying a tidewater glacier, you use an other ISM or you don't use the same framework than us.

*(21) [Line 468] As before, this 'posterior ensemble' is a bit confusing as you are looking at the distribution of ice mass discharge, rather than the distributions of the inferred parameters (which have been analyzed in the previous subsection). Maybe use another name for this subsection, or precise in that name that you are going to talk about SLR predictions.*

To clarify the overall structure, we have removed this sub-title (see answer to main comments).

*(22) [Figure 5] This figure is difficult to read. Consider using brighter colors and larger labels.*

Agreed, we will add hatch patterns and increase the label fontsize.

*(23) [Line 553] It's → It is.*

Thanks, we will correct it.

*(24) [Line 595] dynamics modeling community → ice-sheet dynamics modeling community?*

Thanks, we will correct it.

*(25) [Line 660] it's → it is.*

Thanks, we will correct it.

*(26) [Line 746] Would that still be true if you looked beyond 2100? I have in mind the study of Brondex et al. (2017, 2019) which showed that the form of friction laws does have a strong impact (in particular, there is a significant difference between the regularized Coulomb and Budd laws).*

To make clearer our message about the past reproduction of observations, we will add few words at the end of the sentence: «[...] in reproducing past acceleration of UI».

Otherwise, to answer the question, we think than our figure B1 gives some keys: the impact of the shape of the friction law does not have a major impact in 2100 unlike the use of the parameterisation developed in Jager et al. (2024). In our point of view, the main difference with Brondex et al. is this choice of proxy for the effective pressure N. In their case, they assume a perfect connection to the ocean with N proportional to the height above flotation, while Jager et al. (2024) and Joughin et al. (2019) both use a cut-off (the distance to the front in our case, the height above flotation in the case of Joughin). It means than far enough to the grounding line/the front position, the friction at the base is independent to the height above flotation/distance to the front regardless to the friction law used. On the contrary for Brondex et al., in case of Budd law, it will lead to a dependence to the effective pressure everywhere. However, this will not be the case for a regularised coulomb law because you will have some areas in a Weertman regime where friction only depends on the friction parameter $A_s$.

*(27) [Line 789] Maybe add that 'SSA' also stands for Shallow-Shelf Approximation.*

Agreed, we will add the following words: «also called Shallow-Shelf Approximation»

*(28) [Lines 811, 843, 876] regularisation → regularization.*

Thanks, we will correct it.

*(29) [Line 840] Maybe add that this value of u0 is similar to the one chosen in Joughin et al. (2019).*

Agreed, we will add the following words: «[...], which is similar to the median value of our previous study and to the one used in Joughin et al. (2019).»

*(30) [Line 884] law → flaw .*

Thanks, we will correct it.

---

## Author Comment (AC2)

Original reviewers comments are in *italic and black*, our answers are in blue.

**1 Summary and Main Points**

*This paper presents a probabilistic forecast for mass change at Upernavik Ice Stream in West Greenland. In very broad terms the forecast is produced by running an ensemble of predictions over different model parameters using the ice sheet model Elmer/Ice, and then weighting those ensemble members based on their agreement with a variety of observations. The paper is remarkable in that performs a very detailed methodological exploration of different schemes for weighting ensemble members. The paper also presents an index based sensitivity analysis, allowing for an interesting temporal discussion of the influence of variance in model parameters on the variance in predictions.*

*Overall, I think that this paper is exceptional and represents an important advance in data-constrained forecasting for Greenland. In particular, the paper makes (and justifies) a rather significant claim, which is that at centennial scales, the dominant source of uncertainty for projection remains elements of climate forcing and that contemporary observational constraint on ice sheet models exhibits diminishing returns. I have no major methodological issues with this work and I find the science to be sound. I do think that the paper could be made more accessible to a broad audience – in particular one that isn't complete versed in the language of probabilistic forecasting, and for whom this paper would still be a very useful read – by some clarification in exposition. In particular, I think that the use of acronyms could be reduced, some sections could be shortened, and the lanauage of Bayesian inference modified to be in closer accordance with standard usage.*

We thank Doug Brinkerhoff for his very positive comments. It is truly gratifying to receive such positive feedback about our paper. As suggested, we will add clarifications to make the paper more accessible to a broader audience. In particular, we will include figures that summarize our methodology, which will help to streamline some sections.

**2 Line-by-line comments**

**L23** *'limited' used twice.*

Thanks, we will change the term.

**L24** *What is the difference between 'limited understanding of ice dynamics' and 'uncertainties in ISMs'?*

Agreed, we will change "ice dynamics" by "initial state".

**L30** *It's worth noting that in Aschwanden (2021) the authors state that ISMIP6 actually already does a really good job with quantifying uncertainty in model structure! For the other considerations, yes, those must still be better accounted for.*

Yes, we will change the sentence to add this mention.

**L34** *This would be an appropriate reference for sensitivity studies in Greenland as well: https://doi.org/10.5194/tc-13-1349-2019*

Thanks for the reference. It seems that this paper is on Antarctica, so we will add it with the reference to Hill et al.

**L47** *For item (i), I think it makes sense to characterize this as 'establishing prior distributions' over uncertain model parameters.*

Agree, we will change the sentence.

**L50** *It might be worth noting that this lack of cross-validation is often done because there's just not that much data to work with usually, but the authors' point generally a very fair one. I hope that future studies incorporate the authors' suggested cross-validation framework.*

We partially agree with this comment. Despite the limited data availability, we maintain that dividing the dataset into calibration and validation subsets is always feasible and should not detract from the robustness of the results. However, in scenarios where there are no significant variations, such as unchanged ice dynamics during the validation period, this approach might not significantly enhance the robustness of the findings. To address the concern of data scarcity, we will add a mention to it.

**L111** *The specification of the calving front position rather than it being a prediction from the model was and remains one of the most contentious aspects of the ISMIP6 experiments. While the position of the calving front may be specified with precision by the parameterization, that doesn't necessarily mean that the position will be accurate in the future, and successfully simulating calving rates and front positions remains one of the largest open challenges in glaciology. It is worth noting here that by ignoring this source of uncertainty in model projections, the authors' are making a large and potentially critical assumption. I don't think it's a problem, but it really does need to be discussed.*

The sentence to which the reviewer refers is for the historical period, and so the position comes from observations whose uncertainties are unknown. However, as described below for the future (Ppr), this uncertainty is taken into account (as in ISMIP6) by adjusting the sensitivity parameter of the parameterisation. This choice of parameterisation is also discussed in section 4.2.2 where we agree that it remains a major challenge.

**2.1.1–2.1.3** *I struggle a bit with the semantic separation of the historic ensemble (hpr) from the two future ensembles (cpr and ppr), given that it is the initial period for both. I don't think it's too important but the authors for clarity may wish to refer to the historic ensemble as the 'shared hindcast period' or similar for the two prognostic ensembles.*

We will change the term Hpr to that suggested by the reviewer. To clarify this point, we will also add a figure to summarise our framework.

**L141–150** *I find this section to be a little bit confusing. It might help to have a more complete discription of the parameterization and specifically the role of $\kappa$.*

We will add a description of the parameterisation.

**L195** *'Ice Discharge' should be lower-cased.*

Thanks, we will correct it

**L197** *Given that this producet is based on a model result (RACMO), is there the potential for this product to contain systematic bias?*

Yes, it could, even though we have shown in our previous paper that members using RACMO outputs lead to less bias in surface reconstruction than those using MAR (Jager et al. 2024). Potential bias of the input-output method can be find in Mouginot et al. (2019), so we will not change the text.

**L203** *'ensemblist' → ensemble*

Thanks, we will correct it

**L225** *I think that M needs to be understood as a random vector of model parameters, with P (M) its prior distribution. It will be the case that a sample will be drawn from this distribution, which will be used to create the ensemble, but M is not the ensemble itself.*

Agreed, we will change it "by considering a model ensemble M consisting of $n_m$ members traversing the parameter space Σ" to "by considering a vector of model parameters M from the parameter space Σ"

**Eq. 6** *I think that some of these components are mislabelled. In particular, Eq. 6 is not Bayes' theorem, so I'm not sure it makes sense to refer to a posterior and prior as such. Rather, Eq. 6 is the definition of the 'posterior predictive distribution' (which is what the left-hand side should be labelled), which is the distribution of future sea level outcomes conditioned on data. This is decomposed into the two terms on the right side,*

$$P(M|B),$$

*which is what would usually be called the 'posterior distribution', 'parametric posterior distribution', or 'calibration' (as it already is in the paper) to disambiguate, and*

$$P(SLR|M, B),$$

*which is the distribution of model predictions given a particular parameter value (perhaps called 'model prediction' or 'projection').*

Yes you are right, we will change the terms to "Posterior prediction", "Model Prediction" and "Calibration".

**Eq. 7** *An important condition here is that*

$$Mi \sim P(M),$$

*which is to say that the realizations of the particles need to be drawn from the prior distribution. If that's not the case, then the prior needs to appear in the numerator and denominator of the term in L. 237.*

Thanks for the comment, we will change the sentence before the equation and add an equation to define P(M), as suggested by the second reviewer.

**L240** *'gaussian' should be capitalized.*

Correct

**Eq. 8** *This intersection notation is weird – I think it would be better to just start with Eq. 9.*

Agreed, we will remove this line.

***Eq.11*** *A should be Anobs or a new constant should be used.*

Agreed, we use C instead of A.

***L248*** *Be explicit as to what this measurement operator is, e.g. the evaluation of the Elmer/Ice FEM basis representation of the velocity field at the desired locations in space and/or time.*

We will add this information.

***L255*** *It is also worth noting that even if observational uncertainty were IID, model error definitely is not, which is what ultimately leads to the egregiously peaked distributions over ensemble members and heavily weighting only a single one. Ultimately the problem is that – priors aside – we don't know an appropriate likelihood function for comparing models to data! As such, the more ad hoc methodology described in this work is justified.*

Thanks for the comment, we will write two sentences at the end of this paragraph to add this information.

***L265–282*** *This section is really great. It has significant similarities to lots of previous work on Bayesian inference in the presence of model misspecification, and it might be worthwhile to frame the discussion in terms of that. This is a good reference:* [https://doi.org/10.1146/annurev-statistics-040522-015915](https://doi.org/10.1146/annurev-statistics-040522-015915)

Thanks for the reference, but as the article is very generic, we don't see where we can quote it without having to modify the discussion too much.

***L280 –282*** *I'm not sure I understand this statement.*

We will try to reformulate the sentence to clarify it.

***L304*** *It would be worth describing whether this weighting scheme is more or less permissive than full-period weighting - I don't have a good intuition. It might also be helpful to introduce an equation for each of the weighting schemes to be concrete.*

On the sub-period weighting, we will add sentences at the end of the paragraph and some details to precise the number of RMSEs used for each weighting.

***L308*** *I don't really understand the introduction of fparam weighting. This is very much tied to the particular parameterization and behavior of the authors' existing model setup (thoroughly described in a separate paper) and it's challenging for someone not so involved with that work to understand what this specific experiment is trying to capture. Can this be expanded to provide more substantial justification?*

We will reverse the position of the 2 paragraphs of this sub-section to start with the justifications and add some justification of the use of this weighting.

***L332*** *This sentence changes from passive to active voice in the middle. Probably best to stick with active voice.*

Agreed, we will change it.

***Sec.2.3.3*** *While I appreciate the desire to include SSP as a random variable, I also think that doing so sort of obscures the influence of all the other aspects of the model since this is expected (and*

*turns out to be) a dominant factor in determining predictive variance. Is it possible in later plots to also present ensemble ranges conditioned on SSP (i.e. the sub-ensembles of particles using just SSP2.6, SSP4.5, SSP8.5)? That would be helpful for comparison against existing similar work and would also facilitate a 'best-case versus worst-case' analysis for climate change impacts.*

We will add in the supplementary the results conditioned on SSP for the prior ensemble and the weighting. In the main text, we will also add a figure and a paragraph to discuss the effect of weighting on SSPs sub-ensembles.

**L343** *It would be super helpful to reiterate here what the difference is between the Ppr and the Cpr. I had initially thought to suggest more instructive names, but I can't think of any, so at the very least a brief reminder of the assumptions of each would be great.*

Agreed, we will add sentences to reiterate the difference between the 2 ensembles.

**L356** *Is the agreement between the Hpr median and mass loss observations by design or a happy accident? Fig. 2 and 3 Perhaps consider changing the symbology scheme to something friendly for greyscale/colorblindness, e.g. cross-hatching one of the two shaded regions.*

It is not by design, and therefore a coincidence. For the symbology, we will add hatchs for Cpr and Ppr. For consistency, we will also add hatching to all other figures. Most of the figures will be modified for greater clarity and legibility.

**L361** *It might be worth contextualizing this with respect to Robel, 2019:*
*https://doi.org/10.1073/pnas.190482211. The skew in the distribution is perhaps not surprising.*

We will add a reference to this paper at the end of the sentence: "[…], which is similar to other results in glaciology (e.g. Robel et al. (2019))."

**Sec. 3.3.1** *It's a little cumbersome to start a section with a reference to another section. I understand shunting the methodological details to the appendix, but a recapitulation of the methodological approach would be helpful here.*

Agreed, we will start the section with a summary of the cross-validation method previously introduced.

**Sec. 3.3.1** *More generally, all four points introduced here seem a bit ad hoc. Do there exist references that could help place the current procedure on more sound probabilistic footings? Seems like this problem has to have been studied before.*

We agree that the four points are a bit ad hoc, but it is because this section just summarize our results. We discuss this in section 4.4.1 Use of Bayesian Calibration, where we show that our results are similar to those of Jiang and Forssén (2022).

**L446** *Where is factor mapping previously established?*

It is introduced in section 2.1.1. We will add a reference to this section.

**L448** *The parameter flaw sometimes appears throughout the manuscript as just law. Please revise for consistency.*

Thanks, we will change all terms to $f_{law}$.

***Fig. 5*** *The overlying transparent bars aren't really readable.*

As in Figures 2 and 3, we will add hatch patterns.

***Sec 3.4.1*** *I think that this section would benefit from a bit of extra subdivision. I think it would be helpful to break this into individual subheadings describing the historical period and the forecasts. I think it would also be helpful to separate the principal conclusions about the relative insensitivity of long term forecasts to ISM parameters from the details of weighting. I also don't think that it makes sense to refer to the changes in ranges described around L506 as 'notable' – the more notable thing is that they're almost exactly the same!*

Agreed, we will add sub-headings to help the reader distinguish between the Hindcast ensemble, the Control ensemble, and the Predicted ensemble. We will also change the term 'notable' to 'little'.

***Fig. 6*** *The font in this figure is too small.*

Agreed, we will changed it.

***Sec 3.4.2*** *Again, I would like to reiterate that presenting ensemble results which each of the SSPs held fixed would be useful here, and would help to ameliorate some of the challenges associated with trying to guess the probabilities of future human carbon emissions (which is why previous works have treated these as fixed hypotheses rather than as random variables).*

As mentioned in a previous comment, we have added a figure and a paragraph to demonstrate the effect of weighting when separating the different SSPs.

***Sec. 4.1*** *I am not quite sure I understand the reference to ISMIP6 here. How is that relevant to the present model being able to reproduce observations?*

These papers explain the reason behind the use of a control run in ISMIP6. We will rewrite the sentence.

***Sec. 4.2*** *Again, I think that this section would be clarified by adding some more sub-headings. e.g. at L589, this paragraph could be called 'reducing uncertainty through ISM calibration), whereas at L598, this could be called 'reducing uncertainty through climate forcing calibration', or something like that.*

Agreed, we will added the suggested sub-headings.

***L576*** *The referenced compensatory effect is not clear to me from Fig. 4 or elsewhere. Could this please be clarified?*

We will add a figure in the supplement to illustrate this compensatory effect, and we will change the sentence.

***L585*** *If the front parameterization has such as significant influence, then perhaps this calls into question the validity of imposing the front at all. Would it be possible to make a statement about how the predictions might be influenced if the front were allowed to evolve freely or based on a different parameterization?*

We agree that imposing the front using this parameterisation is not the optimal method for projecting calving in Greenland tidewater glaciers. We believe our paragraph on this specific issue ('In the context of front retreat parameterisation, [...]') demonstrates the limitations of this approach

and highlights future developments needed in the ice sheet modeling community. For Elmer/Ice, implementing calving laws is currently under development but not yet available.

*L602–604 This is a very significant assertion that would have major implications for how ice sheet modeling proceeds in the future! What should the community be doing if improving ice dynamics isn't likely to help? (note that I don't disagree with the assertion – I am genuinely curious where effort should be allocated instead).*

We want to clarify that this assertion is mainly valid for Greenland ice sheet (we will add a mention to Greenland ice sheet in this sentence), and may differ significantly for Antarctica. Besides this, we believe there are numerous challenges to address:

- firstly, we demonstrate here that front parameterisation has a significant influence. Part of this influence stems from ocean processes (such as the transport of warm water into the fjord and melting at the front), but another part is due to uncertainties surrounding calving itself. This aligns with the previous comment: developing and validating calving laws for our Elmer/Ice ice sheet configuration appears to be a priority;
- secondly, in my opinion, we need to better account for uncertainties associated with the bed. Here, we have neglected this uncertainty, as in our previous article, but tests conducted in the previous article indicate a strong influence of this uncertainty on the ice discharge obtained. This factor should also play a crucial role in calving, as the stabilization of the front at a given point heavily depends on the bed. Since Bayesian calibration cannot be applied to this bed calibration, other transient data assimilation techniques will need to be used (e.g., Gillet-Chaulet, 2020);
- thirdly, it is possible that phenomena currently unknown could alter future outcomes. New discoveries often lead to higher predictions of mass loss;

We will add a mention that this assertion concerns the Greenland ice sheet and that some uncertainties have not been explored. We will add the discussion paragraph of this front parameterisation directly after this paragraph on ice dynamics.

*L618 Its foundation in observational data is sort of the problem – no data available in the future.*

Partially agreed, as this statistical downscaling has been partially validated through cross-validation: a training set composed of some glaciers and a validation set with the remaining glaciers.

*L703 There are other possibilities than the Gaussian or T.*

Agreed, we will rewrite the sentence.

*Sec. 4.4 There is a lot of good in this section, but there is also a lot of material that is only applicable to the authors' own modeling setup (which has already been covered) it might be worthwhile to take a critical read to assess which of these insights are going to be generally applicable, and which are more like notes to guide the authors' own continuing work.*

We will add two sub-headings: the first one to highlight insights in terms of Bayesian calibration ('Use of Bayesian Calibration'), which will interest anyone using such data assimilation techniques; and the second one to emphasize insights in ice sheet modeling, particularly focusing on friction ('Friction Law'). We agree that the first paragraph mainly concerns ISMs using data assimilation to calibrate friction fields like ours, but other ISMs using similar techniques may also find these results valuable. Regarding the second paragraph, we also agree that the parameterization mainly applies to our model. Therefore, we will try to rephrase the beginning of this paragraph to emphasize that it is crucial to consider sub-hydrology, at least in a parameterized manner as demonstrated here.

***Sec. C2*** *The student-t distribution has an additional degree of freedom, namely the number of degrees of freedom. What was used for this, or how was it estimated?*

As in Aschwanden and Brinkerhoff 2022, we used 2 degrees of freedom. We will include this information in this section C2.

---

## Author Response (AR1)

Original reviewers comments are in *italic and black*, our answers are in blue. We have prepared a new version of the manuscript taking into account the reviewers' comments, as well as a pdf outlining the changes.

**Reviewer 1 (Brinkerhoff, Douglas)**

**1 Summary and Main Points**

*This paper presents a probabilistic forecast for mass change at Upernavik Ice Stream in West Greenland. In very broad terms the forecast is produced by running an ensemble of predictions over different model parameters using the ice sheet model Elmer/Ice, and then weighting those ensemble members based on their agreement with a variety of observations. The paper is remarkable in that performs a very detailed methodological exploration of different schemes for weighting ensemble members. The paper also presents an index based sensitivity analysis, allowing for an interesting temporal discussion of the influence of variance in model parameters on the variance in predictions.*

*Overall, I think that this paper is exceptional and represents an important advance in data-constrained forecasting for Greenland. In particular, the paper makes (and justifies) a rather significant claim, which is that at centennial scales, the dominant source of uncertainty for projection remains elements of climate forcing and that contemporary observational constraint on ice sheet models exhibits diminishing returns. I have no major methodological issues with this work and I find the science to be sound. I do think that the paper could be made more accessible to a broad audience – in particular one that isn't complete versed in the language of probabilistic forecasting, and for whom this paper would still be a very useful read – by some clarification in exposition. In particular, I think that the use of acronyms could be reduced, some sections could be shortened, and the lanauage of Bayesian inference modified to be in closer accordance with standard usage.*

We thank Doug Brinkerhoff for his very positive comments. It is truly gratifying to receive such positive feedback about our paper. As suggested, we add clarifications to make the paper more accessible to a broader audience. In particular, we include figures that summarize our methodology, which help to streamline some sections.

**2 Line-by-line comments**

**L23** *'limited' used twice.*

Thanks, we have changed 'is the limited ability' to 'stems from the constrained ability'.

**L24** *What is the difference between 'limited understanding of ice dynamics' and 'uncertainties in ISMs'?*

Yes it is quite similar, so we changed "ice dynamics" by "initial state".

**L30** *It's worth noting that in Aschwanden (2021) the authors state that ISMIP6 actually already does a really good job with quantifying uncertainty in model structure! For the other considerations, yes, those must still be better accounted for.*

Yes, we have change the sentence to "First, although ISMIP6 quantifies uncertainty in model structure, the intrinsic uncertainties associated with model parameters, as well as initial and boundary conditions, must be more thoroughly accounted for".

**L34** *This would be an appropriate reference for sensitivity studies in Greenland as well: https://doi.org/10.5194/tc-13-1349-2019*

Thanks for the reference. It seems that this paper is on Antarctica, so we added it along with the reference to Hill et al.

**L47** *For item (i), I think it makes sense to characterize this as 'establishing prior distributions' over uncertaint model parameters.*

Agree, we have changed the sentence to : "These studies typically involve two steps: (i) establishing prior distributions over uncertain model parameters to obtain an ensemble and projecting it into the future to forecast a prior future SLR contribution, and (ii) adjusting prior distributions by giving weights to the members according to their ability to reproduce past observations."

**L50** *It might be worth noting that this lack of cross-validation is often done because there's just not that much data to work with usually, but the authors' point generally a very fair one. I hope that future studies incorporate the authors' suggested cross-validation framework.*

We partially agree with this comment. Despite the limited data availability, we maintain that dividing the dataset into calibration and validation subsets is always feasible and should not detract from the robustness of the results. However, in scenarios where there are no significant variations, such as unchanged ice dynamics during the validation period, this approach might not significantly enhance the robustness of the findings. To address the concern of data scarcity, we have prefaced the sentence with 'due to the limited availability of observational data'.

**L111** *The specification of the calving front position rather than it being a prediction from the model was and remains one of the most contentious aspects of the ISMIP6 experiments. While the position of the calving front may be specified with precision by the parameterization, that doesn't necessarily mean that the position will be accurate in the future, and successfully simulating calving rates and front positions remains one of the largest open challenges in glaciology. It is worth noting here that by ignoring this source of uncertainty in model projections, the authors' are making a large and potentially critical assumption. I don't think it's a problem, but it really does need to be discussed.*

The sentence to which you refer is for the historical period, and so the position comes from observations whose uncertainties are unknown. However, as described below for the future (Ppr), this uncertainty is taken into account (as in ISMIP6) by adjusting the sensitivity parameter of the parameterisation. This choice of parameterisation is also discussed in section 4.2.2 where we agree that it remains a major challenge.

**2.1.1–2.1.3** *I struggle a bit with the semantic separation of the historic ensemble (hpr) from the two future ensembles (cpr and ppr), given that it is the initial period for both. I don't think it's too important but the authors for clarity may wish to refer to the historic ensemble as the 'shared hindcast period' or similar for the two prognostic ensembles.*

We have change the term Hpr to that you suggest. To clarify this point, we will also add a figure to summarise our framework.

***L141–150** I find this section to be a little bit confusing. It might help to have a more complete discription of the parameterization and specifically the role of κ.*

We have added the following description of the Slater parameterisation: "For the future position of the front, we used the parameterisation employed in ISMIP6 (Slater et al., 2019, 2020) depending on a constant scalar parameter $\kappa$ and defined as:

$$\Delta L = \kappa \Delta(Q^{0.4} \times TF) \quad (1)$$

where Q denotes the mean summer (June–July–August) subglacial runoff (in $m^3.s^{-1}$) from the RCM, and TF represents the ocean thermal forcing (in °C) outside of the fjord from the AOGCM. This parameterisation is contingent upon the front sensitivity $\kappa$, the distribution of which has been determined through the calibration of this relationship for each glacier from 1960 to 2018 across each GrIS sector (Slater et al., 2019). The distribution of $\kappa$ effectively encapsulates the uncertainties arising from several critical parameters, e.g. calving rates, thermal transport into the fjord."

***L195** 'Ice Discharge' should be lower-cased.*

Thanks, corrected

***L197** Given that this producet is based on a model result (RACMO), is there the potential for this product to contain systematic bias?*

Yes, it could, even though we have shown in our previous paper that members using RACMO outputs lead to less bias in surface reconstruction than those using MAR (Jager et al. 2024). Potential bias of this input-output method can be find in Mouginot et al. (2019), so we have not change the text.

***L203** 'ensemblist' → ensemble*

Thanks, corrected

***L225** I think that M needs to be understood as a random vector of model parameters, with P (M) its prior distribution. It will be the case that a sample will be drawn from this distribution, which will be used to create the ensemble, but M is not the ensemble itself.*

Agreed, we have changed the sentence to "We adopt the formalism introduced by Brinkerhoff (2022), which updates a model prediction by considering a vector of model parameters M from the parameter space Σ, a collection of untraversed model assumptions H, the evolution of external forcings F, and a set of observations B."

***Eq. 6** I think that some of these components are mislabelled. In particular, Eq. 6 is not Bayes' theorem, so I'm not sure it makes sense to refer to a posterior and prior as such. Rather, Eq. 6 is the definition of the 'posterior predictive distribution' (which is what the left-hand side should be labelled), which is the distribution of future sea level outcomes conditioned on data. This is decomposed into the two terms on the right side,*

$$P(M|B),$$

*which is what would usually be called the 'posterior distribution', 'parametric posterior distribution', or 'calibration' (as it already is in the paper) to disambiguate, and*

$$P(SLR|M, B),$$

*which is the distribution of model predictions given a particular parameter value (perhaps called 'model prediction' or 'projection').*

Yes you are right, we have changed the terms to "Posterior prediction", "Model Prediction" and "Calibration".

**Eq. 7** *An important condition here is that*

$$Mi \sim P\,(M),$$

*which is to say that the realizations of the particles need to be drawn from the prior distribution. If that's not the case, then the prior needs to appear in the numerator and denominator of the term in L. 237.*

Thanks for the comment, we've changed the sentence before the equation from "with particles Mi corresponding to different members" to "which use an ensemble of particles Mi, corresponding to different members, to approximate the prior probability P(M) by :" and an equation to define P(M).

**L240** *'gaussian' should be capitalized.*

Corrected

**Eq. 8** *This intersection notation is weird – I think it would be better to just start with Eq. 9.*

Agreed, we have removed this line.

**Eq.11** *A should be Anobs or a new constant should be used.*

Agreed, we use C instead of A

**L248** *Be explicit as to what this measurement operator is, e.g. the evaluation of the Elmer/Ice FEM basis representation of the velocity field at the desired locations in space and/or time.*

We have changed the sentence from "$H(M_i)$ is the measurement operator corresponding to $Q_{m,i}$ in Eq. 5, which projects the state of the model onto observation $b_j$" to "$H(Mi)$ is the observation operator corresponding to $Q_{m,i}$ in Eq. 6, which project the model results onto the observation regular grid using the natural finite element interpolation functions of Elmer/Ice."

**L255** *It is also worth noting that even if observational uncertainty were IID, model error definitely is not, which is what ultimately leads to the egregiously peaked distributions over ensemble members and heavily weighting only a single one. Ultimately the problem is that – priors aside – we don't know an appropriate likelihood function for comparing models to data! As such, the more ad hoc methodology described in this work is justified.*

Thanks for the comment, we have added two sentences at the end of this paragraph: "Thirdly, even supposing observational uncertainties were independent and identically distributed, it is clear that the modelling errors are not. Ultimately, the crux of the matter lies in our lack of a suitable likelihood function for effective model-data comparison."

**L265–282** *This section is really great. It has significant similarities to lots of previous work on Bayesian inference in the presence of model misspecification, and it might be worthwhile to frame the discussion in terms of that. This is a good reference: https://doi.org/10.1146/annurev-statistics-040522-015915*

Thanks for the reference, but as the article is very generic, I don't see where I can quote it without having to modify the discussion too much.

*L280 –282 I'm not sure I understand this statement.*

We tried to reformulate the sentence; I hope it is better: "This assumption underpins the weighting methodology adopted in earlier studies which employ a singular performance metric; for instance, in references Pollard et al. (2016) and Albrecht et al. (2020), the median of such a performance metric is utilised as an estimate for σ.''

*L304 It would be worth describing whether this weighting scheme is more or less permissive than full-period weighting - I don't have a good intuition. It might also be helpful to introduce an equation for each of the weighting schemes to be concrete.*

On the sub-period weighting, we add the following sentence at the end of the paragraph: "Because this weighting involves more RMSEs than the full-period weighting (8 versus 4), it leads to a narrower posterior distribution.". For the equation, it is always the same equation 14, so I have not written this equation but I add some details to precise the number of RMSEs used. For the Full-period weighting: " with $n_s$ the number of sub-basins, i.e. $n_s$=3 for the cross-validation and $n_s$=4 for posterior ensemble"; For the Sub-period weighting: " with this time $n_s$ the total number of periods, i.e. 8 for posterior ensemble (3 for UI-N and UI-C, 1 for UI-S and UI-SS)."

*L308 I don't really understand the introduction of fparam weighting. This is very much tied to the particular parameterization and behavior of the authors' existing model setup (thoroughly described in a separate paper) and it's challenging for someone not so involved with that work to understand what this specific experiment is trying to capture. Can this be expanded to provide more substantial justification?*

We have reversed the position of the 2 paragraphs of this sub-section to start with the justifications.

We have also add the following sentence: "Indeed, in most of the large-scale applications of Elmer/Ice (e.g. Goelzer et al. (2018); Seroussi et al. (2020); Hill et al. (2023)), friction is considered to be constant over time with no dependence on subglacial hydrology. The parameterisation developed in this previous study addresses this limitation of Elmer/Ice, which should be more robust using this parameterisation. […]  and (ii) to compare this weighting with the other weighting to see if they are able to highlight this characteristic without going into as much detail as this previous study.

*L332 This sentence changes from passive to active voice in the middle. Probably best to stick with active voice.*

Agreed, we have changed the sentence to "We base this weighting approach on the challenges we face in achieving SSP5-8.5 under current policies (Intergovernmental Panel on Climate Change (IPCC), 2022), which leads us to assign more weight to SSP2-4.5"

*Sec.2.3.3 While I appreciate the desire to include SSP as a random variable, I also think that doing so sort of obscures the influence of all the other aspects of the model since this is expected (and turns out to be) a dominant factor in determining predictive variance. Is it possible in later plots to also present ensemble ranges conditioned on SSP (i.e. the sub-ensembles of particles using just SSP2.6, SSP4.5, SSP8.5)? That would be helpful for comparison against existing similar work and would also facilitate a 'best-case versus worst-case' analysis for climate change impacts.*

We have added in the supplementary results conditioned on SSP for the prior ensemble and the weighting (Figs. S1 and S2). We have added a sentence to mention it at the end of the first paragraph of 3.1: "It should be noted that the results obtained by the various SSPs are mixed, and that readers wishing to see the results by SSP can find them in the supplement (figure S1)." We have also added a figure in the main text for the Sub-period weighting (Figure 8) and a paragraph to briefly discuss this figure:

"An alternative method to assess the influence of Bayesian calibration with reduced SSP-related uncertainty entails presenting results for each distinct SSP (Fig. 8). This approach reveals effects that the aggregation of SSPs otherwise conceals. For SSP2-4.5, the application of Sub-period weighting significantly tightens the 95% confidence interval across short, medium, and long-term projections. Concerning SSP1-2.6 and SSP5-8.5, the reduction in uncertainty is less pronounced, not mirroring the levels seen in previous studies on Greenland ice sheet (e.g. Aschwanden and Brinkerhoff (2022)). This modest reduction is attributed to the robustness of our model, with a prior close to observations. Nonetheless, a notable shift towards higher probability values is observed for each SSP, as shown by histograms, boxplots, and median values of figure 8. Similar results for the Full-period and $f_{param}$ weightings are illustrated in the supplement (fig. S2)."

**L343** *It would be super helpful to reiterate here what the difference is between the Ppr and the Cpr. I had initially thought to suggest more instructive names, but I can't think of any, so at the very least a brief reminder of the assumptions of each would be great.*

Agreed, we have added the following sentences: "Hpr simulates the historical evolution of UI from 1985 to 2019 using forcings that most closely align with past observed conditions, including recorded front positions and SMB derived from RACMO forced by reanalysis data. Cpr projects the future evolution of UI from 2015 to 2100 under constant forcing conditions, maintaining the front position as observed in 2015 and using the averaged SMB spanning 1960-1990. Conversely, Ppr forecasts UI changes from 2015 to 2100 under evolving forcing conditions as outlined in the ISMIP6 framework; this involves a parameterized front position based on Slater et al. (2019, 2020) and SMB that is downscaled using a RCM from an AOGCM."

**L356** *Is the agreement between the Hpr median and mass loss observations by design or a happy accident? Fig. 2 and 3 Perhaps consider changing the symbology scheme to something friendly for greyscale/colorblindness, e.g. cross-hatching one of the two shaded regions.*

It is not by design, and therefore a coincidence. For the symbology, we have added hatchs for Cpr and Ppr. For consistency, we have also added hatching to all other figures. Most of the figures have been modified for greater clarity and legibility.

**L361** *It might be worth contextualizing this with respect to Robel, 2019: https://doi.org/10.1073/pnas.190482211. The skew in the distribution is perhaps not surprising.*

We add a reference to this paper at the end of the sentence: "[…], which is similar to other results in glaciology (e.g. Robel et al. (2019))."

**Sec. 3.3.1** *It's a little cumbersome to start a section with a reference to another section. I understand shunting the methodological details to the appendix, but a recapitulation of the methodological approach would be helpful here.*

Agreed, we start now the section with a summary of the cross-validation method previously introduced:
"In this section, we outline the results of our cross-validation process, which evaluates the effectiveness of various weighting methodologies designed to prevent overfitting and ensure the reliability of our new ensemble. We initially compute weights using three sub-catchments of UI,

testing methods such as Full-period, Sub-period, and $f_{\text{param}}$ weighting (detailed in Section \ref{MethodEvelWeighting}). For the Full-period approach, we examine the impact of different probability density functions (Gaussian or Student's-t), $\sigma$ estimates (minimum, median, mean, or maximum of the RMSE distribution), and calibration data types (surface elevations, velocities, ice discharge).

We then assess ensemble performance using the Continuous Ranked Probability Score (CRPS, Eq. \ref{CRPS}) on a validation set comprising the remaining sub-catchment. The CRPS is applied to various observed fields, including velocity, surface elevation, and ice discharge metrics.

The analysis of weighting sensitivity unveiled the following key findings (further details are given in Appendix C):"

**Sec. 3.3.1** *More generally, all four points introduced here seem a bit ad hoc. Do there exist references that could help place the current procedure on more sound probabilistic footings? Seems like this problem has to have been studied before.*

We agree that the four points are a bit ad hoc, but it is because this section just summarizes our results. We discuss them in section 4.4.1 Use of bayesian calibration, where we show that our results are similar to the those of Jiang and Forssén (2022).

**L446** *Where is factor mapping previously established?*

It is introduced in section 2.1.1. We add a reference to this section.

**L448** *The parameter flaw sometimes appears throughout the manuscript as just law. Please revise for consistency.*

Thanks, we have changed all terms to $f_{\text{law}}$.

**Fig. 5** *The overlying transparent bars aren't really readable.*

As in Figures 2 and 3, we have added hatch patterns.

**Sec 3.4.1** *I think that this section would benefit from a bit of extra subdivision. I think it would be helpful to break this into individual subheadings describing the historical period and the forecasts. I think it would also be helpful to separate the principal conclusions about the relative insensitivity of long term forecasts to ISM parameters from the details of weighting. I also don't think that it makes sense to refer to the changes in ranges described around L506 as 'notable' – the more notable thing is that they're almost exactly the same!*

Agreed, we have added sub-headings to help the reader distinguish between the Hindcast ensemble, the Control ensemble, and the Predicted ensemble. We have also changed the term 'notable' to 'little'.

**Fig. 6** *The font in this figure is too small.*

Agreed, we have changed it. Figures 8 and D1 will be bigger in two-column format.

**Sec 3.4.2** *Again, I would like to reiterate that presenting ensemble results which each of the SSPs held fixed would be useful here, and would help to ameliorate some of the challenges associated with trying to guess the probabilities of future human carbon emissions (which is why previous works have treated these as fixed hypotheses rather than as random variables).*

As mentioned in a previous comment, we have added a figure and a paragraph to demonstrate the effect of weighting when separating the different SSPs.

***Sec. 4.1*** *I am not quite sure I understand the reference to ISMIP6 here. How is that relevant to the present model being able to reproduce observations?*

These papers explain the reason behind the use of a control run in ISMIP6. We rewrite the sentence to: "One of the reasons behind the use of a control run in ISMIP6 was to address the limitations of the models in accurately reproducing recently observed changes of the ice sheets due to artificial model drift, thus making it easier to assess the deviation of each projection from this drift (Goelzer et al., 2020; Seroussi et al., 2020; Nowicki et al., 2020)"

***Sec. 4.2*** *Again, I think that this section would be clarified by adding some more sub-headings. e.g. at L589, this paragraph could be called 'reducing uncertainty through ISM calibration), whereas at L598, this could be called 'reducing uncertainty through climate forcing calibration', or something like that.*

Agreed, we have added the suggested sub-headings.

***L576*** *The referenced compensatory effect is not clear to me from Fig. 4 or elsewhere. Could this please be clarified?*

We have added a figure in the supplement to illustrate this compensatory effect, and we have changed the sentence to: "This lack of influence can be attributed to a compensatory effect: the AOGCM exerts a non-zero influence on both the ice discharge and the SMB as depicted in Figure 4. Nevertheless, AOGCMs with higher discharge rates are associated with lower SMB, and vice versa, culminating in a comparable net ice mass change across different AOGCMs (Fig. S1)"

***L585*** *If the front parameterization has such as significant influence, then perhaps this calls into question the validity of imposing the front at all. Would it be possible to make a statement about how the predictions might be influenced if the front were allowed to evolve freely or based on a different parameterization?*

We agree that imposing the front using this parameterisation is not the optimal method for projecting calving in Greenland tidewater glaciers. We believe our paragraph on this specific issue ('In the context of front retreat parameterization, [...]') demonstrates the limitations of this approach and highlights future developments needed in the ice sheet modeling community. For Elmer/Ice, implementing calving laws is currently under development but not yet available.

***L602–604*** *This is a very significant assertion that would have major implications for how ice sheet modeling proceeds in the future! What should the community be doing if improving ice dynamics isn't likely to help? (note that I don't disagree with the assertion – I am genuinely curious where effort should be allocated instead).*

We want to clarify that this assertion is mainly valid for Greenland ice sheet (we will add a mention to Greenland ice sheet in this sentence), and may differ significantly for Antarctica. Besides this, we believe there are numerous challenges to address:
- firstly, we demonstrate here that front parameterisation has a significant influence. Part of this influence stems from ocean processes (such as the transport of warm water into the fjord and melting at the front), but another part is due to uncertainties surrounding calving itself. This aligns with the previous comment: developing and validating calving laws for our Elmer/Ice ice sheet configuration appears to be a priority;

- secondly, in my opinion, we need to better account for uncertainties associated with the bed. Here, we have neglected this uncertainty, as in our previous article, but tests conducted in the previous article indicate a strong influence of this uncertainty on the ice discharge obtained. This factor should also play a crucial role in calving, as the stabilization of the front at a given point heavily depends on the bed. Since Bayesian calibration cannot be applied to this bed calibration, other transient data assimilation techniques will need to be used (e.g., Gillet-Chaulet, 2020);
- thirdly, it is possible that phenomena currently unknown could alter future outcomes. New discoveries often lead to higher predictions of mass loss;

We have added a mention that this assertion concerns the Greenland ice sheet and that some uncertainties have not been explored. We add the discussion paragraph of this front parameterisation directly after this paragraph on ice dynamics.

**L618** *Its foundation in observational data is sort of the problem – no data available in the future.*

Partially agreed, as this statistical downscaling has been partially validated through cross-validation: a training set composed of some glaciers and a validation set with the remaining glaciers.

**L703** *There are other possibilities than the Gaussian or T.*

Agreed, we have rewritten the sentence from "Opting for an overly narrow distribution or favoring a Gaussian distribution over a Student distribution can result in overfitting, wherein only a few high-performing members are emphasized, thereby disregarding crucial information." to "Opting for an overly narrow distribution or favoring a distribution with very thin tails as the Gaussian over a distribution with fatter tails as the Student's t can result in overfitting, wherein only a few high-performing members are emphasized, thereby disregarding crucial information."

**Sec. 4.4** *There is a lot of good in this section, but there is also a lot of material that is only applicable to the authors' own modeling setup (which has already been covered) it might be worthwhile to take a critical read to assess which of these insights are going to be generally applicable, and which are more like notes to guide the authors' own continuing work.*

We have added two sub-headings: the first one to highlight insights in terms of Bayesian calibration ('Use of Bayesian Calibration'), which will interest anyone using such data assimilation techniques; and the second one to emphasize insights in ice sheet modeling, particularly focusing on friction ('Friction Law'). We agree that the first paragraph mainly concerns ISMs using data assimilation to calibrate friction fields like ours, but other ISMs using similar techniques may also find these results valuable. Regarding the second paragraph, We also agree that the parameterization mainly applies to our model. Therefore, We tried to rephrase the beginning of this paragraph to emphasize that it is crucial to consider sub-hydrology, at least in a parameterized manner as demonstrated here.

We changed this "Regarding the weighting approach that assigns higher weights to ensemble members utilizing a friction law that accounts for the sub-hydrology effect in a parameterised manner, our findings indicated a somewhat reduced overall performance compared to the Full-period weighting approach. However, it is noteworthy that the outcomes obtained through this approach align with the expectations outlined in \cite{Jager_2024}. This prior study suggested that our parameterisation would likely lead to increased mass loss."
to this: "With regard to the law of friction, our findings emphasize that incorporating the sub-hydrological effect—albeit in a parameterized manner as demonstrated in this study—is crucial for accurately simulating the historical dynamics of the UI. Additionally, inclusion of this effect is associated with an amplification of projected future mass loss of the glacier, consistent with

expectations outlined in \cite{Jager_2024}. In that study, we highlighted the significance of the sub-hydrological effect and posited that its consideration would likely exacerbate glacial mass loss."

**Sec. C2** *The student-t distribution has an additional degree of freedom, namely the number of degrees of freedom. What was used for this, or how was it estimated?*

As in Aschwanden and Brinkerhoff 2022, we used 2 degrees of freedom. We've included this information in this section C2.

**Reviewer 2 (Anonym)**

**1 Summary and Main Points**

*In the paper 'The future of Upernavik Isstrøm through ISMIP6 framework: Sensitivity analysis and Bayesian calibration of ensemble prediction', Jager and co-authors study several aspects associated with the evolution of the Upernavik Isstrøm Glacier, Greenland. Based on a statistical framework, numerical results obtained with the Elmer/Ice finite-element code, and observational data, they quantify the uncertainties associated with predictions of the future sea-level contribution. They improve the robustness of their analysis by considering a cross-validation step and by studying several weighting methods for assigning a likelihood score to the uncertain parameters.*

*I think that the paper will make a great addition to the scientific literature as it deals with an important topic, namely the quantification of uncertainties, and, more generally, the study of the methods that are used to produce such analyses. Nonetheless, I have a series of comments that I would like the authors to address prior to the publication of the manuscript. As described hereafter, those are mainly related to the form of the paper, rather than its scientific content.*

**2 General**

*My main comment is related to the way the paper is written. I have found the methods and results to be particularly interesting, but the style of the paper makes it quite difficult to grasp them efficiently. My main complaints are that the whole paper is very long (45 pages), that some parts are difficult to follow because of the lack of visual data, and that some subsections are particularly long. I would suggest the following changes:*

*• Streamlining the manuscript, in particular by focusing on the key points in each paragraph, and removing unnecessary repetitions.*

*• Focusing on the novel aspects of your study. To my understanding, these are the cross-validation (which I believe has not really been done previously in a glaciological context), and the use of different weighing methods.*

*• Adding figures/tables/schematics that allow to understand the content of the text in a visual and summarized way. For example, in Section 2, you could create a table with the different scenarios (Hpr, Cpr, Ppr), and for each of these scenarios specify the SMB that is used, as well as the front position and the uncertain parameters. For the observational data, you could create a table with the different types of observations that you have, their type, and where they come from.*

*• I wonder if the fparam weighing makes much sense, giving that this parameter is one of the uncertain parameters that are calibrated in the Bayesian process. Note that a classical way to favor specific values of fparam , given your knowledge of its importance, would be to change its prior distribution.*

*To reiterate, I find the paper to be both useful and significant. But I still think that it is important to improve its style, as it will help the audience to better understand the key points presented in the manuscript.*

We thank the anonymous reviewer for their constructive review and positive comments. To improve the readability of the overall paper, we have made the following changes:

• We have tried to clean up text that is redundant or heavy to read.

• We have changed the structure of the headings in the results section to better highlight the two main parts of our study: the sensitivity analysis and the Bayesian calibration. We still think that these two parts are novel in our study, not only the novelties of the Bayesian calibration. No other study has explored the uncertainties of the ISMIP6 framework as exhaustively as we do here, with 3 SSPs and various RCMs.

• We add two summarizing figures of the method. The first one describe the different ensembles of the study to show their differences in forcing and summarize the different elements taken into account for the sensitivity analysis. The second one summarize our methodology to produce robust Bayesian calibration. We hope this has helped the readers and also reduced the size of the text.

• We add some justification for the use of the fparam weighting. It still makes sense for us to use it because it allows us to see the effect of the parameterization developed in Jager et al. (2024) on the projections and shows that taking into account the effect of subglacial hydrology, at least in a parameterized way, significantly increases the projected mass loss of Upernavik Isstrøm. This also lets us evaluate this weighting against others to check if they can underscore the use of the parameterisation without the extensive detail used in the earlier study.

**3 Specific comments**

*(1) [Line 24] It is a bit unclear at this stage what distinguishes the 'limited understanding of ice dynamics ' and 'uncertainties in Ice Sheet Models'. Maybe specify what you mean for the latter, e.g., 'uncertainties in the parameters of Ice Sheet Models'.*

Yes it's quite similar, so we changed "ice dynamics" by "initial state".

*(2) [Line 34] A paper that is missing for Antarctica is Bulthuis et al. (2019).*

Agreed, we add it.

*(3) [Line 63] 'initialisation' → 'initialization' as you use American English in your manuscript. Also check Lines 97, 158, 340, 546, 756, 774, 775, 792, and 864.*

Thanks, we have corrected it.

*(4) [Line 65] The use of the active voice in this sentence is a bit weird here, given that the rest of the paragraph is written with the passive voice.*

Agreed, we have changed the sentence from «Additionally, we prescribed the front positions and Surface Mass Balance (SMB) for each year.» to «Additionally, the front positions and Surface Mass Balance (SMB) were prescribed for each year.».

*(5) [Line 168] I am guessing that the equation mentioned here should be Eq. 1, not Eq. 4.*

Thanks, we have corrected it.

*(6) [Line 168] It is a bit confusing that the sensitivity indices Si are called 'first-order sensitivity indices' here, and not before. I would suggest discussing why the Si are called 'first-order indices', or directly mentioning Line 161 that the indices that you introduce are of first order. Otherwise, the*

*reader might wonder which indices you are talking about in this paragraph, as it is not clear that you are talking about the Si indices.*

Agreed, we changed «the following indices» to «the first-order sensitivity indices» in the line 161 and changed «Accurately computing sensitivity indices usually requires [...]» to «Accurately computing sensitivity indices of an order greater than one usually requires [...]» in the line 165.

*(7) [Line 172] 'Y' needs to be written in italics (Y ) here.*

Thanks, we have corrected it.

*(8) [Equation 6] The first factor is not a prior distribution for the problem considered in the paper. Going back to Aschwanden and Brinkerhoff (2022), a possible name for this factor would be 'projection'.*

*The distinction between prior and posterior distributions (i.e., Bayes' theorem) appears later, implicitly, through the computation of the term P(M|B ) in equation (6). Specifically, Bayes' theorem writes*

$$P(M|B) = \frac{P(B|M)P(M)}{P(B)} = \frac{P(B|M)P(M)}{\int P(B|M)P(M)dM}$$

*where:*

*• P(M|B ) is the posterior distribution;*

*• P(B |M) is the likelihood distribution;*

*• P(M) is the prior distribution.*

*For all intents and purposes, you will find at the end of this review a few equations that show how, starting from (R1), I arrive at your equation (7).*

We had a similar comment from the other reviewer and have relabeled the two terms from "Prior" and "Posterior" to respectively "Model Prediction" and "Posterior Prediction."

*(9) [Line 214] Ideally, you should define every variable that appear in the equations, so $F_m^j$ , $F_o^j$ , $Q_{m,i}^j$ , ... should be defined. To save space, it makes sense no to do so, but please at least mention in this paragraph that the subscript i is associated with the i-th member of the ensemble and that the superscript j is associated with the j-th observation.*

Yes agreed, we add the following sentence: «The subscript i is associated with the i-th member of the ensemble and that the superscript j is associated with the j-th observation.»

*(10) [Line 216] I am guessing there is an 'it' missing before 'is common' here.*

Thanks, we have corrected it.

*(11) [Line 237] It really is a detail, but please avoid using fractions within the text. Instead, write the definition of wi as a full new equation, or write it as $w_i = P(B |M_i )/\sum_{k=1}^{n} P(B |M_k )$. Same comment for the factions that appear later on in the text.*

Agreed, we have changed it to a new full equation while we have changed the writing of the other ones.

*(12) [Equations (8)–(12)] I suggest removing equations (8)–(10), as these equations do not add much to the discussion, and might even appear unnecessarily technical. It seems to me that the reader should be able to deduce from the Gaussian and independence assumptions that P(B |Mi ) has the form shown in (11), which is quite standard.*

Agreed, we have removed these equations.

*(13) [Line 248] Technically, H is the measurement operator, not H(Mi ) (which is the value taken by this operator when M = Mi ).*

Agreed, we have changed this part of the sentence from «H(Mi) is the measurement operator corresponding to Qm,i» to «H is the measurement operator with H(Mi) corresponding to Qm,i»

*(14) [Line 266] To be consistent, write f (RMSE, σ), not just f (RMSE).*

We have deleted this paragraph as it duplicated the one below.

*(15) [Line 271] Please read again this paragraph, it seems that you repeat yourself.*

Agreed. As said in the previous comment, we have deleted the duplication.

*(16) [Subsection 2.3.1] Overall, I think that this subsection is not structured in an efficient way: you first present the equations (12) and (13), corresponding to the 'classical' approach. Reading the beginning of this section, it seemed to me that you are going to use those expressions. But then you discuss limitations (which always is a real plus), and consequently modify you formulas. It might make more sense to directly state that while expressions (12) and (13) are the usual approach, you are not going to use them, and instead will use the formula (14) instead. On a related note, it is a bit surprising that you mention Line 249 that σ is the standard deviation of the observation error (while it is common, as you mention later, to include the model error in it). So maybe directly state the difficulty associated with σ, and that your equation (14) is a possible solution for it.*

Thanks for the comment, we have restructured this section and rewriten partially some paragraphs for a smoother reading. We start now with the presentation of the different equations and mentionning than we will use equation 14. We then explain the limitation of equation 13 and how the equation 14 allows us to bypass these limits.

About σ, we have added some elements in the paragraph dedicated to it. We precise it corresponds to the standard deviation of the observation error in equation 13, but, in equation 14 it takes into account the structural error of the model as done in previous works (Murphy et al., 2009; Nias et al., 2019; Edwards et al., 2019).

*(17) [Line 294] I wonder if the discussion of the assumptions that must be examined should not be included in the 'full-period weighting' item, Line 300.*

Agreed, we have added it after the description of the full-period weighting.

*(18) [Line 382] I do not agree with the contradiction indicated by the 'On the contrary' here: the fact that the sum of the first-order Sobol indices is greater than one does not prevent a substantial impact of specific parameter combinations. Furthermore, the fact that the sum of the first-order Sobol indices is smaller or equal than one does guarantee that the inputs are independent.*

Agreed, it wasn't very clear that "on contrary" was there only to position the "smaller than 1". So, we changed the «On the contrary» to «otherwise».

*(19) [Line 462] law → flaw .*

Thanks, we have corrected it.

*(20) [Line 462] The fact that the priors and posteriors distributions are similar for several parameters is an important result. Maybe you could elaborate on that, both in terms of the interpretation that you give to this observation, and the conclusions that can be drawn for it.*

We mention it briefly in the appendix B3: «In hindsight, our initial choice of distribution for these three parameters proves to be suitable due to the absence of significant changes observed in their posterior distributions.». We agree that it's an important result for our future perspectives but remains restricted to our study. Not everyone will use the same range of parameters because it can be specific to our catchment area. Moreover, you may need different parameters if you are not studying a tidewater glacier, you use an other ISM or you don't use the same framework than us.

*(21) [Line 468] As before, this 'posterior ensemble' is a bit confusing as you are looking at the distribution of ice mass discharge, rather than the distributions of the inferred parameters (which have been analyzed in the previous subsection). Maybe use another name for this subsection, or precise in that name that you are going to talk about SLR predictions.*

To clarify the overall structure, we have removed this sub-title (see answer to main comments).

*(22) [Figure 5] This figure is difficult to read. Consider using brighter colors and larger labels.*

Agreed, we have added hatch patterns and increase the label fontsize.

*(23) [Line 553] It's → It is.*

Thanks, we have corrected it.

*(24) [Line 595] dynamics modeling community → ice-sheet dynamics modeling community?*

Thanks, we have corrected it.

*(25) [Line 660] it's → it is.*

Thanks, we have corrected it.

*(26) [Line 746] Would that still be true if you looked beyond 2100? I have in mind the study of Brondex et al. (2017, 2019) which showed that the form of friction laws does have a strong impact (in particular, there is a significant difference between the regularized Coulomb and Budd laws).*

To make clearer our message about the past reproduction of observations, we added few words at the end of the sentence: «[...] in reproducing past acceleration of UI».

Otherwise, to answer the question, we think than our figure B1 gives some keys: the impact of the shape of the friction law does not have a major impact in 2100 unlike the use of the parameterisation developped in Jager et al. (2024). In our point of view, the main difference with Brondex et al. is this choice of proxy for the effective pressure N. In their case, they assume a perfect connection to the ocean with N proportional to the height above flotation, while Jager et al. (2024) and Joughin et al. (2019) both use a cut-off (the distance to the front in our case, the height above flotation in the case of Joughin). It means than far enough to the grounding line/the front position, the friction at the base is independent to the height above flotation/distance to the front regardless to the friction law used. On the contrary for Brondex et al., in case of Budd law, it will lead to a dependence to the effective pressure everywhere. However, this will not be the case for a

regularised coulomb law because you will have some areas in a Weertman regime where friction only depends on the friction parameter $A_s$.

*(27) [Line 789] Maybe add that 'SSA' also stands for Shallow-Shelf Approximation.*

Agreed, we have added the following words: «also called Shallow-Shelf Approximation»

*(28) [Lines 811, 843, 876] regularisation → regularization.*

Thanks, we have corrected it.

*(29) [Line 840] Maybe add that this value of u0 is similar to the one chosen in Joughin et al. (2019).*

Agreed, we have added the following words: «[...], which is similar to the median value of our previous study and to the one used in Joughin et al. (2019).»

*(30) [Line 884] law → flaw .*

Thanks, we have corrected it.

---

## Author Response (AR2)

We would like to thank Benjamin Smith for his corrections, which have improved this new version of the manuscript. Original editors comments are in italic and black, our answers are in blue. We have prepared a new version of the manuscript taking into account the reviewers' comments, as well as a pdf outlining the changes.

*Line 7: "While all sources of uncertainty contribute at least 15% to uncertainty until the end of the century, SSp-related uncertainty dominates at 40%" – this sentence is not at all clear. 15% of what? Which undercertainty? If it's mass loss, AOGCM seems to contribute nothing at the end of the century. Please clarify or delete.*

We've changed the sentence to «At the end of the century, SSP-related uncertainty constitutes the predominant component of total uncertainty, accounting for 40\%, while uncertainty linked to the ISM represents 15\% of the overall uncertainty.»

*Line 30 : Addition of -> addition to*

Thanks, corrected

*Line 42: as evidenced by -> as is evident from*

Agreed, corrected

*58: Please justify or delete "providing more robust results." This needs to be demonstrated, not just asserted.*

We've changed the sentence to « This approach mitigates the risk of over-interpretation that may arise when focusing on a single tidewater glacier, providing more robust results; i.e., if the model successfully reproduces these varied behaviors, it is likely to do so for other tidewater glaciers as well. »

*110: Please add a reference to appendix A to help the reader understand what the model parameters are*

We've added the following sentence : « See appendix A to understand what are the ISM parameters. ».

*115: model led -> modelled*

Thanks, corrected

*119: observations from wood et al (2021) -> observations (Wood et all, 2021)*

Agreed, corrected

*134: mitigate appears twice*

Thanks, corrected

*141: check volume unit notation*

I've checked in the paper of Slater et al. (2019) - e.g. Fig. 4 - and the unit of Q is $m^3 .s^{-1}$.

*156: Would it make more sense to say "the uncertain parameters related to the SSP do not affect Hpr and Cpr"?*

The sentence was a bit confused, we've changed it to : « As defined by our set-up, the uncertain parameters of the forcing (SSP, AOGCM, RCM, fronts) do not affect Hpr and Cpr (Fig. 2). »

*184: The notation for the Heaviside function would be clearer if it were written F0(Q; Q0) or F0(Q, Q0), or even F0(Q-Q0) (to match the standard definition of a Heaviside function)*

Agreed, we've changed it to F0(Q, Q0).

*203: "data with" -> "data for"*

Thanks, corrected

*207: check notation on nobs*

Thanks, corrected

*217: delete 'considerable"*

Thanks, corrected

*217: observations is necessary -> observations would be necessary*

Agreed, corrected

*229: change to parenthetical reference : "as previously identified (wernecke et al, 2020)."*

Agreed, corrected

*244: Please justify or provide a citation for the statement that "the minimum RMS corresponds to the configuration with the least structural error." Is this statement remarkable because it has been discovered in this study?*

When we wrote it, it seemed logical, but now we are not sure and we haven't found a good justification. So we've changed this sentence to: « Specifically, we will evaluate the minimum, median, and maximum RMSE values as potential estimates for σ. »

*Line 318 / Figure 5: This figure is too small in the inset version. The tick lines would be helpful here, but they are too light to see.*

Agreed, we have increased labels and legend and made the tick lines darker. For consistency, we have also made the tick lines of Fig. 4 darker.

*356 "which demonstrate the most pronounced impact"- this seems redundant to the first clause of the sentence. Please either give a numerical description of the impact, or delete this phrase.*

Agreed, we've deleted this part of the sentence.

*377: provide a name for the m parameter*

Agreed, we've replaced « parameter » by «  exponent of the friction law ».

*Figure 7: This figure is also too small and too faint. Please make the font larger and the lines bolder*

Agreed, we've increased the size of the legends and made the lines more visible.

*412: It would be good if, before this point, some explanation had been provided about what happens when you try to derive calibration parameters in areas where the ice front has already retreated (I think this is the problem under discussion here). This is a problem with the way that the model is calibrated using the later time periods, and it would be good to acknowledge earlier in the paper that it causes difficulties. Please find a place to add this to the text.*

Agreed, we've added the explanation of how we extrapolate the friction in ice-free regions where we describe the calibration of the friction field (Appendix C1) : « Due to the presence of ice-free regions in certain areas of the glacier following its retreat around 2005, it is imperative to extrapolate the friction field in these regions for members using observational data from 2005-2015 or 2015-2019. When the parameterization of the effective pressure change effect is enabled (i.e.,$f^{param}$ = True), the time-independent reference field $\beta_{ref}$ is set to 0 in the extrapolated areas. Conversely, when parameterization is disabled ($f_{param}$ = False), either $\beta_W$ or $\beta_{RC}$ are set to 0 is utilized. The implications of the chosen extrapolation method are further examined and discussed in Jager et al. (2024). »

*442: Please explain what "lower members" are.*

We've changed this part of the sentence to « while retaining the inclusion of members with a lower mass loss, which are members using inversion data before the retreat. »

*Figure 8: Again, the labels on the axes are too small, and the lines are too faint.*
*Caption to figure 8 : please refer to panels by letter, rather than by "top left" etc.*

Agreed, we've increased the size of the legends and made the lines more visible. We've also changed the caption to refer to panels by letter and a sentence on sub-plots. In the one-column format, the figure takes up more space, so we've reduced its size a little, but it would be possible to increase its size in the two-column format. The same applies to figure D1.

*475, and the remainder of this section "The SSP weighting on the future prediction has a very significant effect" and similar statements: It would be good to establish a standard for what*

*changes are significant. For example, is it significant that a confidence interval changes by ~10% of its previous value? It would also be good to use expressions like "essentially unchanged" when values do not change very much (e.g. the medians in lines 477, 483, and 486).*

We've used the term significant in this sentence because we had a change of the median by around 20%. We are agree to caracterize the change of line 483 can be written as 'essentially unchanged' and we have rewritten the sentence to: « Moreover, the combination remains essentially unchanged with the median shifting upwards from 0.79 mm to 0.80 mm. ». For line 477, the change does not appear to be 'unchanged' so we have removed the slight (median change of around 10%). For line 486, the change seems slight (just under 5%) so we have not replaced the term with 'unchanged'.

*Figure 10: Label panel with letters, and add references in the text to each panel.*

We added letters to the panels and rewrote the caption and text accordingly.

*493: Please check the use of "significantly"*

We have added the percentage of change, which is over 20% and therefore considered significant.

*497: Please be specific about what the panels in figure 10 to which you are referring, and explain the significance of "a shift to higher probabilities"*

Thanks, we have added letters to fig. 10 and the reference to it in the text. We've changed the term « a shift to higher probabilities » to « a notable shift in probability towards higher values ».

*533: delineate -> describe*

Thanks, corrected

*565: check use of "significant"*

We have added percentage a mention to the reduction of the 95\% confidence interval : « with a 20\% reduction in the 95\% confidence interval ». In this paragraph, we have also added 1 sentence at the end : « unless other sources of uncertainty are addressed first. Specifically, after reducing the uncertainty associated with the SSPs, applying a weighting to the ISM further reduces uncertainty by 10% by 2100 (Ppo in fig. 8.j and 8.a). »

*580: Explain what is influenced by front position*

Agreed, we've added: « on the ice discharge and the mass loss of UI »

*619: bay -> embayment*

Thanks, corrected